# Evolution of hind limb morphology of Titanosauriformes (Dinosauria, Sauropoda) analyzed via 3D geometric morphometrics reveals wide-gauge posture as an exaptation for gigantism

**Adrián Páramo[1,2,3]\*, Pedro Mocho[3,4], Fernando Escaso[3], Francisco Ortega[3]**

[1]Scientific Computation Research Institute (SCRIUR), Universidad de La Rioja, Logroño, Spain; [2]Centro de Interpretación Paleontológica de La Rioja, La Rioja, Spain; [3]Grupo de Biología Evolutiva (GBE), Universidad Nacional de Educación a Distancia, Las Rozas, Spain; [4]Instituto Dom Luiz, Faculdade de Ciências, Universidade de Lisboa, Lisboa, Portugal

## eLife Assessment

The authors present **convincing** findings on trends in hind limb morphology through the evolution of titanosaurian sauropod dinosaurs, the land animals that reached the most remarkable gigantic sizes. The **important** results include the use of 3D geometric morphometrics to examine the femur, tibia, and fibula to provide new information on the evolution of this clade and on evolutionary trends between morphology and allometry.

**\*For correspondence:**
paramoblazquez@gmail.com

**Competing interest:** The authors declare that no competing interests exist.

## Abstract

The sauropod hind limb was the main support that allowed their gigantic body masses and a wide range of dynamic stability adaptations. It was closely related to the position of the center of masses of their multi-ton barrel-shaped bodies and experienced one of the most noticeable posture changes during macronarian evolution. Deeply branched macronarians achieved increasingly arched hind limbs in what is known as wide-gauge posture. However, it is not clear if this evolutionary trend is related to the evolutionary cascade toward gigantism even though some titanosaurians were the largest terrestrial vertebrates that ever existed. We tested evolutionary changes in hind limb morphology in the Macronaria phylogenetic tree by 3D geometric morphometrics. The macronarian hind limb does become progressively more arched toward deeply branched groups, specifically Saltasauridae. However, there is morphological convergence between different macronarian subclades. Wide-gauge posture does not correlate with changes in body size deeper in the macronarian evolutionary tree and acted as an exaptation to gigantism. Despite some titanosaurian subclades becoming some of the largest vertebrates, there is not a statistically significant trend toward a particular body size, but we identify a phyletic body size decrease in Macronaria.

## Introduction

Sauropod dinosaurs evolved a distinct body plan that allowed them to reach some of the largest body masses of terrestrial vertebrates (e.g. *Bonaparte and Coria, 1993*; *Carballido et al., 2017*; *Lacovara et al., 2014*; *Sander, 2013*; *Sander et al., 2011a*) by one order of magnitude compared to other

vertebrates, including other megaherbivore dinosaurs (e.g. *Maher et al., 2022*). Sauropods were dominant herbivorous dinosaurs throughout much of the Mesozoic (*Sander, 2013*; *Sander et al., 2011a*). Several morphological features can be related to the evolutionary cascade that allowed the acquisition of their colossal sizes, such as the vertebral pneumaticity related to avian-like air sacs, high metabolic rates, cranial morphology, and feeding mechanisms (including the characteristic long necks among several other traits, see *Sander, 2013*). Several features on its appendicular skeleton are related to the evolution of columnar limbs. The mechanical stability of the columnar limbs allowed them to support their multi-ton body masses (*Bates et al., 2016*; *Lefebvre et al., 2022*; *Salgado et al., 1997*; *Sander, 2013*; *Ullmann et al., 2017*).

Previous studies based on sauropod anatomical description, systematics, traditional morphometrics, biomechanics, and GMM suggested that there was an important evolutionary trend in the appendicular skeleton toward the acquisition of a stable limb posture known as wide-gauge (*Klinkhamer et al., 2019*; *Lefebvre et al., 2022*; *Salgado et al., 1997*; *Sander, 2013*; *Ullmann et al., 2017*; *Upchurch et al., 2004*; *Voegele et al., 2021*; *Wilson and Carrano, 1999*; *Wilson, 2002*). This feature appears among deeply branched Neosauropoda, in particular, among titanosauriforms, which exhibit a progressively arched limb posture (*Salgado et al., 1997*; *Upchurch et al., 2004*; *Voegele et al., 2021*; *Wilson and Carrano, 1999*; *Wilson, 2002*). The wide stance may enable enhanced lateral stability during locomotion, allowing them to exploit more efficiently inland environments (*Mannion and Upchurch, 2010*; *Ullmann et al., 2017*). However, the widening of the body in Titanosauriformes and the acquisition of the wide-gauge stance is still poorly understood. Although the arched limbs and wider postures appeared in the largest sauropods among Titanosauriformes (*Bates et al., 2016*; *Henderson, 2006*; *Ullmann et al., 2017*; *Wilson and Carrano, 1999*), these features may at least be acquired independently among both small and large titanosauriforms of different subclades (*Henderson, 2006*; *Bates et al., 2016*). Limb posture may have been related to achieving greater ranges of ecological niches through biomechanical stability rather than allowing increasing their body size itself (e.g. *Bates et al., 2016*; *Henderson, 2006*; *Ullmann et al., 2017*). The study of the hind limb through 2D and 3D geometric morphometrics (GMM) allows us to analyze complex morphological changes across the titanosauriformes phylogeny (e.g. *Lefebvre et al., 2022*; *Ullmann et al.,*

**Table 1.** Specimen sample used in this study.
Proximodistal hind limb length measured in mm.

| | Clade | Hind limb length (mm) |
|---|---|---|
| *Aeolosaurus* | Aeolosaurini | 1839 |
| *Ampelosaurus* | Lirainosaurinae | 1411 |
| *Antarctosaurus* | Titanosauria | 2351 |
| *Bonatitan* | Lithostrotia | 963 |
| *Bonitasaura* | Lithostrotia | 1987 |
| *Dreadnoughtus* | Titanosauria | 3184 |
| *Euhelopus* | Euhelopodidae | 1683 |
| *Jainosaurus* | Titanosauria | 2188 |
| *Ligabuesaurus* | Somphospondyli | 2866 |
| *Lirainosaurus* | Lirainosaurinae | 1074 |
| *Lohuecotitan* | Lithostrotia | 1120 |
| *Magyarosaurus* | Lithostrotia | 750 |
| *Mendozasaurus* | Colossosauria | 2334 |
| *Muyelensaurus* | Colossosauria | 1450 |
| *Neuquensaurus* | Saltasauridae | 1200 |
| *Oceanotitan* | Macronaria | 1892 |
| *Saltasaurus* | Saltasauridae | 1230 |

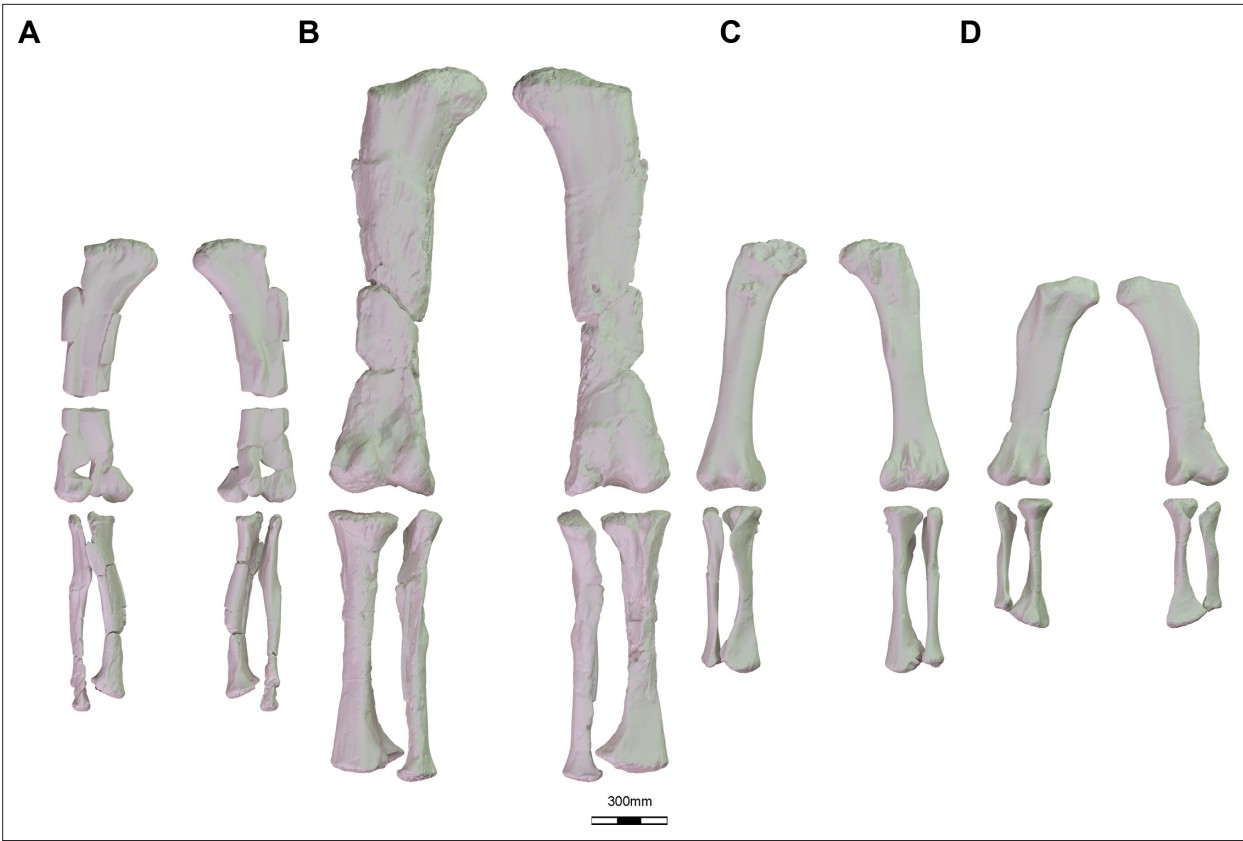

**Figure 1.** Sample of several 3D reconstruction of macronarian hind limbs used in this study. (**A**) *Oceanotitan dantasi* in anterior and posterior view; (**B**) *Ligabuesarusu leanzai* in posterior and anterior view; (**C**) *Lohuecotitan pandafilandi* in anterior and posterior view; (**D**) *Saltasaurus loricatus* in posterior and anterior view.

*2017*). Here, we will use 3D-digitized and reconstructed titanosauriform hind limbs (*Table 1*, see also *Figure 1*) to test whether there is a relationship between wider hind limb posture (as the hind limbs are the main weight support of the sauropod body, see *Bates et al., 2016*) and their body size, as proxied by the hind limb size. We chose the centroid size of the hind limb because the size of the femur and tibia correlates well with sauropod body mass (*Campione and Evans, 2020*; *Mazzetta et al., 2004*), but alternative tests using femoral length or body mass estimations are provided in Appendix 1-2 and *Supplementary file 2*. We will also test for potential morphological convergence between different titanosauriform subclades regardless of their body size. The analysis presented here can be seen as an expansion of the *Lefebvre et al., 2022* study on sauropod limb evolution as we analyze a sample comprised mostly of Late Cretaceous lithostrotian sauropods (their study included a broad diversity of sauropodomorph taxa including several titanosaurs).

## Results

### Titanosauriformes morphospace occupation

A Principal Component Analysis (PCA) was performed to generate an occupation morphospace, obtaining a total of 16 shape Principal Components (after Anderson's $\chi$ test; PCs from now on). The first six PCs accounted for 78.3% of the cumulative morphological variation (*Table 2*). The non-parametric Kruskal-Wallis test shows that no single shape variable reports significant differences among sauropod subclades (*Tables 3 and 4*). Here, we only comment on the results of the first three shape PCs (>50% of the cumulative variance) due to space limitations, but a full description and visualization of the complete PCA and phylomorphospace projections can be found in Appendix 2.

PC1 (summarizing 29.92% of the total variance; *Figure 2a–b*) is associated with characters describing the orientation and compression of the femoral shaft, the length of the distal femoral condyles, the

**Table 2.** PCA results over GPA-aligned coordinates.
Variance explained by each shape PC.

|  | Explained variance (%) | Cumulative variance (%) |
| --- | --- | --- |
| PC1 | 29.92 | 29.92 |
| PC2 | 14.83 | 44.75 |
| PC3 | 11.61 | 56.36 |
| PC4 | 8.83 | 65.19 |
| PC5 | 7.02 | 72.21 |
| PC6 | 6.09 | 78.3 |
| PC7 | 4.18 | 82.48 |
| PC8 | 3.75 | 86.23 |
| PC9 | 3.12 | 89.35 |
| PC10 | 2.19 | 91.53 |
| PC11 | 2.15 | 93.68 |
| PC12 | 1.95 | 95.64 |
| PC13 | 1.56 | 97.19 |
| PC14 | 1.18 | 98.38 |
| PC15 | 0.92 | 99.29 |
| PC16 | 0.71 | 100 |

orientation of the fourth trochanter, and the length and width of the zeugopod bones (*Figure 2c*). Non-titanosaurian macronarians occupy negative values, whereas non-lithostrotian titanosaurs are distributed between weakly negative (e.g. *Bonatitan*) and positive PC1 values (*Figure 2b*). Lithostrotian titanosaurs occupy a wide range of values, with *Aeolosaurus*, the specimens of Colossosauria and

**Table 3.** Kruskal-Wallis test on shape PCA variables between the most inclusive subclades analyzed.

|  | Chi-sq | p-value | p-adjusted |
| --- | --- | --- | --- |
| PC1 | 11.954 | 0.153 | 1 |
| PC2 | 5.592 | 0.693 | 1 |
| PC3 | 7.886 | 0.445 | 1 |
| PC4 | 9.647 | 0.291 | 1 |
| PC5 | 5.17 | 0.739 | 1 |
| PC6 | 7.618 | 0.472 | 1 |
| PC7 | 8.915 | 0.35 | 1 |
| PC8 | 8.941 | 0.347 | 1 |
| PC9 | 10.206 | 0.251 | 1 |
| PC10 | 10.422 | 0.237 | 1 |
| PC11 | 4.768 | 0.782 | 1 |
| PC12 | 5.66 | 0.685 | 1 |
| PC13 | 8.886 | 0.352 | 1 |
| PC14 | 9.856 | 0.275 | 1 |
| PC15 | 10.248 | 0.248 | 1 |
| PC16 | 4.578 | 0.802 | 1 |

**Table 4.** Phylogenetic ANOVA test on shape PCA variables between the most inclusive subclades studied.

| | | Df | SS | MS | R² | F | Z | Pr(>F) |
|---|---|---|---|---|---|---|---|---|
| PC1 | ~Clade | 8 | 0.073 | 0.009 | 0.466 | 0.872 | –0.105 | 0.531 |
| | Residuals | 8 | 0.083 | 0.010 | 0.533 | - | - | - |
| | Total | 16 | 0.157 | - | - | - | - | - |
| PC2 | ~Clade | 8 | 0.001 | 1.50E+04 | 0.685 | 0.218 | 0.573 | 0.313 |
| | Residuals | 8 | 5.52E+04 | 6.90E+09 | 0.314 | - | - | - |
| | Total1 | 16 | 0.002 | - | - | - | - | - |
| PC3 | ~Clade | 8 | 1.54E+04 | 1.93E+09 | 0.285 | 0.399 | –0.925 | 0.82 |
| | Residuals | 8 | 3.86E+04 | 4.83E+08 | 0.714 | - | - | - |
| | Total | 16 | 5.40E+04 | - | - | - | - | - |
| PC4 | ~Clade | 8 | 1.49E+04 | 1.87E+09 | 0.3534 | 0.546 | –0.496 | 0.658 |
| | Residuals | 8 | 2.73E+04 | 3.41E+09 | 0.646 | - | - | - |
| | Total | 16 | 4.22E+04 | - | - | - | - | - |
| PC5 | ~Clade | 8 | 3.77E+04 | 4.72E+09 | 0.556 | 125.362 | 0.121 | 0.452 |
| | Residuals | 8 | 3.01E+04 | 3.77E+09 | 0.443 | - | - | - |
| | Total | 16 | 6.79E+04 | - | - | - | - | - |
| PC6 | ~Clade | 8 | 3.24E+04 | 4.06E+09 | 0.590 | 144.158 | 0.590 | 0.264 |
| | Residuals | 8 | 2.25E+04 | 2.82E+09 | 0.409 | - | - | - |
| | Total | 16 | 5.50E+04 | - | - | - | - | - |
| | ~Clade | 8 | 1.63E+04 | 2.05E+09 | 0.434 | 0.768 | –0.315 | 0.625 |
| | Residuals | 8 | 2.13E+04 | 2.66E+09 | 0.565 | - | - | - |
| | Total | 16 | 3.76E+04 | - | - | - | - | - |

*Lirainosaurus* clustering at negative PC1 values (however, *Ampelosaurus* occupies negative PC1 values approaching zero; *Figure 2b*). Most of the non-saltasaurid, non-colossosaurian, non-lirainosaurine, and non-aeolosaurine lithostrotians are broadly distributed in the morphospace between negative and positive values, with *Bonitasaura* occupying the farthest negative PC1 values and overlapping with Colossosauria and Lirainosaurinae (*Figure 2b*). The highest positive PC1 values are occupied by saltasaurids (i.e. *Saltasaurus* and *Neuquensaurus*; *Figure 2b*). There is a trend from non-titanosaurian macronarians at negative values to the titanosaurian node at weakly negative values (*Figure 2b*). In this PC1, the Titanosauria node splits near a zero score, with *Lohuecotitan* occupying weakly positive PC1 values near a zero score [in recent phylogenetic analyses, *Lohuecotitan* has been recovered as a member of Lithostrotia (*Díez Díaz et al., 2021a*; *Navarro et al., 2022*; *Mocho et al., 2024b*)], but other deeply nested titanosaurs occupy positive scores in PC1 (*Figure 2b*). The trend toward positive values follows with several other deeply nested lithostrotians. However, both Colossosauria and the two analyzed members of Lirainosaurinae fall into negative PC1 values (Lirainosaurinae was not recovered as a monophyletic group in our current topology *Figure 2b*). There are no significant differences between the different sauropod subclades in this PC (*Supplementary file 2*).

PC2 (summarizing 14.83% of the variance; *Figure 3a–b*) is associated with characters describing the orientation of the femoral shaft; the length and orientation of the femur proximal end; the length, width, and orientation of the distal femoral condyles; the orientation of the tibial shaft; the length and orientation of the cnemial crest; the length, width, and orientation of the tibial distal end; the orientation of the tibial ascending process; the orientation of the fibular shaft and fibular anterior crest (*Figure 3c*). Both the early branching macronarian *Oceanotitan* - a possible member of Sompho-spondyli (*Mocho et al., 2019b*) - and the euhelopodid somphospondyli *Euhelopus* occupy positive and negative values near a zero score, whereas the early-diverging somphospondylan *Ligabuesaurus*

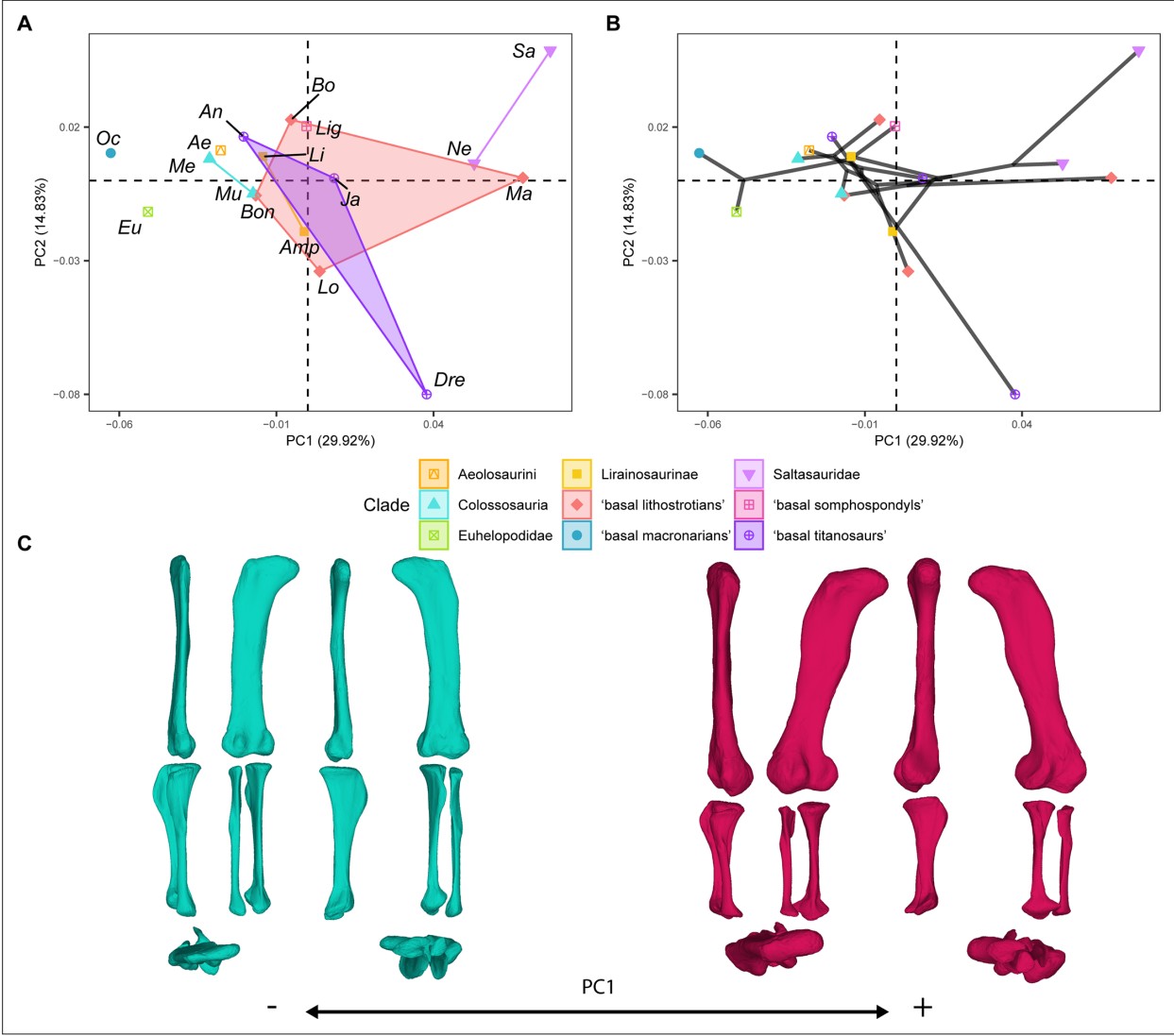

**Figure 2.** PCA results on the GPA aligned landmark and semilandmark curves of the hind limb. (**A**) PC1-PC2 biplot. (**B**) PC1-PC2 phylomorphospace with projected phylogenetic tree. (**C**) Representation of the shape change along PC1, blue are negative scores, red are positive scores. Percentage of variance of each PC in brackets under corresponding axis. *Ae – Aeolosaurus, Amp – Ampelosaurus, An – Antarctosaurus, Bo – Bonatitan, Bon – Bonitasaura, Dre – Dreadnoughuts, Eu – Euhelopus, Ja – Jainosaurus, Li – Lirainosaurus, Lig – Ligabuesaurus, Lo – Lohuecotitan, Ma – Magyarosaurus, Me – Mendozasaurus, Mu – Muyelensaurus, Ne – Neuquensaurus, Sa – Saltasaurus.*

is distributed at slightly more positive PC2 values than *Oceanotitan* (***Figure 3b***). However, there is no clear pattern among the titanosaurian subclade morphospace (***Figure 3a***) or the phylomorphospace (***Figure 3b***) among progressively deeply branching titanosauriforms. The saltasaurids and *Aeolosaurus* occupy positive PC2 scores (***Figure 3b***) showing some overlap with Colossosauria at weakly positive values of PC2 (***Figure 3b***). Lirainosaurines and colossosaurians are distributed at weakly negative to positive values in PC2 (***Figure 3b***). However, non-saltasaurid, non-aeolosaurine, non-colossosaurian, and non-lithostrotian titanosaurs are broadly distributed across the morphospace; most of them between negative values (i.e. *Dreadnoughtus*) and positive PC2 values (i.e. *Jainosaurus*, *Antarctosaurus*; ***Figure 3a–b***). The hind limb occupation of the titanosaur *Dreadnoughtus* at strongly negative PC2 values is noteworthy (***Figure 3a–b***). The Mann-Whitney U test found no significant differences in the PC2 in any of the pairwise comparisons (***Supplementary file 2***).

Finally, PC3 (summarizing 11.61% of the variance; ***Figures 3a–b and 4a–b***) is related to characters describing the orientation of the femoral shaft and head; the length of the femoral lateral bulge; the location and orientation of the fourth trochanter; the orientation of the distal femoral condyles; the

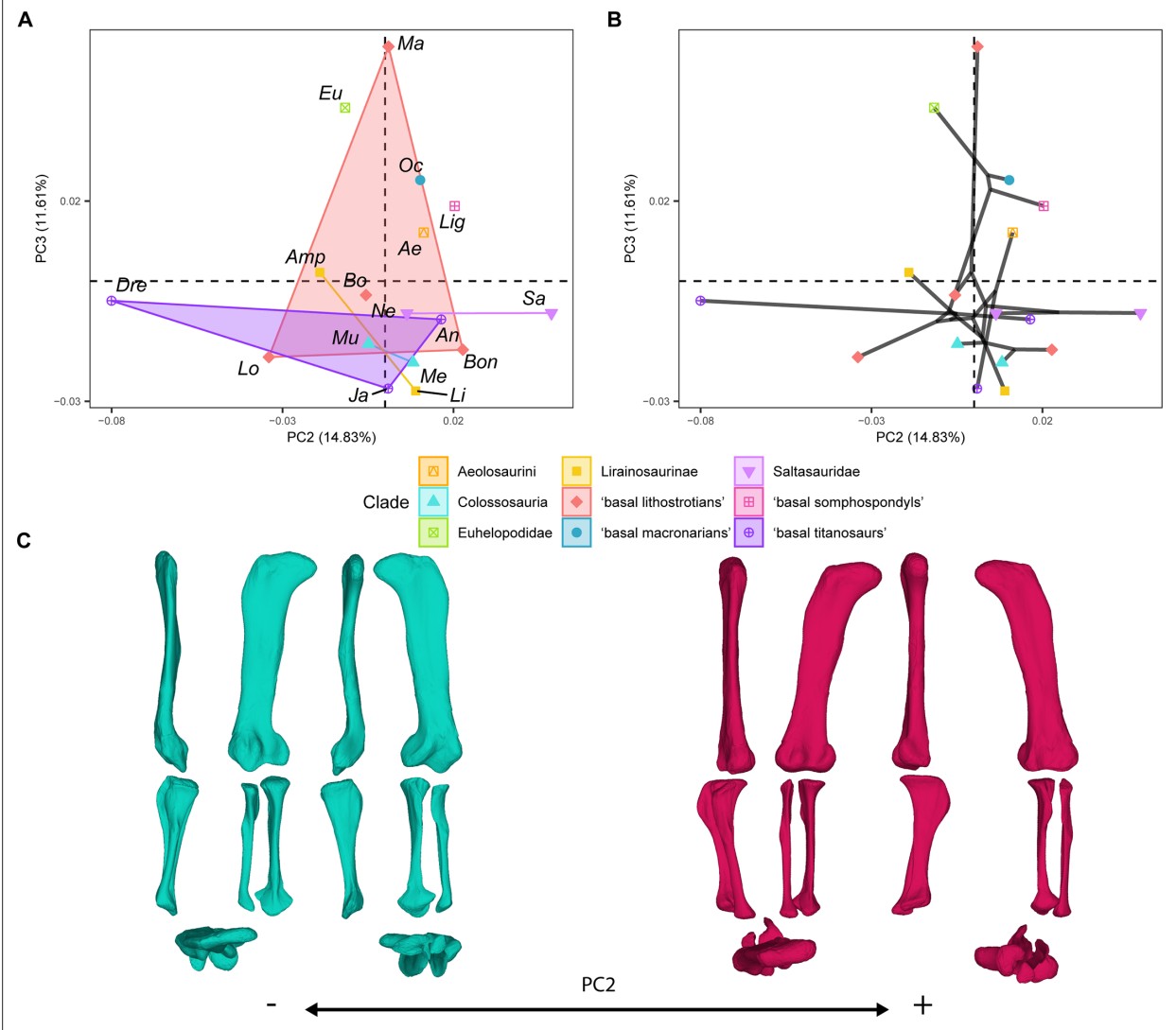

**Figure 3.** PCA results for the GPA aligned landmark and semilandmark curves of the hind limbs. (**A**) PC2-PC3 biplot. (**B**) PC2-PC3 phylomorphospace with projected phylogenetic tree. (**C**) Representation of the shape change along PC2, blue are negative scores, red are positive scores. Percentage of variance of each PC in brackets under corresponding axis. *Ae – Aeolosaurus, Amp – Ampelosaurus, An – Antarctosaurus, Bo – Bonatitan, Bon – Bonitasaura, Dre – Dreadnoughtus, Eu – Euhelopus, Ja – Jainosaurus, Li – Lirainosaurus, Lig – Ligabuesaurus, Lo – Lohuecotitan, Ma – Magyarosaurus, Me – Mendozasaurus, Mu – Muyelensaurus, Ne – Neuquensaurus, Sa – Saltasaurus.*

length and orientation of the tibial cnemial crest; the length of the tibial distal end; the length of the tibial ascending process; and the length and morphology of the fibula (***Figure 4c***). *Oceanotitan* and *Euhelopus* were plotted toward progressively more positive values (***Figure 4a***), and the node was estimated at slightly more negative values of PC3 than *Euhelopus* (***Figure 4b***). *Ligabuesaurus* occupies positive values of PC3 closer to zero than early branching titanosauriforms (***Figure 4a***). In the phylomorphospace, there is a trend from positive values to negative values of PC3 (***Figure 4b***) from more basally branching titanosauriforms to more deeply nested ones. Non-lithostrotian titanosaurs occupy negative PC3 values, but the non-saltasaurid, non-lirainosaurine, and non-colossosaurian lithostrotians trend toward positive PC3 values (***Figure 4a–b***), whereas Lirainosaurinae, Colossosauria, and Saltasauridae occupy negative values of PC3 (***Figure 4a–b***). Colossosauria, Saltasauridae, and *Lirainosaurus*, the most deeply branched representatives of Lirainosaurinae according to our phylogenetic hypothesis, plotted toward increasingly more negative PC3 values (***Figure 4a–b***). *Magyarosaurus* is the only lithostrotian occupying high positive values, clearly separate from all other sauropods (***Figure 4a–b***).

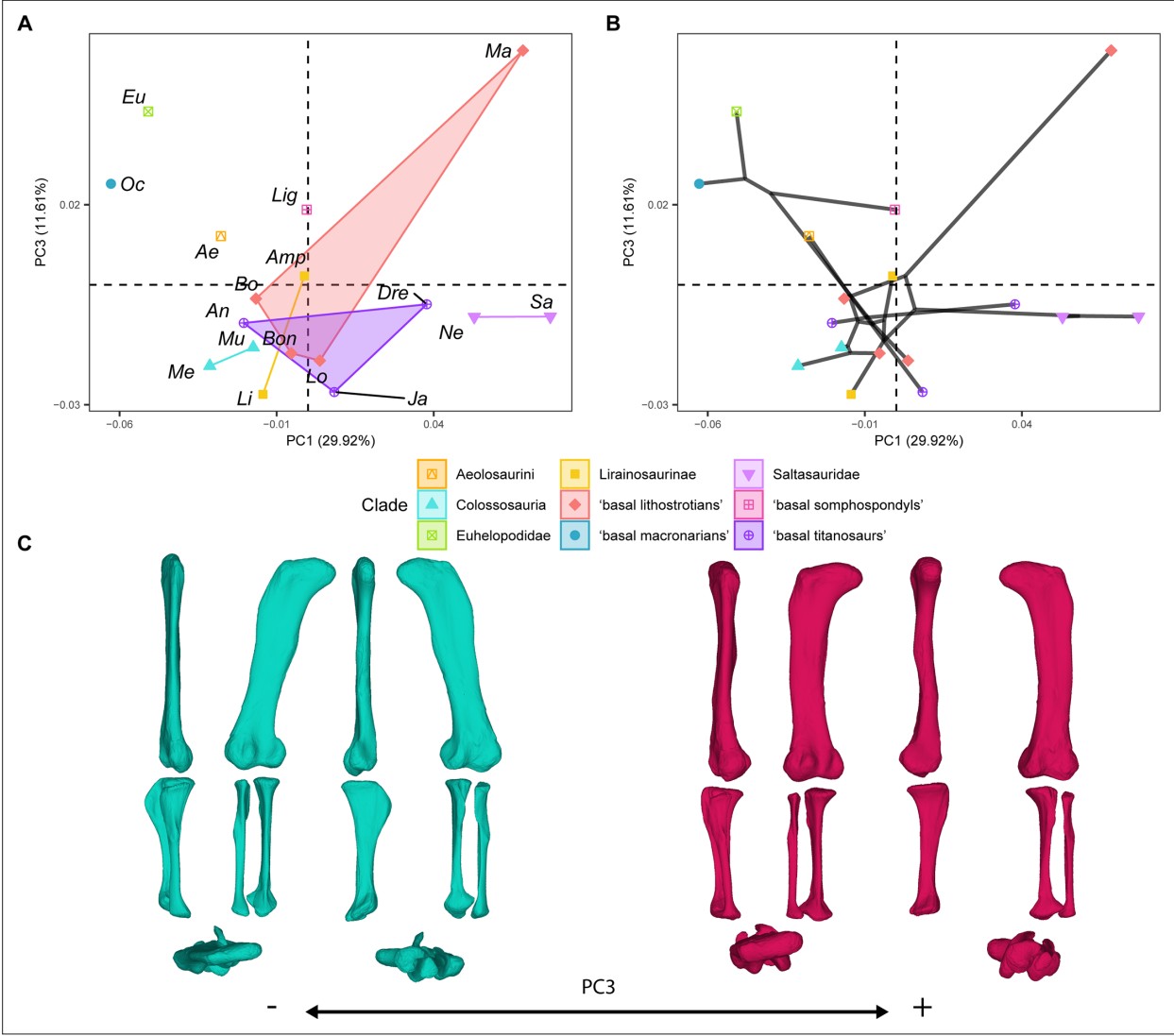

**Figure 4.** PCA results for the GPA aligned landmark and semilandmark curves of the hind limbs. (**A**) PC1-PC3 biplot. (**B**) PC1-PC3 phylomorphospace with projected phylogenetic tree. (**C**) Representation of the shape change along PC3, blue are negative scores, red are positive scores. Percentage of variance of each PC in brackets under corresponding axis. *Ae – Aeolosaurus, Amp – Ampelosaurus, An – Antarctosaurus, Bo – Bonatitan, Bon – Bonitasaura, Dre – Dreadnoughtus, Eu – Euhelopus, Ja – Jainosaurus, Li – Lirainosaurus, Lig – Ligabuesaurus, Lo – Lohuecotitan, Ma – Magyarosaurus, Me – Mendozasaurus, Mu – Muyelensaurus, Ne – Neuquensaurus, Sa – Saltasaurus.*

The pair-wise Mann-Whitney U test found no significant differences between sauropod subclades in this PC (***Supplementary file 2***).

## Size distribution

There is a trend in hind limb centroid size distribution from large, non-titanosaurian macronarians in the Early Cretaceous (e.g. *Euhelopus*) to small, deeply nested lithostrotian titanosaurs in the Late Cretaceous (***Figure 5***). This trend coincides with the morphospace occupation recovered by PC1 early-branching and larger non-titanosaurian macronarians among negative PC1 values and progressively more deeply nested and smaller titanosaurs toward positive PC1 values (***Figure 5***). The smallest lithostrotians are concentrated at deeply branching nodes, including the Ibero-Armorican lirainosaurines, *Bonatitan reigi*, *Magyarosaurus* spp. and members of Saltasaurinae (***Figure 5***).

However, the RMA models found no significant correlation between the shape variables and the log-transformed centroid size (***Table 5***, see ***Figure 6*** and Appendix 2). The PC1 model ($r^2$=0.105, p-value = 0.204; ***Figure 6a***) found negative allometry but no significant correlation and the percentage

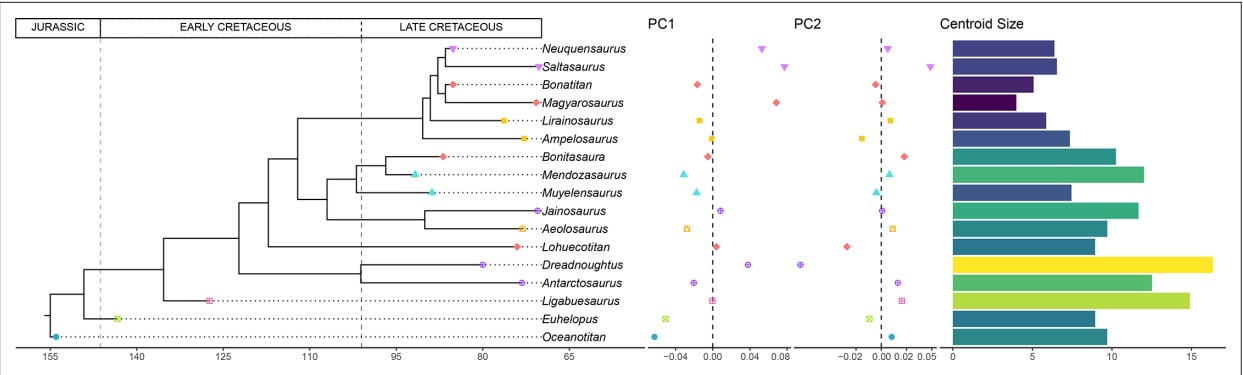

**Figure 5.** Time-calibrated supertree with PC1-PC2 results and hind limb centroid size for each sauropod.

of variance explained by hind limb size differences was small. Almost all of the RMA found a negative relationship, except for several sub-sampled RMA models like the PC2 against log-transformed centroid size for the sample of lithostrotian titanosaurs only (*Figure 6b*). None of the RMA models for the sub-samples found a significant correlation (see Appendix 2).

## Phylogenetic trends

Pagel's lambda ($\lambda$) estimation shows a significant phylogenetic signal in log-transformed hind limb centroid size ($\lambda$ =0.982), and PC1 ($\lambda$ =0.715), PC3 ($\lambda$ =0.760), PC5 ($\lambda$ =0.778) and PC6 ($\lambda$ =0.697) therefore exhibiting a trend in the evolution of Titanosauriformes (*Table 6*). We estimated ancestral characters (ACEs) using log-transformed hind limb centroid size and those shape variables (the PCs) that exhibit a significant signal during the evolution of Titanosauriformes and tested for a directionality or trend (*Figure 7*). The resulting tests recover significant trends toward a decrease in hind limb size across all titanosauriform subclades, with positive PC1 values including somphospondyli titanosauriformes, and negative PC3 values across all titanosauriform subclades (*Supplementary file 2*).

## Discussion
### Hind limb morphological convergence in Titanosauriformes

Analysis of the shape variables extracted by PCA on the Procrustes coordinates of the sauropod taxa reveals a large overlap between the different titanosauriform subclades, in particular within Titanosauria (e.g. *Figures 2–4*), across all the resulting shape PCs. Both the non-parametric tests on the hind limb shape variables and the size, and the phylogenetic ANOVA accounting for the time-calibrated supertree topology, suggest the lack of sufficient and significant morphological differences between the different titanosauriform subclades studied in this analysis. Based on the lack of significant phylogenetic differences and the presence of morphological similarities, the evolutionary pattern observed for the titanosaurian hind limb may be explained by convergent evolution, consistent with previous analyses (*Lefebvre et al., 2022*; *Páramo, 2020*; *Ullmann et al., 2017*).

**Table 5.** RMA models of the shape PCs against log-transformed Centroid size. CI – confidence interval.

|  | intercept | slope | CI 2.5% Slope | CI 97.5% Slope | r² | P |
|---|---|---|---|---|---|---|
| PC1 | 0.221 | –0.102 | –0.168 | –0.062 | 0.105 | 0.204 |
| PC2 | 0.155 | –0.072 | –0.12 | –0.043 | 0.054 | 0.371 |
| PC3 | 0.137 | –0.064 | –0.106 | –0.038 | 0.055 | 0.363 |
| PC4 | 0.12 | –0.055 | –0.093 | –0.033 | 0.026 | 0.534 |
| PC5 | –0.107 | 0.049 | 0.03 | 0.082 | 0.079 | 0.275 |
| PC6 | –0.1 | 0.046 | 0.028 | 0.076 | 0.086 | 0.254 |

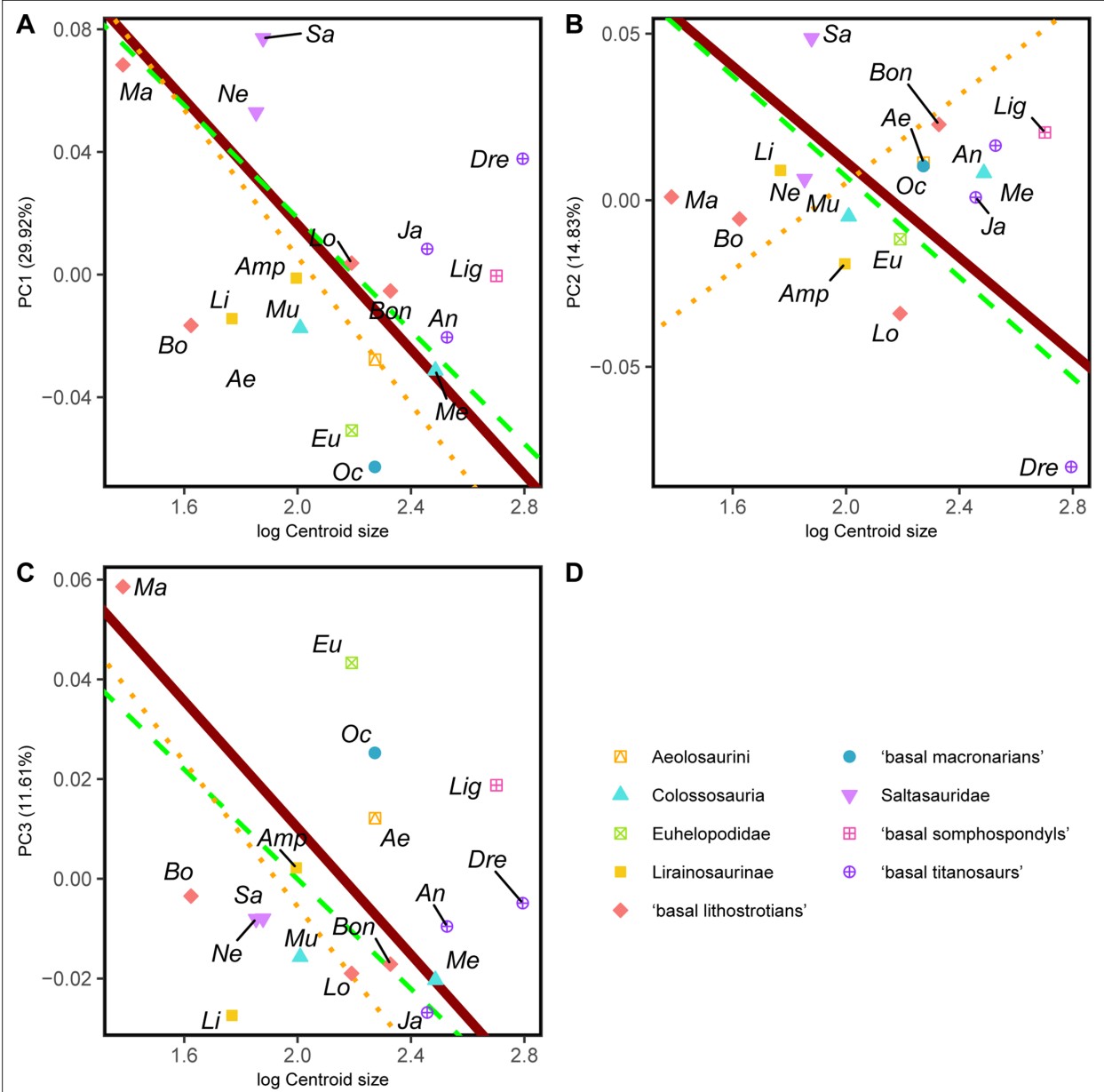

**Figure 6.** RMA results of the first three shape PCs against the logarithm of the hind limb centroid size. (**A**) PC1 against log-Centroid size, all taxa RMA in dark red: intercept = 0.221, slope = –0.102, $r^2$=0.105, p=0.204; Titanosauria only partial RMA in dashed green: intercept = 0.203, slope = –0.092, $r^2$=0.118, p=0.229; Lithostrotia only partial RMA in dotted orange: intercept = 0.246, slope = –0.120, $r^2$=0.319, p=0.07; (**B**) PC2 against log-Centroid size, all taxa RMA in dark red: intercept = 0.155, slope = –0.072, $r^2$=0.054, p=0.371; Titanosauria only partial RMA in dashed green: intercept = 0.158, slope = –0.075, $r^2$=0.117, p=0.232; Lithostrotia only partial RMA in dotted orange: intercept = –0.127, slope = 0.066, $r^2$=0, p=0.952; (**C**) PC3 against log-Centroid size (Csize), all taxa RMA in dark red: intercept = 0.137, slope = –0.064, $r^2$=0.055, p=0.363; Titanosauria only partial RMA in dashed green: intercept = 0.110, slope = –0.055, $r^2$=0.236, p=0.078; Lithostrotia only partial RMA in dotted orange: intercept = 0.140, slope = –0.073, $r^2$=0.313, p=0.074. *Ae – Aeolosaurus, Amp – Ampelosaurus, An – Antarctosaurus, Bo – Bonatitan, Bon – Bonitasaura, Dre – Dreadnoughtus, Eu – Euhelopus, Ja – Jainosaurus, Li – Lirainosaurus, Lig – Ligabuesaurus, Lo – Lohuecotitan, Ma – Magyarosaurus, Me – Mendozasaurus, Mu – Muyelensaurus, Ne – Neuquensaurus, Sa – Saltasaurus.*

Considering the analyzed sample, the acquisition of wide-gauge locomotion would be the main source of hind limb morphological variability in titanosaurian sauropods (*Figures 2 and 5*) and possesses a significant phylogenetic signal. The trend toward a more arched limb posture persists in the more deeply nested titanosaurs. *Oceanotitan dantasi*, a possible representative of Somphospondyli, exhibits a columnar hind limb characterized by no deflection of the femoral head with regard to

**Table 6.** Estimated phylogenetic signal via Pagel's lambda.

p value of log-likelihood ratio test after 1000 simulations. * Indicates significant relationships for an alpha of 0.05. log-Csize – Log-transformed hind limb centroid size.

| | Lambda | p |
|---|---|---|
| log-Csize | 0.982 | 0.003* |
| PC1 | 0.715 | 0.000* |
| PC2 | 0 | 1 |
| PC3 | 0.76 | 0.002* |
| PC4 | 0 | 1 |
| PC5 | 0.778 | 0.031* |
| PC6 | 0.697 | 0.01* |

the tibial condyle (although some titanosauriforms exhibit this feature, see *Royo-Torres, 2009*; recent phylogenetic approaches suggest that *O. dantasi* might represent non-titanosauriform macronarian, see *Mocho et al., 2024a*), and a straight and long, lateromedially narrow zeugopod with few anterior rotations of the fibula (*Figure 2*). In contrast, the more deeply branching titanosauriforms exhibit the typical titanosaurian hind limb configuration with a more arched posture, lateromedially more robust femora, and increased medial or proximo-medial deflection of the femoral head. Distally, the hind limb exhibits a slight rotation of the femoral distal ends and increasingly lateromedially robust zeugopods. The robust zeugopod elements are the only resemblance to *O. dantasi* as our outgroup (e.g. *Figure 2*). The highest positive PC1 (*Figure 6*) values correlate with several hyper-robust taxa of different titanosaur subclades that exhibit lateromedially and anteroposteriorly wide stylopods and zeugopods. In these taxa, the zeugopod bones are extremely shorter proximodistally, extremely arched with a predominance of tibiae characterized by short cnemial crests but somewhat rotated, interlocking with anteriorly deflected and robust fibulae as in the saltasaurine *Saltasaurus loricatus*, the lithostrotian *Magyarosaurus* spp. and the possible non-lithostrotian titanosaur *Dreadnoughtus schrani* (based on the super-tree: *Figures 2 and 5*). The acquisition of this particular morphology is correlated with the development of gigantism within Titanosauria (*Carrano, 2005*; *Lefebvre et al., 2022*; *Ullmann et al., 2017*), at least when analyzing early branching members of Lithostrotia. However, specimens occupying higher PC1 values exhibit the most hyper-robust and arched hind limbs, including representatives of the smallest lithostrotians and the largest non-lithostrotian titanosaur studied (*Figure 5*). The analyses of evolutionary trends presented here reveal that the trend toward titanosauriform gigantism shifted toward adaptation to dwarfism in some lithostrotian titanosaurs like *Magyarosaurus* and *Neuquensaurus* (see discussion on hind limb size variability below). In general, the results obtained here confirm the previously proposed trend toward the acquisition of robust and arched hind limbs in Titanosauria (*Figure 7*). Nevertheless, once hind limb mechanical stability was acquired (following e.g. *Ullmann et al., 2017*) the increasingly arched and robust morphologies established within Titanosauria cannot be fully related to an increase in body mass (*Figure 6a*) and are better explained as convergence between different subclades (*Figures 2 and 6*, see evolutionary trend breakdown in *Table 7*). Saltasaurine lithostrotians are characterized by this type of extreme morphology, with hyper-robust limb bones. Even large saltasaurid sauropods like *Opisthocoelicaudia skarzinski* exhibit this type of hind limb (*Borsuk-Bialynicka, 1977*). However, this morphology is not exclusive of saltasaurids, since other titanosaurs exhibit homoplastic hyper-robust and arched hind limbs (i.e. *Dreadnoughtus* and *Magyarosaurus*). This progressively arched limb was probably hard-coded in the macronarian bauplan and, after the somphospondylan stable posture was acquired, was still present but relatable to a significant variability of biomechanical adaptations (*Voegele et al., 2021*) as our analyses suggest. Despite the biomechanical diversity associated with hind limb morphology being still unclear, several studies point out that morphological differences in the fore limb elongation in sauropods may be related to different feeding niche capabilities (*Bates et al., 2016*; *Upchurch et al., 2021*; *Vidal et al., 2020*; *Voegele et al., 2020*), including discussion on possible bipedal/tripodal rearing abilities that are much more developed, particularly in sauropods with a hyper-robust hind limb (*Upchurch et al., 2021*). Interestingly, *Dreadnoughtus* exhibits this

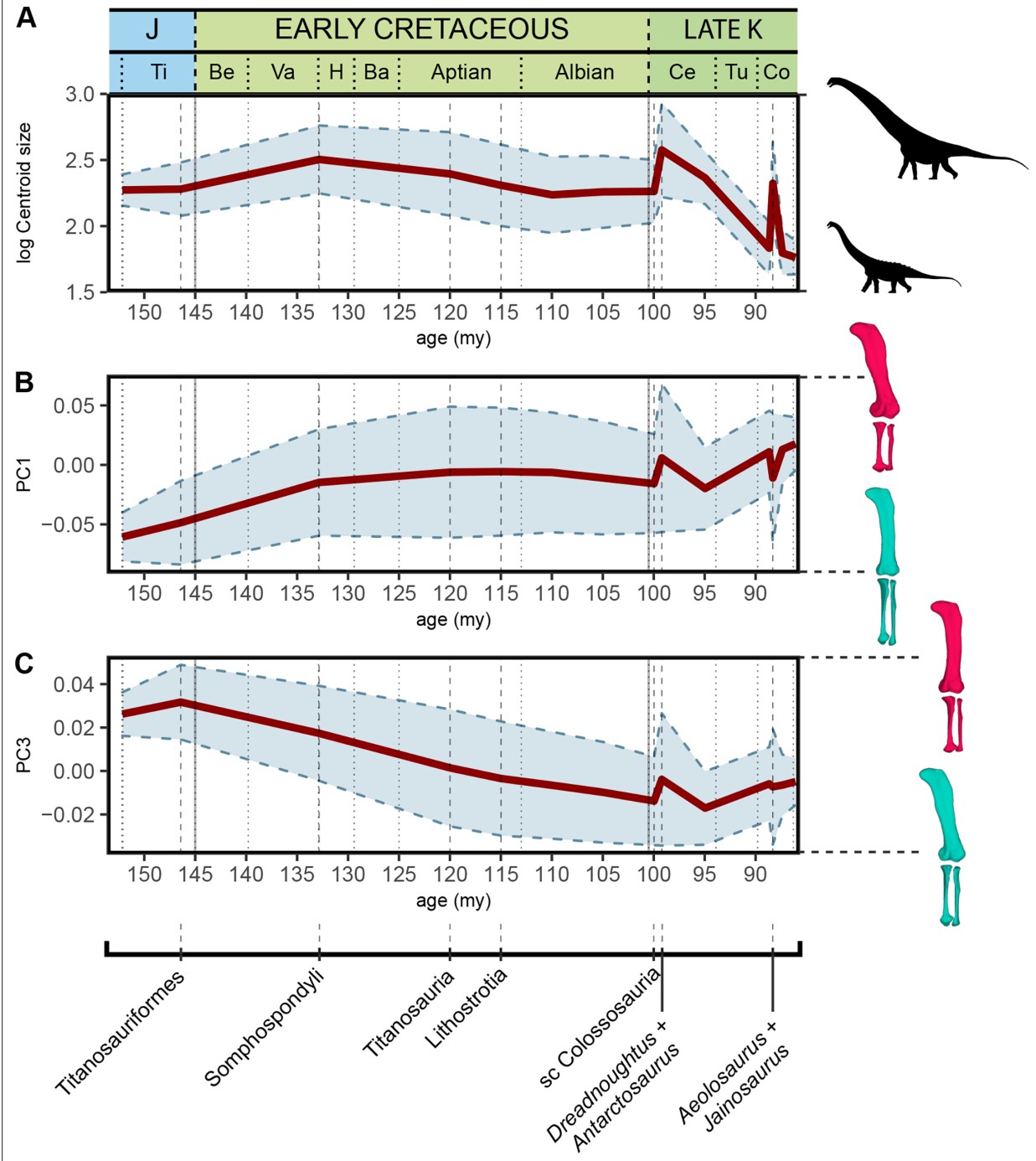

**Figure 7.** Evolution of log-transformed hind limb centroid size, shape PC1, and PC3 according to our sample and time-calibrated supertree topology. Ba – Barremian, Be – Berriasian, Ce – Cenomanian, Co – Conacian, H – Hauterivian, J – Jurassic, Late-K – Late Cretaceous, Ti – Tithonian, Tu – Turonian, Va – Valanginian. my – million years from present, sc – subclade Colossosauria + *Bonitasaura*.

hyper-robust hind limb with subequal autopodial lengths and a wedged sacrum (*Vidal et al., 2020*; *Voegele et al., 2021*), an anteriorly placed body Centre of Masses (CoM; *Bates et al., 2016*) and possibly high browsing feeding capabilities (e.g. *Vidal et al., 2020*). While the small saltasaurids in this study exhibit short necks, stout bodies, and a posteriorly placed CoM (*Bates et al., 2016*), the pneumaticity of some saltasaurid tails (*Upchurch et al., 2021*; *Zurriaguz and Cerda, 2017*) may suggest the possibility of a more anterior location of the CoM. It is possible that the acquisition of a hyper-robust and arched hind limb morphology is related to improved skeletal support in high browsing

**Table 7.** Results of pairwise ancestor-descendant comparisons for log-transformed centroid size in macronarian sauropods in our time-calibrated supertree (17 terminal taxa).

n=ingroup internal nodes + terminal taxa. *=accepted as significant with alpha <0.05. °=Lithostrotia + *Antarctosaurus*.

| Clade | Mean | Sum | Skew | Median | n | Positive changes | Negative changes | $\chi^2$ | p |
|---|---|---|---|---|---|---|---|---|---|
| Titanosauriformes | −0.127 | −3.819 | −0.223 | −0.032 | 30 | 12 | 18 | 1.2 | 0.273 |
| Somphospondyli | −0.366 | −10.252 | −0.148 | −0.256 | 28 | 4 | 24 | 14.286 | 0.000* |
| Titanosauria | −0.288 | −7.488 | −0.088 | −0.182 | 26 | 5 | 21 | 9.846 | 0.002* |
| Lithostrotia° | −0.283 | −6.22 | −0.193 | −0.307 | 22 | 5 | 17 | 6.545 | 0.011* |

and extremely large sauropods like *Dreadnoughtus*, which in combination with their size and neck dorsiflexion capabilities (e.g. *Vidal et al., 2020*) allowed them to feed in a high niche stratification, independent of the acquisition of rearing capabilities.

Although PC2 does not provide a phylogenetic signal for the titanosaurian evolution, it reveals significant differences among smaller titanosaurian taxa with robust hind limbs and especially hyper-robust zeugopodia (*Figure 3*), like *Magyarosaurus* and the saltasaurines *Neuquensaurus* and *Saltasaurus*, which can be related to different roles in the necessary mechanical stability for rearing capabilities (following *Upchurch et al., 2021*). *Magyarosaurus*, *Neuquensaurus,* and *Saltasaurus* occupy increasingly positive PC2 values (*Figure 3*). *Magyarosaurus* exhibits a short zeugopod in which the tibia is slightly laterally rotated, the cnemial crest is laterally projected, and the fibula is slightly sigmoidal with its proximal third displaying an anterior projection and a medially deflected anterolateral crest. Saltasaurines exhibit an extreme condition characterized by short and robust zeugopods. Members of this clade show laterally rotated tibiae with posteriorly deflected distal ends and sigmoidal fibulae that articulate in an oblique position with the anteriorly projected proximal third and anteriorly placed lateral crest. Additionally, this anatomical fibular configuration produces a more anterior displacement of the distal attachment of the *M. iliofibularis* and a probable different distribution of the stress as the main beam of the shafts is rotated from the hind limb trunk, for which previous myological studies suggest that there is no evident body size/phylogenetic pattern (*Voegele et al., 2021*). In contrast, *Dreadnoughtus* exhibits a slightly arched hind limb posture with a robust zeugopod, a medially deflected cnemial crest of the tibia, and a sigmoidal fibula that is slightly posteriorly deflected, and it is projected at negative PC2 (*Figure 3*). In this taxon, the fibula, although sigmoidal, articulates in a mostly straight anatomical position with the particularly robust tibia. The differences among robust hind limb titanosaurian morphologies may suggest that their convergence is due to other sources of variation (i.e. different biomechanical adaptations). It is also important to note that all the lithostrotian titanosaurs that exhibit a slightly rotated fibula, whether sigmoid or not, and extremely arched hind limb morphology independent of their size, exhibit femora with reduced fibular epicondyles (after *Otero et al., 2020*).

Previous studies also pointed to the distinct morphology of the members of Colossosauria, especially in their extremely elongate and gracile zeugopod (*González Riga et al., 2018*; *Páramo, 2020*). Here, we found a distinctive plotting for the rinconsaurian hind limb of *Muyelensaurus* and *Mendozasaurus,* often overlapping with *Aeolosaurus,* as in the PC1 and PC3 morphospaces (*Figures 2–4*). It is especially relevant that these specimens occupy distinctive PC1 and PC3 scores within Rinconsauria/Colossosauria, which exhibit a signal regarding titanosaurian evolution and corresponding to a specific lineage of this titanosaurian subclade. In addition, the phylomorphospaces indicate that some lithostrotians shift away from the main early-branching lithostrotian area of the morphospace towards a straighter, elongated hind limb convergent with non-titanosaurian macronarians (*Figures 2a–b and 4a–b*). In the case of *Aeolosaurus,* the phylomorphospace produces a long, branched, and wide morphospace occupation among titanosaurs (*Figures 2–4*) as the whole titanosaurian clade re-occupies an entirely new area of the morphospace. Representatives of this group exhibit extremely distinct morphologies, with *Aeolosaurus* and *Antarctosaurus* often much more similar to the early branching non-titanosaurian Titanosauriformes than to other deeply-branched titanosaurs (e.g. *Figures 2b and 4b*). Colossosauria includes several of the largest titanosaurs ever known (*Carballido et al., 2017*; *Riga et al., 2019*) as well as some of the smallest lithostrotian titanosaurs in South America that are placed within Rinconsauria, which has been recently recovered as an early branching lineage

of Colossosauria (*Carballido et al., 2017*; *Riga et al., 2019*). Additionally, some recent phylogenetic hypotheses recovered the aeolosaurine lithostrotians as deeply branching rinconsaurians as well (*Carballido et al., 2017*; *Silva Junior et al., 2022*). Whether the appendicular skeleton of Aeolosaurini and Colossosauria exhibits morphological convergence between these two different lithostrotian subclades or they represent the same morphological trend within the same lineage of specialization is still unknown. This depends on whether the latter phylogenetic hypotheses including Aeolosaurini within Colossosauria are accepted as valid. It is important to note that despite the small to medium size of *Muyelensaurus* and *Aeolosaurus*, *Mendozasaurus* can be considered a large sauropod and it exhibits morphological differences from other large sauropods in our sample (i.e. the titanosaurs *Dreadnoughtus* and *Jainosaurus* or the non-titanosaurian somphospondylan *Ligabuesaurus*), similar to previous analyses of separate specimens (*Páramo, 2020*). *Antarctosaurus* seems to be the only large non-colossocaurian titanosaur that shares a similar morphospace with the members of Colossosauria analyzed here (i.e. PC1-PC3, *Figures 2–4*). The inclusion of hind limb elements of other members of this clade (i.e. *Patagotitan*, *Epachthosaurus*) in future analyses might shed light on this morphological pattern, as the zeugopod elements of several of these taxa resemble those of other non-colossosaur titanosaurs (e.g. *Dreadnoughtus* and *Patagotitan*; see also *Argentinosaurus* in Páramo et al. 2020). Also, *Aeolosaurus* exhibits a different morphology from other post-early Coniacian lithostrotians titanosaurs, thus producing a widening of the occupied morphospace (*Figure 7b–c*).

Despite these small differences in the hind limb morphology in key features that may be related to the giant titanosaurian body size, our analyses indicate the great morphological convergence between the different titanosaurian subclades (e.g. *Aeolosaurus* and Colossosauria; or the latter closer to the plesiomorphic morphology of *Euhelopus*, *Figure 2*; see also *Tables 3–4*). Also, no significant differences were found in any of the shape variables, either in the pairwise subclade comparisons (*Supplementary file 2*), or in the shape PCs that indicate slightly less variance than expected by evolution under Brownian motion (i.e. shape PC1, PC3 in this text, *Table 5*; shape PC5-6 in *Supplementary file 2*).

## Hind limb size evolution

Our results with the current time-calibrated supertree topology indicate that there is a trend in the evolution of titanosaurs towards a decrease in size (*Figures 5 and 7*, *Tables 5–6*). In the light of our current sample, once many of the hind limb features relatable to wide-gauge posture are acquired, there is a phylogenetical trend, close to a pure Brownian motion model, toward an overall decreasing size (log-transformed hind limb centroid size $\lambda$ =0.982), which is consistent with previous results that indicated that lithostrotians (or even all macronarian sauropods) may not follow Cope's rule (*Carrano, 2006*; *de Souza and Santucci, 2014*). This could be due to our current lithostrotian sample as some of the Saltasaurinae or closely related taxa (i.e. *Opisthocoelicauda* and *Alamosaurus*, which are usually considered members of a more inclusive clade, Saltasauridae) are also large lithostrotians, especially *Alamosaurus*. However, it is important to note that we also found several key traits in our shape variables (most importantly summarized in PC1 and PC3, 41.53% of the total sample variance between them; *Table 7*) that are usually related to the acquisition of gigantism and that exhibit significant signals about titanosaurian evolution. When we tested for a significant correlation between these traits and the log-transformed hind limb centroid size, no significant correlation was found between any of the shape variables and size (*Table 4*, *Figure 6*).

Within Titanosauria, the hind limb arched morphology (increasing wide-gauge posture) does not correspond to the significant trend toward a size decrease observed in this sauropod clade (see *Figure 7*, *Tables 6–7*). Large titanosaurs show very different hind limb morphologies. Some lithostrotians exhibit plesiomorphic columnar, slightly arched hind limbs with elongated or even gracile zeugopod elements (*Antarctosaurus*, *Aeolosaurus,* or the extremely slender hind limb of the colossosaurians *Mendozasaurus* and *Muyelensaurus*; *Figure 2*), whereas other large titanosaurs (i.e. *Dreadnoughtus*) exhibits an extremely robust hind limb with slightly reduced zeugopods, as may be expected from the typical trend toward the acquisition of an arched morphology in Titanosauriformes (*Figure 2a*). In this context, it is remarkable that most of the lithostrotians that exhibit extremely arched hind limbs, robust elements, and reduced zeugopod elements decrease in size, like *Saltasaurus* or *Magyarosaurus* (*Figure 2*, see discussion on morphological convergence above). Our test found that the trend toward this type of hind limb breaks in titanosaurian sauropods, with more

**Table 8.** Results of ancestor-descendant pairwise comparisons for shape PC1 in macronarian sauropods in our time-calibrated supertree (17 terminal taxa).

n=ingroup internal nodes + terminal taxa. *=accepted as significant with alpha <0.05. °=Lithostrotia + *Antarctosaurus*.

| Clade | Mean | Sum | Skew | Median | n | Positive changes | Negative changes | $\chi^2$ | p |
|---|---|---|---|---|---|---|---|---|---|
| Titanosauriformes | 0.051 | 1.518 | 0.98 | 0.043 | 30 | 29 | 1 | 26.133 | 0.000* |
| Somphospondyli | 0.019 | 0.538 | 1.158 | 0.012 | 28 | 20 | 7 | 6.259 | 0.012* |
| Titanosauria | 0.011 | 0.291 | 1.076 | 0.003 | 26 | 14 | 10 | 0.667 | 0.414 |
| Lithostrotia° | 0.011 | 0.24 | 1.079 | 0.002 | 22 | 11 | 10 | 0.048 | 0.827 |

variable morphologies and large overlapping of morphospaces, whereas the hind limb size decreases (*Figure 7*, *Tables 6–7*). Most importantly, PC1-PC3 exhibits some of the morphological features that are classically related to wide-gauge arching morphology (e.g.) and our analyses found a significant trend toward increasingly arched hind limbs (*Tables 8 and 9*). However, the post-Cenomanian lithostrotians exhibit great morphological variability, with several taxa exhibiting plesiomorphic columnar femora (i.e. *Ampelosaurus*, *Aeolosaurus*, *Magyarosaurus*) and slightly straight zeugopodial elements, with anteriorly expanded tibial proximal ends, and straight non-rotated fibula expanded in the anterior view (*Figures 4 and 7*; see discussion on morphological convergence above). Among these lithostrotians, *Ampelosaurus* and *Aeolosaurus* are medium-sized sauropods, whereas *Magyarosaurus* exhibits both plesiomorphic hind limb morphology and the smallest size. Similarly, the large size variability in the post-early Coniacian cannot be related to the morphological features that are classically associated with the progressively arched morphology (*Figure 7*), as discussed before. Therefore, titanosaurs that produce large PC1-PC3 changes at this peak are both moderate-to-large in size, inducing a displacement of the morphospace: (i) toward the plesiomorphic hind limb morphology, plotted in negative PC1 and positive PC3 values (i.e. *Aeolosaurus*) or (ii) toward a morphology intermediate between the main PC1-PC3 typical robust arched titanosaurian hind limb morphology (i.e. *Jainosaurus*; *Figure 7*).

Here, it is important to point out that when comparing our results with other lithostrotian titanosaurs not included in the current analysis, similar hind limb posture variability not relatable to hind limb size increases (and therefore body size) is observed. The arched morphology with extremely robust zeugopodial elements exhibited in saltasaurid sauropods (e.g. *Saltasaurus*) is also found in the dwarf non-lithostrotian titanosaur *Diamantinasaurus* (*Poropat et al., 2021*; *Poropat et al., 2015*). Its tibia shares similarities with those of *Dreadnoughtus* (anteroposteriorly and lateromedially wide proximal end, extremely short and laterally projected cnemial crest and lateromedially expanded distal end; *Poropat et al., 2015*; *Ullmann and Lacovara, 2016*), whereas the fibula is similar to the slightly straight fibula of *Magyarosaurus*, with a proximally anterior deflection (*Poropat et al., 2015*; APB direct observation on *Magyarosaurus* sp. specimens). Despite slight differences in the proportion of the zeugopodial elements, the hind limb morphology of *Diamantinasaurus* is similar to those of other small taxa. However, *Rapetosaurus* exhibits a completely different lithostrotian hind limb configuration that is more similar to that observed in *Lirainosaurus*. Thus, whereas the fibula is slightly straight with an anteriorly expanded proximal third (*Rogers, 2009*) as in *Magyarosaurus* and *Diamantinasaurus*, both the tibia and fibula are extremely lateromedially compressed as in *Lirainosaurus* or *Muyelensaurus* (APB pers. obs.). Our results are congruent with the observation in other, non-sampled small titanosaurian taxa that exhibit similar morphological variability. Notice that *Rapetosaurus* is based on

**Table 9.** Results of ancestor-descendant pairwise comparisons for shape PC3 in macronarian sauropods in our time-calibrated supertree (17 terminal taxa).

n=ingroup internal nodes +terminal taxa. *=accepted s significant with alpha <0.05. °=Lithostrotia + *Antarctosaurus*.

| Clade | Mean | Sum | Skew | Median | n | Positive changes | Negative changes | $\chi^2$ | p |
|---|---|---|---|---|---|---|---|---|---|
| Titanosauriformes | −0.035 | −1.047 | 1.697 | −0.038 | 30 | 2 | 28 | 22.533 | 0.000* |
| Somphospondyli | −0.023 | −0.639 | 2.174 | −0.024 | 28 | 2 | 26 | 20.571 | 0.000* |
| Titanosauria | −0.008 | −0.21 | 2.483 | −0.009 | 26 | 3 | 23 | 15.385 | 0.000* |
| Lithostrotia° | −0.003 | −0.074 | 2.323 | −0.004 | 22 | 4 | 17 | 8.048 | 0.005* |

juvenile and subadult specimens, and it may retain a plesiomorphic slender morphology that changes in the adult (see precocial development in the limbs of *Rapetosaurus*) (*Curry Rogers et al., 2016*).

Among large titanosaurs, *Elaltitan* exhibits a (virtually restored) robust femur, a slightly robust and plesiomorphic straight tibia but with a posteriorly rotated distal end, and an extremely sigmoidal and anteriorly projected fibula like those of members of Saltasaurinae but slightly lateromedially narrow compared to the latter (*Páramo, 2020*). Despite some differences, other robust large lithostrotians (i.e. *Dreadnoughtus*) exhibit similar hind limb morphology. However, other large lithostrotians, like *Argentinosaurus,* exhibit a plesiomorphic straight fibula (i.e. *Páramo, 2020*). The hind limb of *Patagotitan* exhibits a less arched posture, with a slightly straighter femur, whereas the fibula is robust, sigmoidal, and has an anterior expansion of the proximal third (*Otero et al., 2020*). The stylopodium is not as robust and arched as in *Dreagnoughtus* but also exhibits the slightly plesiomorphic titanosaurian straight hind limb, which is also part of the shift toward robust morphology after the earlier branching Colossosauria, such as *Mendozasaurus* (*Figures 2 and 4*). *Patagotitan* still lacks the extreme medial deflection of the femoral head seen in our PC1 positive and PC3 negative values. *Petrobrasaurus* and *Narambuenatitan* are both large titanosaurs that exhibit similar medial deflection of the femoral head to *Dreadnoughtus*, and in the case of *Narambuenatitan*, even more than *Saltasaurus* and *Neuquensaurus* (*Páramo, 2020*). However, the tibia of *Petrobrasaurus* is extremely lateromedially narrow as in *Mendozasaurus* (*Páramo, 2020*; APB pers. obs.) instead of the typical robust tibiae of the extremely arched hind limb found in our results (see *Figure 2*). Only *Uberabatitan* exhibits a clear morphology like the hind limb morphology found in *Dreadnoughtus* and our positive PC1 scores. Another large titanosaur that exhibits a lateromedially narrow proximal tibial end is *Ruyangosaurus* (*Lu et al., 2009*). Despite being fragmentary, the femur is straight (with a rounded shaft) and has a tibia with lateromedially narrow proximal end similar to that observed in *Ligabuesaurus* and the members of Colossosauria (which are closely positioned in our analyses; *Figures 2 and 4*). Other more deeply branching lithostrotians either exhibit a hind limb that closely resembles the early-branching colossosaurs or a plesiomorphic tibia as in *Abditosaurus* (*Vila et al., 2022*). In this taxon, the femur is anteroposteriorly narrow with a highly eccentric shaft, like other Ibero-Armorican lithostrotians (e.g. *Ampelosaurus*). *Abditosaurus* also preserves an extremely lateromedially narrow tibial proximal end (*Vila et al., 2022*). Only the fibula is sigmoid and anteriorly projected, but lateromedially narrow with an expanded anteromedially crest, which is larger anteroposteriorly than in *Lohuecotitan*, *Lirainosaurus* (*Vila et al., 2022*) and probably like those of *Magyarosaurus*, and is the single character that resembles our PC1 results.

Moreover, we must consider that our age estimations for the nodes of our supertree are extremely conservative and are based on the topology after the reduced tips of our sample (see *Table 1*). Titanosaurian sauropods appeared unambiguously during the Early Cretaceous (*D'emic, 2012*; *Mannion et al., 2019b*; *Poropat et al., 2016*) just as lithostrotian titanosaurs appeared early after the Valanginian-Hauterivian (*Mannion et al., 2019b*; *Poropat et al., 2016*). Considering several recent phylogenetic hypotheses, the colossosaurian node may have branched in the early Albian (*Gorscak et al., 2022*; *Sallam et al., 2018*; *Vila et al., 2022*). Saltasauridae have also often been estimated at the transit between the Early Cretaceous and Late Cretaceous, with *Jiangshanosaurus* considered an early-branching saltasaurid (*Poropat et al., 2016*). However, the recent redescription of *Jiangshanosaurus* material has shed light on its phylogenetic affinities, and it may be an euhelopodid or at least not as a deeply branched titanosaur (*Mannion et al., 2019a*). Saltasaurid lithostrotians are nevertheless traced back to the late Albian in recent studies (*Sallam et al., 2018*; *Vila et al., 2022*). The trends observed in our study may be accentuated with still a general body size decrease among lithostrotian sauropods (*Figure 7a*). However, a series of convergences in the appendicular skeleton: for example, the acquisition of the plesiomorphic columnar titanosaur hind limb among medium to large lithostrotians seems to evolve independently of body size (e.g. *Figure 7b*). Recent phylogenetic hypotheses show uncertain phylogenetic affinities for some of the sauropods studied, particularly for some of the lithostrotian taxa. This is the case of the opisthocoelicaudiine affinities that have been suggested for some members of Lirainosaurinae and *Lohuecotitan*, which is also proposed as a subclade within Saltasauridae (*Vila et al., 2022*) or Saltasauroidea (*Mocho et al., 2024b*) as Opisthocoelicaudiinae. *Atsinganosaurus,* an Ibero-Armorican lithostrotian, was recently recovered as a member of Lirainosaurinae (; *Díez Díaz et al., 2018*, *Mocho et al., 2024b*) or as a lognkosaurian colossosaur (*Gorscak and O'Connor, 2019*; *Vila et al., 2022*). *Atsinganosaurus* is a small lithostrotian with a hind

limb morphology similar to that of *Lirainosaurus* (*Díez Díaz et al., 2018*). The extremely gracile limbs of *Atsinganosaurus* resemble those of the small rinconsaurian lithostrotians (*Riga et al., 2019*; *Pérez Moreno et al., 2023* this study) or even the large *Mendozasaurus* (*González Riga et al., 2018*). However, its affinities to Colossosauria will still indicate a convergence between small Opisthocoelicaudiinae and lognkosaurian colossosaurs according to the phylogenetic hypothesis of *Vila et al., 2022*. This phylogenetic hypothesis still indicates that the large-sized titanosaurs like *Dreadnoughtus* and *Alamosaurus* (robust medially bevelled femur and sigmoid fibula; see *Lehman and Coulson, 2002*; *Wick and Lehman, 2014*) exhibit a morphologically convergent hind limb with the small saltasaurines like *Neuquensaurus* and *Saltasaurus*.

It seems that these morphological similarities are due to other biomechanical aspects after the acquisition of the arched hind limb within Somphospondyli, as well as to other morphological features related to the wide-gauge posture of the appendicular skeleton. It is possible that minor differences that do not show a significant phylogenetic signal and recover in other shape-PCs (*Supplementary file 2*) are key features for other adaptations that are also important such as the trade-off between speed and rearing stability, which may have shaped limb morphology of Titanosauriformes, particularly in the zeugopods (e.g. *Upchurch et al., 2021*).

## Caveats of this study

Our analyses include a wide range of titanosaurs from most of the proposed subclades. However, many hind limb elements are fragmentary and required virtual restoration according to *Páramo et al., 2020*. Traditional studies propose excluding incomplete specimens from the sample. However, in this case, several of our taxa from an already small sample (n=17) could not be included in this study because they do not have complete specimens to calculate the mean shape for each hind limb element type (e.g. *Mendozasaurus* with several tibia and fibula specimens, *Oceanotitan* with only the fragmentary elements of its left hind limb). The exclusion of potentially informative areas or taxa may hinder paleobiological studies (*Brown et al., 2012*), and landmarks estimation may be a more informative procedure (*Arbour and Brown, 2014*; *Brown et al., 2012*). Also, it may be interesting to include several taxa that have been examined in previous studies (e.g. *Páramo, 2020*), but many of these taxa lack one or more hind limb element types. In this case, we did not choose to estimate the entire morphology of an element type, because we lack the necessary and more powerful tools to do so, such as partial least squares estimation methods (e.g. *Torres-Tamayo et al., 2020*). The virtual restoration does not appear to contain large artifacts due to deformation of the specimens, and most of the sample is not affected by extreme taphonomic artifacts. Some complete specimens may affect the results nonetheless, as it is common that some shafts might be more eccentric due to crushing (i.e. *Ampelosaurus atacis* could be even closer to *Lirainosaurus astibiae* in PC1) or the deformation of distal condyles in *Dreadnoughtus schrani* which affects its extreme position in PC2. Despite this, we assessed potential biases following *Lefebvre et al., 2020*; Appendix 1-2, and they do not affect significantly any of the shape variables which exhibit phylogenetic signal (e.g., PC2; *Table 6*; see in detail Appendix 2 and *Supplementary file 2*).

Our reconstructions of the analyzed titanosaurian hind limbs can also bias our study. The lack of the astragalus in most of the titanosaurs studied (whether due to the lack of available 3D-scanned specimens or the lack of a preserved astragalus) hinders the estimation of the position of the zeugopodial elements. This study uses the most conservative assumption in the overall position of the distal part of the hind limb as the fibula could be positioned even more distally, without reaching proximally the femur, and interlocked with the tibia in several specimens (e.g. *Neuquensaurus australis*, specimen MCS-5-25/26, APB. pers. obs.). This is especially relevant for the robust and arched hind limb with extremely robust zeugopodia, as *Dreadnoughtus schrani*, *Neuquensaurus* spp., and *Saltasaurus loricatus*, but see also *Uberabatitan riberoi* (*Salgado and De Souza Carvalho, 2008*).

The sample is also small, and we did not opt to include several advanced statistical hypothesis tests like phylogenetic convergence (*Stayton, 2015*) because it did not meet the requirements. Instead, we chose a conservative set of tests to discuss the true morphological convergence over titanosaurian hind limb evolution with a mix of phylogenetic and non-phylogenetic methods. We chose (*Butler and Goswami, 2008*) change frequencies analysis because, although its statistical properties are less well known than those of independent contrast analyses, it is less sensitive to topological imprecision and somewhat independent of branch length differences (*Butler and Goswami, 2008*; *de Souza and*

*Santucci, 2014*). Following this reasoning, we also decided not to estimate independent phylogenetic contrasts that may be sensitive to large differences in branch length in our lithostrotian-biased sample to test for trait correlation between shape-PCs against log-transformed centroid size, contrary to previous studies (*Bates et al., 2016*). Instead, we opted for the traditional use of test for correlation between tree tips (specimens) shape-PCs against log-transformed centroid sizes via RMA and without incorporating the phylogenetic tree topology of our current time-calibrated supertree.

## Conclusions

Our results suggest that the main features related to the acquisition of an arched hind limb posture (presence of lateromedially wide femora with robust zeugopods) are typical of more deeply nested titanosaurs such as saltasaurines (*Carrano, 2005*; *Lefebvre et al., 2022*; *Sander et al., 2011a*; *Ullmann et al., 2017*; *Vila et al., 2022*; *Wilson and Carrano, 1999*) and exhibit a significant phylogenetic signal about titanosaurian evolution (*Table 5*, *Figure 7*). The arched morphology usually related to wide-gauge posture is an exaptation initially related to increasing body size. However, once fully acquired within Somphospondyli, these features are no longer related to increasing body size, as increases in the arched posture and the development of hyper-robust zeugopods are features that evolved independently in several lineages and are shared by several of both the smallest and largest taxa in different titanosaurian lineages (*Figure 6*, *Table 4*). Also, there is an evolutionary trend toward decreasing titanosaurian hind limb size (and body size; *Figure 7*), based on the size of the hind limb centroid. This trend is congruent with previous studies focused on the size evolution within Titanosauriformes (*de Souza and Santucci, 2014*), despite an increase in arched morphology and robustness of the hind limb in deeply nested titanosaurian subclades.

The lack of correlation between the arched posture, position, and robustness of zeugopod, among other traits with the body size, may also be related to a wide morphological variation within Lithostrotia (*Figures 2–4*). It can be noted that, together with a trend toward increasingly arched hind limbs and reduced zeugopods with increasingly rotated fibulae, there is also a large morphological convergence between the different titanosaurian subclades, as indicated by both our non-phylogenetic and phylogenetic analyses (*Tables 2–3*). Many of the morphological changes without a significant signal in titanosaurian evolution are related to the morphology and anatomical position of zeugopod bones. Large morphological convergences unrelated to body size may be a response to different morphofunctional adaptations. Our results show several differences in the morphology of zeugopod elements without a signal in the evolution of titanosaurs, especially those regarding the fibula (rotation, sigmoidal morphology, anterior or posterior displacement of the proximal end that affects the anteroposterior position of the lateral bulge). These morphological differences are also related to its relative position and articulation with the tibia and correlate with the expression of femoral features like the relative development of the posterior epicondyle of the fibular condyle. These features translate into changes in the length and morphology of the hind limb distal musculature that exhibit no clear evolutionary trend within titanosaurs, as previous studies have indicated. Also, our results show that specimens in different subclades share similar morphologies across the shape-PCs that exhibit no phylogenetic signal in titanosaurian evolution. The observed changes in the zeugopod morphology may be related to different morphofunctional and ecomorphological adaptations, and the convergences in hind limb morphology of titanosaurs may explain biomechanical similarities. This may explain the differences observed between small and medium-to-large titanosaurs with either arched or columnar hind limbs (i.e. PC1 similarities) across those shape-PCs that lack a significant signal in the evolution of titanosaurs. However, to test the hypothesis of differences in biomechanical adaptation (i.e. movement speed or differences in feeding niche specialization), further analyses with additional parts of the skeleton must be included.

## Methods

### 3D geometric morphometrics

To analyze the morphology, 17 macronarian sauropod hind limbs (*Table 1*) were 3D-digitized and analyzed using 3D Geometric Morphometrics tool-kit (3D-GMM; *Gunz et al., 2005*; *Páramo, 2020*). The 3D digitizing process was based on the methodology proposed in previous analyses (i.e. *Mallison,*

*2011*; *Páramo, 2020*) and 3D-GMM analyses were conducted in R statistical software v4.1.3 (*R Development Core Team, 2022*). Code for the analyses can be accessed in *Source code 1*.

To reconstruct each taxon hind limb, we first virtually restored the 3D reconstructions of the digitized femora, tibiae, and fibulae and calculated the grand mean shape for each sampled taxon using previous datasets (*Páramo, 2020*). The grand mean specimens for each taxon were mounted on an estimated anatomical position (accounting for the lack of astragalus in our sample to guide the distal zeugopod articulation). A comprehensive description of the methodology can be accessed in Appendix 1.

Each element of the hind limb is separated by an articular cap much larger than in other known extant archosaurs (*Bonnan et al., 2013*; *Holliday et al., 2010*; *Schwarz et al., 2007*; *Voegele et al., 2022*). The femur, and the hind limb overall, may exhibit the best correlation between cartilage cap and bone morphology constrained by its role as support under most of the stress of the body mass (*Bonnan et al., 2013*; *Voegele et al., 2022*). Although cartilaginous cap thickness is not constant and may vary among taxa, we set up our model hind limbs with a similar constant space between the femur and the zeugopod bones in all the sampled taxa. The anatomical mounts include an additional space of 2% of the element length between stylopod and zeugopod (following *Voegele et al., 2022*).

A total of 28 landmarks and 12 semilandmark curves were placed on the hind limb bones partly based on previous studies (*Lefebvre et al., 2022*; *Páramo et al., 2020*; *Páramo, 2020*; *Table 2*, *Figure 2*, *Supplementary file 1*) using IDAV Landmark Editor software (*Wiley et al., 2005*; dataset with the landmark and semilandmark curves coordinates can be accessed in *Supplementary file 1*). Semilandmarks were then slid in R using package 'Morpho' *Schlager, 2017* following *Gunz et al., 2005*. To remove size differences, spatial position, and orientation, the resulting landmark and semilandmark configurations were superimposed via Generalized Procrustes Analysis (GPA) using the 'procSym' function in *Morpho* package. Morphological variance was analyzed with PCA (results in *Table 3*), saving the expected number of PCs which summarize a significant amount of variance after an Anderson Chi's test (see *Bonnan, 2007*).

Evolutionary trend analyses were accomplished after the estimation of a consensus tree topology using the MRP-supertree methodology (*Bininda-Emonds, 2004*) with the *phangorn* package (*Schliep et al., 2017*; *Schliep, 2011*). For supertree construction, we compiled several of the more recent phylogenetic hypotheses that include all the available sampled data (resulting supertree can be accessed in Appendix 1; trimmed supertree in *Figure 2*). The resulting phylogenetic relationships were projected onto the shape-PCA to visualize the phylomorphospace. To analyze true morphological convergences between titanosaurian sub-clades, we tested for morphological differences in the hind limb skeleton using the shape variables (PCs) without the phylogenetic relationships involved using: (i) Mann Whitney U's test and Kruskal-Wallis non-parametric tests accounting for the uneven distribution of the group (sub-clade) samples and (ii) phylogenetic ANOVA using the time-calibrated supertree topology with the *phytools* R package (*Revell, 2012*).

## Hind limb size distribution and phylogenetic signal

Sauropod body mass was proxied by hind limb centroid size collected from the GPA. Body mass can be calculated preferably using both humeral and femoral measurements (*Mazzetta et al., 2004*) or a whole-body volumetric estimation (*Bates et al., 2016*). However, as the hind limb is the main sauropod body mass support, it can be better used as a 'conservative-minimal' approach to its body mass, with larger hind limb corresponding to giant titanosaurian taxa. We tested for allometric relationships between the sauropod hind limb shape variables (PCs) and the centroid size as proxy to body mass via Reduced Major Axis (RMA) regression using *lmodel2* R package (*Legendre, 2018*); but an alternative set of analyses was carried out using femoral length and body mass estimations (see *Supplementary file 2* and Appendix 2).

We used the time-calibrated supertree topology and our shape variables (PCs) and hind limb centroid size to generate the Pagel's lambda ($\lambda$) with the 'phytools' R package and test for the phylogenetic signal of these traits. All hypotheses of statistical correlations, dissimilarity tests, and phylogenetic signal tests were accepted as significant using an alpha level of 0.05 (a comprehensive report of the results can be accessed in *Supplementary file 2*, and a copy of the R code and packages used can be accessed in *Source code 1*). Once those PCs that exhibit significant phylogenetic signal were identified, as well as the log-transformed centroid size, we estimated their ancestral characters (ACEs)

using maximum-likelihood and a simple Brownian evolutionary model similar to the one assumed for estimation of Pagel's lambda. We used the ACEs to observe trends in the evolution of titanosaurian hind limb size and morphology (based on our shape-PCs). We evaluated the differences in body size (proxied by log-transformed hind limb centroid size ACEs) between terminal taxa and internal nodes and between internal nodes of Titanosauriformes, Somphospondyli, Titanosauria, and Lithostrotia, as well as their subclades. The sum of changes, mean change, median change, positive, negative, and the total amount of changes were evaluated for each of the above clades following *Butler and Goswami, 2008*. We used a $\chi 2$ goodness-of-fit test to evaluate whether body size is increasing or decreasing, and shape-PCs occur at the same frequency (50–50% null hypothesis) or have a positive or negative tendency over titanosaurian evolution.

## Acknowledgements

This work is funded by Ministry of Science and Innovation of Spain project PID2019-111488RB-I00 and PID2023-148083NB-I00, and the Junta de Comunidades de Castilla-La Mancha projects (SBPLY/19/180801/000044 and SBPLY/21/180801/000045). Access to fossil vertebrate collections in Argentina was made possible by Spanish Ministry of Economy and Competitiveness grant EEBB-I-16–11875. The authors would like to thank BP. Kear and Uppsala University Natural History Museum staff for facilitating access to their collections. Access to Uppsala University Museum of Natural History paleovertebrate collections was possible thanks to the ERG-2020 EAVP research grant. Author PM was funded by FCT/MCTES for one CEECIND/00726/2017 /CP1387/CT0034 individual contract (https://doi.org/10.54499/CEECIND/00726/2017/CP1387/CT0034) and SFRH/BD/68450/2010 PhD scholarship. The authors are very grateful to S Chapman and Natural History Museum of London (UK) staff for facilitating access to their collections. Access to NHM (UK) was made possible thanks to the European Union Council Synthesys program GB-TAF-6153. They also want to thank the students of the Faculty of Arts in the UCM (Madrid, Spain) and preparation staff for providing the sample for this study. They equally thank many others for access to specimens under their care, including S Langreo at the Museo de Paleontología de Castilla-La Mancha (Cuenca, Spain); JA Ramírez de la Peciña and C Corral at the MCNA (Vitoria, Spain); R Coria at the MCF (Plaza Huincul, Argentina); L Filippi at the MRS (Rincón de los Sauces, Argentina); M Reguero at the MLP (La Plata, Argentina); A Otero at CONICET (La Plata, Argentina); S Devincenzi at the IANIGLA (Mendoza, Argentina); and B González Riga at the UNCUYO (Mendoza, Argentina); C Muñoz at the MPCA (Cipolletti, Argentina); I Cerda fromat the CONICET (Cipolletti, Argentina); P Ortiz at the PVL (Tucumán, Argentina); AG Kramarz and M Ezcurra at the MACN (Buenos Aires, Argentina), J Le Loeuff at the Musée des Dinosauries d'Espéraza (Espéraza, France) and B Silva at the SHN (Torres Vedras, Portugal). We want to ask M Christopher for access to *D. schrani* skeleton 3D scan. We want to thank PM Sander, the editorial team and two anonymous reviewers for their comments which helped improve this manuscript.

## Additional information

### Funding

| Funder | Grant reference number | Author |
|---|---|---|
| Ministerio de Ciencia e Innovación | PID2019-111488RB-I00 | Adrián Páramo<br>Pedro Mocho<br>Fernando Escaso<br>Francisco Ortega |
| Ministerio de Ciencia e Innovación | PID2023-148083NB-I00 | Adrián Páramo<br>Pedro Mocho<br>Fernando Escaso<br>Francisco Ortega |
| Junta de Comunidades de Castilla-La Mancha | SBPLY/19/180801/000044 | Adrián Páramo<br>Pedro Mocho<br>Fernando Escaso<br>Francisco Ortega |

| Funder | Grant reference number | Author |
|---|---|---|
| Junta de Comunidades de Castilla-La Mancha | SBPLY/21/180801/000045 | Adrián Páramo<br>Pedro Mocho<br>Fernando Escaso<br>Francisco Ortega |
| Ministerio de Economía y Competitividad | EEBB-I-16-11875 | Adrián Páramo |
| Fundação para a Ciência e a Tecnologia | 10.54499/CEECIND/00726/2017/CP1387/CT0034 | Pedro Mocho |

The funders had no role in study design, data collection and interpretation, or the decision to submit the work for publication.

### Author contributions

Adrián Páramo, Conceptualization, Data curation, Software, Funding acquisition, Investigation, Visualization, Methodology, Writing – original draft, Writing – review and editing; Pedro Mocho, Fernando Escaso, Formal analysis, Validation, Investigation, Writing – review and editing; Francisco Ortega, Formal analysis, Supervision, Validation, Investigation, Writing – review and editing

### Author ORCIDs
Adrián Páramo (iD) https://orcid.org/0000-0002-7746-5493
Francisco Ortega (iD) https://orcid.org/0000-0002-7431-354X

Joint Public Review: https://doi.org/10.7554/eLife.92498.3.sa1
Author response https://doi.org/10.7554/eLife.92498.3.sa2

## Additional files

### Supplementary files
MDAR checklist

Supplementary file 1. Sauropod sample dataset and landmarks.

Supplementary file 2. Study test results.

Source code 1. Source code for the geometric morphometric and evolutionary analyses.

### Data availability
All data and supplementary materials, including code for the analyses in R, are available on Zenodo.

The following dataset was generated:

| Author(s) | Year | Dataset title | Dataset URL | Database and Identifier |
|---|---|---|---|---|
| Páramo A | 2024 | Data from: Evolution of hind limb morphology of Titanosauriformes (Dinosauria, Sauropoda) analyzed via 3D Geometric Morphometrics reveals wide-gauge posture as an exaptation for gigantism | https://doi.org/10.5281/zenodo.16272404 | Zenodo, 10.5281/zenodo.16272404 |

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

# Appendix 1

## Methodology

In this appendix, we will describe comprehensively the methodology used in this study for reconstructing the 3D digital morphology of the hind limb for each sampled taxon, the Virtual Restoration of the incomplete hind limbs following our previous workflow (*Páramo, 2020*). The virtual restoration and the shape analyses were all carried out via landmark-based 3D Geometric Morphometrics (3D-GMM; e.g., *Gunz et al., 2005*; *Rohlf, 1999*). The shape analyses were complemented by a series of tests using a phylogenetic ingroup label ('clade' as per most-inclusive Titanosauriformes subclades). The evolutionary analyses were carried out using the shape variables obtained from the 3D-GMM analyses and a summary of current phylogenetic hypotheses via supertree methodology (*Bininda-Emonds, 2004*). Using the summary phylogenetic supertree and the shape variables, we estimated the ancestral characters (ACE; *Pagel, 1999*; *Pagel et al., 2004*), the phylomorphospaces and tested for changes in the hind limb shape across the summary phylogenetic supertree.

We selected a random sample of 16 titanosauriform sauropods and *Oceanotitan dantasi* as representative of non-titanosauriform Macronaria as an outgroup for our study and base for all the evolutionary comparison (17 macronarian sauropod hind limbs in total). The study is slightly biased toward Lithostrotian sauropods nonetheless (see Main Text, potential biases in our results) but we took precautions and conservative conclusions based on our current sample of macronarian sauropods.

## 3D digitizing and reconstruction

### Specimen 3D digitizing

For each sampled Titanosauriformes taxon of this study, all the available specimens of each bone element were digitized individually. The 3D digitizing of each individual specimen was carried by stereophotogrammetry following previous workflows (*Díez Díaz et al., 2021b*; *Mallison, 2010*; *Páramo, 2020*; see also *Figure 1*). The sequences of pictures had been taken with a Canon EOS 1100D DSLR and a Canon EOS 80D DSLR and Canon 18–55 mm f3.5, Canon 50mm f1.8 and Sigma 17–50 mm f2.8 lenses. All the lenses used except for Canon 50mm f1.8 produce a noticeable distortion (e.g. *Collins and Gazley, 2017*) so the pictures are in undistorted RAW format with the specimen centered far from the margins where distortion is concentrated, and the variable focus lenses were set to 35 mm, as in this focal length most of the lens distortion is reduced. The pictures were processed with Agisoft Metashape v.1.8.1 and the resulting 3D model reconstruction of each fossil specimen exported as OBJ for retopology under Instant Meshes (*Jakob et al., 2015*) and mesh correction in Blender v.2.79b (*Blender Online Community, 2018*).

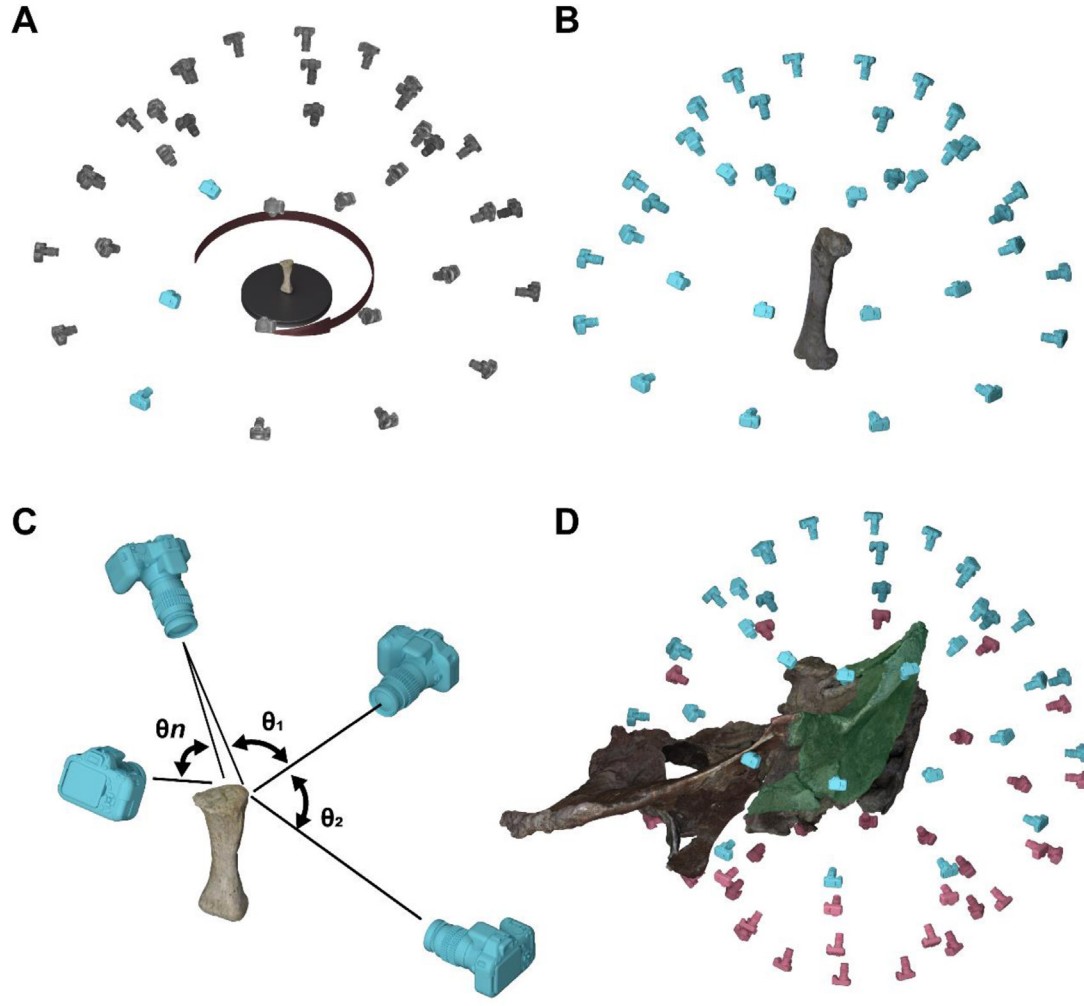

**Appendix 1—figure 1.** Stereophotogrammetry protocol. (**A**) Turntable method of photographing the specimen for small specimens. (**B**) Turn-around method for larger specimens. (**C**) Angles between different pictures to recreate the pixel point-cloud in 3D space in the software. (**D**) Partial digitizing of separated fragments of the same specimen.

## Hind limb reconstruction

Each element type was sampled with the same set of landmarks and semilandmark curves defined (*Páramo et al., 2020*; *Páramo et al., 2022*; *Figures 2 and 3*). The landmark and semilandmark curves were sampled in IDAV Landmark Editor v.3.66 (*Wiley et al., 2005*) using the Atlas template method (*Botton-Divet et al., 2015*) and a custom hypothetical macronarian hind limb 3D model used for defining the landmarks. The landmark and semilandmark curves were imported into R statistical software v4.1.3 (*R Development Core Team, 2022*) with the packages *geomorph* (*Adams et al., 2019*) and *Morpho* (*Schlager, 2017*).

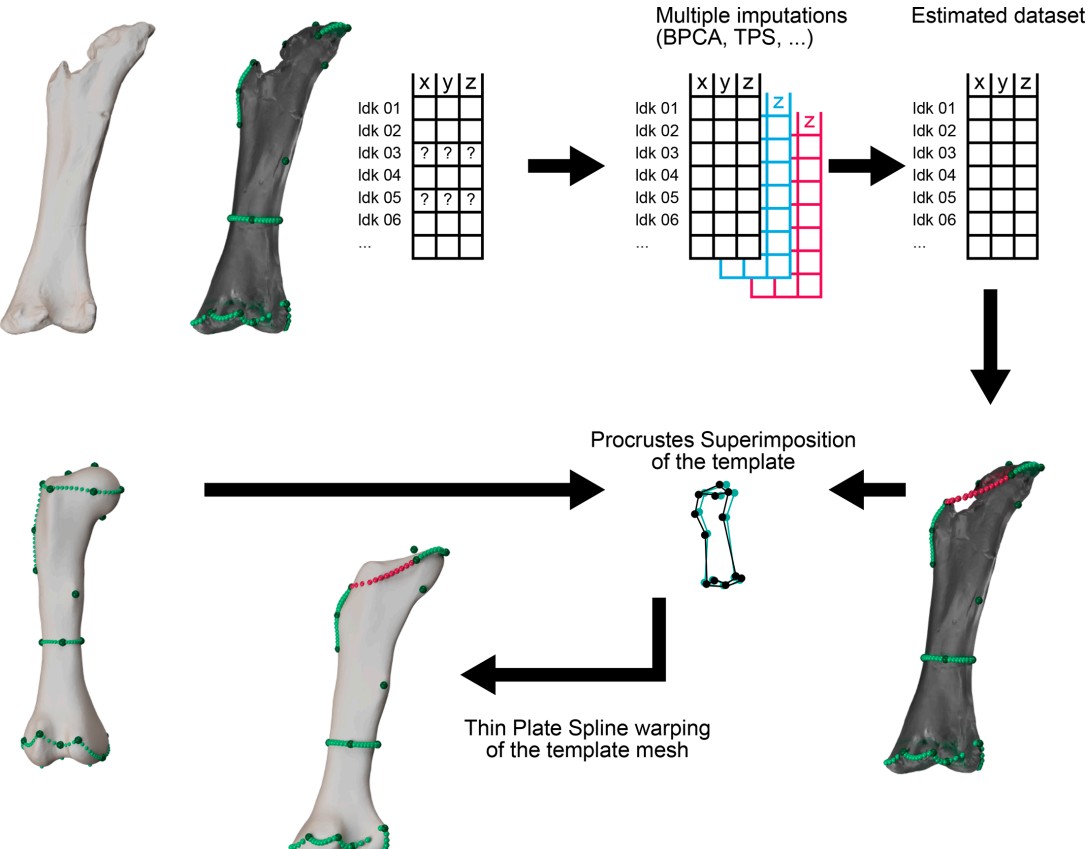

**Appendix 1—figure 2.** Separate specimen Statistical Virtual Restoration workflow as in *Páramo et al., 2020*.
(**A**) Initial 3D specimen reconstruction landmark sampling. (**B**) Multiple imputation methods (TPS in this case).
(**C**) Procrustes superimposition of the template mesh and the estimated landmark configuration in order to obtain
a reconstructed specimen 3D mesh for anatomical mount.

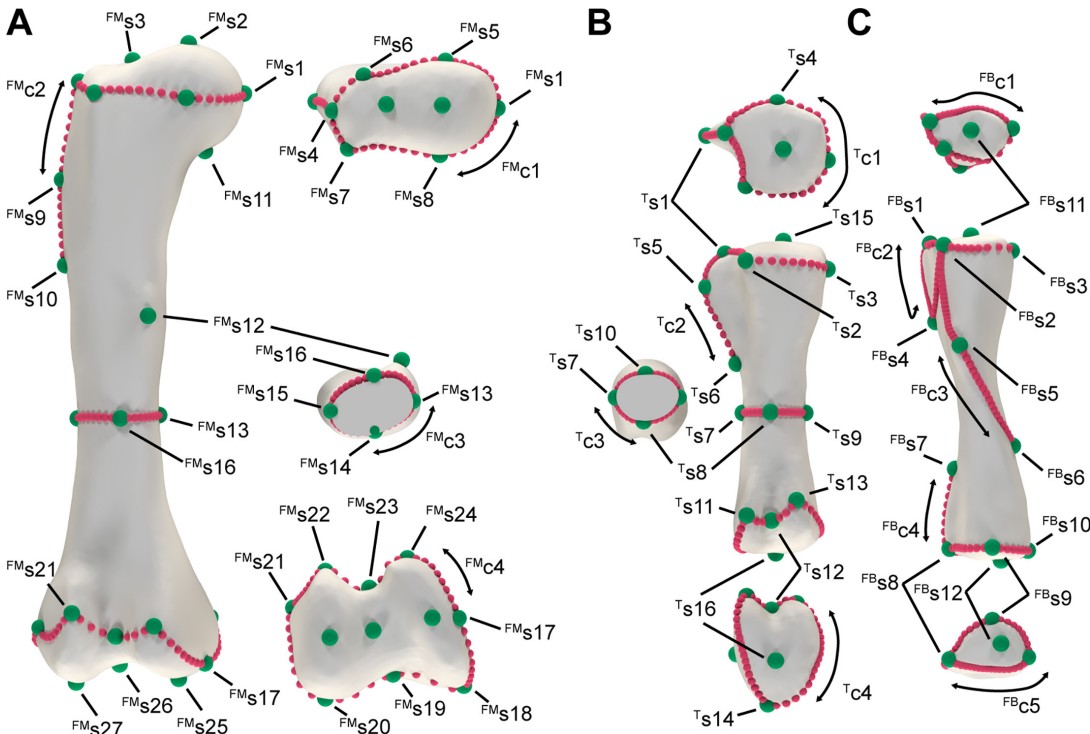

**Appendix 1—figure 3.** Landmarks and semilandmark curves used for the separated specimen prior landmark estimation during Statistical Virtual Restoration procedure. (**A**) Femur. (**B**) Tibia (**C**). Fibula.

Not all the available specimens are complete, and we opted for estimation of missing information instead of excluding them from the study, as the available sample would not be possible to analyze with a rational conclusion (see considerations prior to missing data estimation instead of specimen exclusion in *Brown et al., 2012*). Therefore, in order to obtain a 'mean shape' for each taxon, we firstly need a Virtual Restoration procedure using the same techniques as per the 3D-GMM. We opted for a statistical virtual restoration procedure instead of manually sculpting or undistorting a specimen in a 3D modeling software in order to not increase the errors and biases derived from the author manipulation of the starting data prior to the analyses. The single exception was made with *Oceanotitan dantasi* femur (fragmentary, lacking part of the shaft, total length unknown; *Mocho et al., 2019b*). We first estimated the complete length based on several femoral measurements (e.g. lateromedial width of the proximal end of the femur, the lateromedial width of each distal condyle, the proximodistal length of the lateral bulge, etc.) taken in the entire sample of macronarian sauropod specimens as well as some Ibero-Armorican morphotypes included in previous analyses (supplied in ). With this linear morphometric dataset, we estimated the missing measurements using multiple imputation methods and Predictive Mean Matching (pmm; *Allison, 2000*; *Morris et al., 2014*; *Royston, 2004*) with *mice* package (*Buuren and Groothuis-Oudshoorn, 2011*). Using the estimated length, we placed the preserved 3D-digitized reconstruction of the proximal and distal ends of the femur of *Oceanotitan dantasi* in their corresponding positions in Blender. The resulting femur, lacking the shaft but with the proximal and distal end in their estimated position, was exported as OBJ after retopologizing and mesh reparation similar to other specimens. The lacking landmark and semilandmark curves, especially concentrated in the missing shaft, were estimated like the other specimens of the sample within R; no manual resculpt of the missing shaft was carried.

The landmark estimation was carried out with the same Thin Plate Spline used for semilandmark sliding (TPS; *Brown et al., 2012*; *Gunz et al., 2009*; *Gunz et al., 2005*) using *LOST* package (*Arbour and Brown, 2017*) and *geomorph* (*Figure 2*). We obtained a 'mean shape' of each element type for each taxon of our sample using the estimated complete set of landmarks and semilandmark curves. We also reconstructed the complete 3D morphology for each element type and sampled taxon warping the template hypothetical 3D mesh to the landmark configuration of each macronarian sauropod. The resulting 3D models were used to reconstruct the macronarian hind limbs, mounting

in Blender in their anatomical position. The femur is slightly bevelled to achieve a neutral anatomical position where the distal end offers a somewhat straight plane surface for articulation with the tibia and fibula. The tibia is straight but slightly rotated in proximal view to accommodate the fibular anterior trochanter in the proximal end with the space between the cnemial crest and the shaft (or the cnemial crest and the secondary cnemial crest were present). Also, in order to accommodate the fibula in articulation with the tibia, some fibulae are slightly tilted in lateral view in order to position the anterior end within the cnemial crest but also contact the anterolateral crest and distal end of the fibula within the ascending processes of the tibia distal end (e.g. *Lirainosaurus astibiae*; *Díez Díaz et al., 2013*; *Sanz et al., 1999*).

Also, each element of the hind limb is separated by an articular cap much larger than in other known extant archosaurs (*Bonnan et al., 2013*; *Holliday et al., 2010*; *Schwarz et al., 2007*; *Voegele et al., 2022*). The femur, and the hind limb overall, may exhibit the best correlation between cartilage cap and bone morphology constrained by its role as support under most of the stress of the body mass (*Bonnan et al., 2013*; *Voegele et al., 2022*). Despite cartilaginous cap thickness not being constant and may vary among taxa, we set up our model hind limbs with a similar constant space between the femur and the zeugopod bones in all the sampled taxa in order to have a standardized initial point for comparisons and that there is no way to estimate the volume of the articular cap in the sampled specimens. The anatomical mounts include an additional space of 2% of the element length between stylopod and zeugopod articular surfaces in anatomical position (following *Voegele et al., 2022*).

## Landmark-based 3D-GMM
### 3D landmark sampling
The complete taxon hind limbs were used as a basis for a new sample of landmarks and semilandmarks, considering the new position of the different bone elements and the need of less landmark information for summarizing morphological changes in this study. Many of the sampled landmarks and semilandmark curves resemble the ones used in the analyses of the hind limb elements separated (*Páramo, 2020*; *Páramo et al., 2020*; *Páramo et al., 2022*) or used during the Statistical Virtual Restoration phase previously described. However, the different focus of this study allows us to reduce the number of landmarks of our previous studies (e.g. no curve defined along the femoral or tibial midshaft, no landmarks in the proximal or distalmost surfaces of the femur or zeugopodial elements, etc.; see *Appendix 1—figure 4*). A total of 28 landmarks and 12 semilandmark curves were placed on the hind limb bones partly based on previous studies (*Lefebvre et al., 2022*; *Páramo, 2020*; see Main Text *Table 2* and *Figure 2*; data available in *Supplementary file 1*) using the same IDAV Landmark Editor software (*Wiley et al., 2005*). The semilandmarks were then slid in R using package *Morpho Schlager, 2017* following *Gunz et al., 2005*. To remove size differences, spatial position, and the differences of the 3D mesh orientation, the resulting landmark and semilandmark configurations were superimposed via Generalized Procrustes Analysis (GPA) using the 'procSym' function in *Morpho* package. Morphological variance was analyzed with Principal Component Analysis (PCA; results in Main Text *Table 3*) saving the expected number of Principal Components (PCs) which summarize a significant amount of variance after an Anderson Chi's test (see *Bonnan, 2007*).

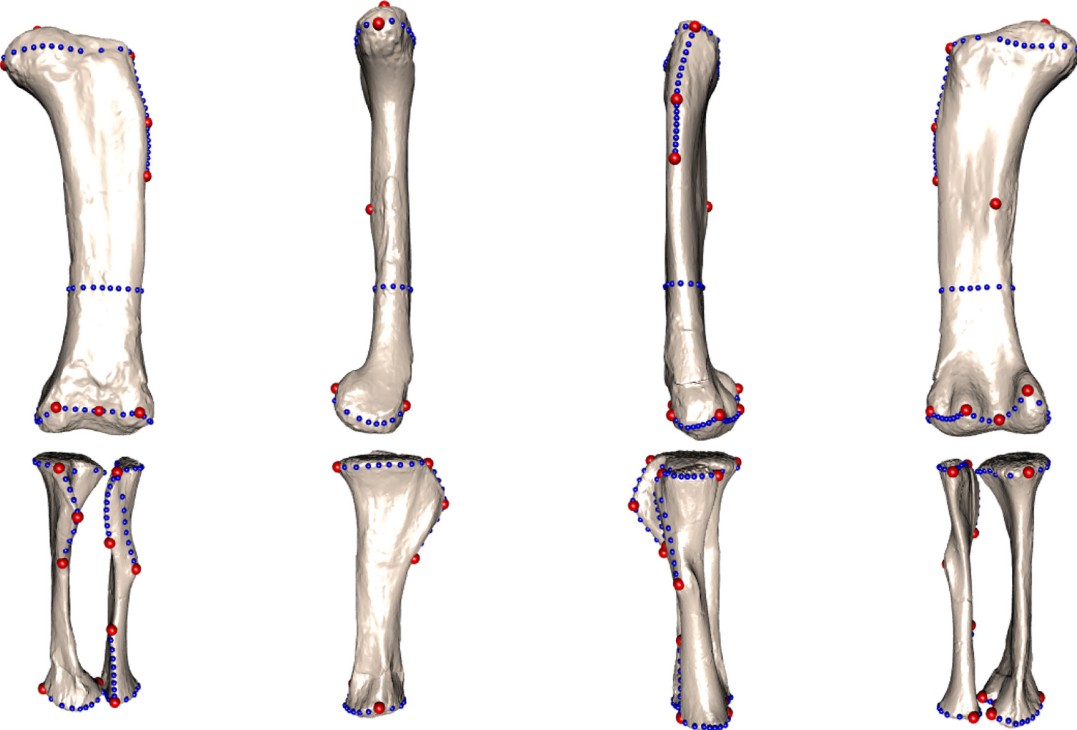

**Appendix 1—figure 4.** Landmark and semilandmark curves used in this study for the entire hind limb. Anterior, medial, lateral, and posterior view.

## Summary phylogenetic supertree

The analysis of the macronarian hind limb morphology evolution requires a stable phylogenetic tree topology. However, the deeply branched macronarian systematics and phylogenetics are still debated (e.g. *Díez Díaz et al., 2021a*; *Jakob et al., 2015*; *González Riga et al., 2018*; *Gorscak et al., 2022*; *Mannion et al., 2019a*; *Mannion et al., 2017*; *Sallam et al., 2018*). We estimated a consensus tree topology using the matrix-representation parsimony supertree methodology (MRP; *Bininda-Emonds, 2004*) with the *phangorn* package (*Schliep et al., 2017*; *Schliep, 2011*). For supertree construction, we compiled several of the more recent phylogenetic hypotheses that include all the available sampled taxa (i.e. *Carballido et al., 2017*; *Csiki et al., 2010*; *Díez Díaz et al., 2018*; *González Riga et al., 2018*; *Mannion et al., 2019a*; *Mocho et al., 2019b*; *Appendix 1—figures 5–10*). The resulting supertree (*Appendix 1—figure 11*) had their tips trimmed to our current macronarian sample. The tree topology was also time-calibrated using the taxon oldest and youngest appearance derived from the bibliography (see cites in age dataset, Supplementary Material 1; see resulting supertree in *Appendix 1—figure 12*). The resulting phylogenetic relationships were projected onto the shape-PCA to visualize the phylomorphospaces.

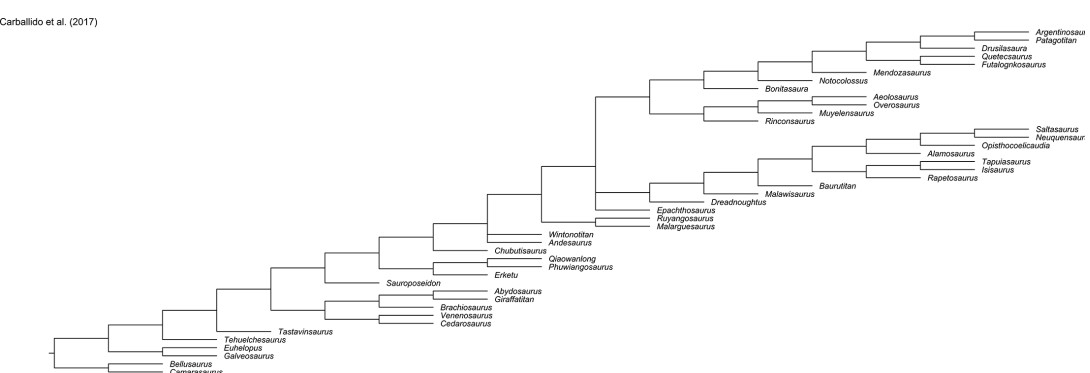

**Appendix 1—figure 5.** Phylogenetic trees used for MRP supertree estimation.

Csiki et al. (2010)

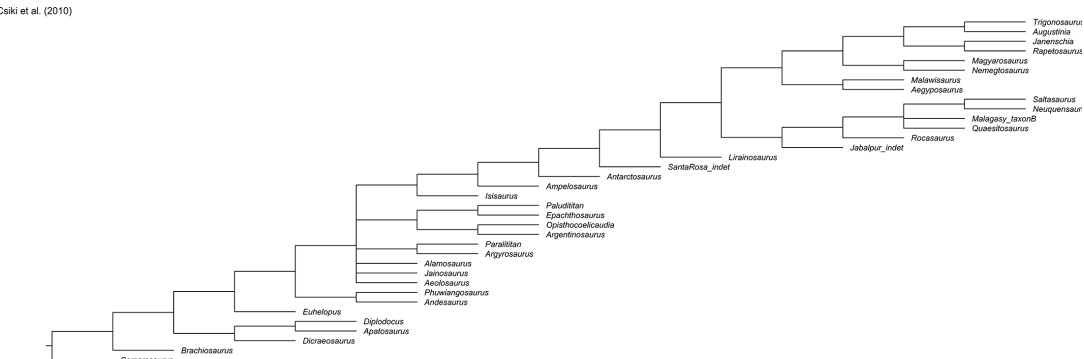

**Appendix 1—figure 6.** Phylogenetic trees used for MRP supertree estimation (continued).

Díez Díaz et al. (2018)

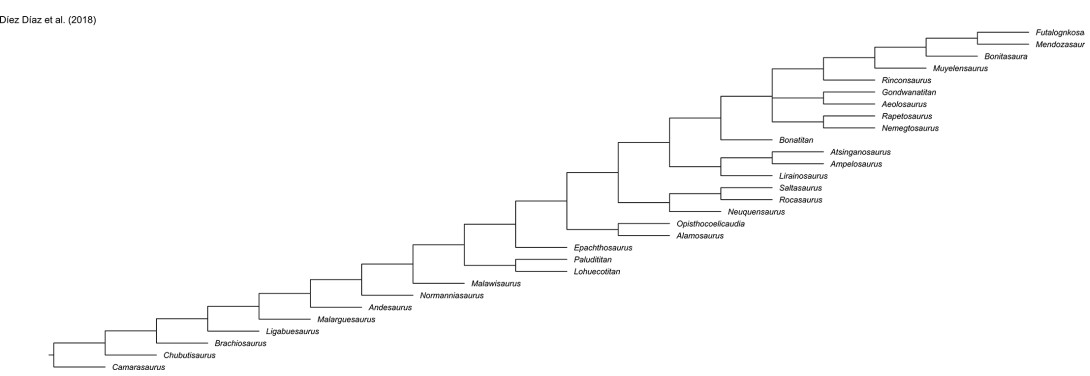

**Appendix 1—figure 7.** Phylogenetic trees used for MRP supertree estimation (continued).

González-Riga et al. (2019)

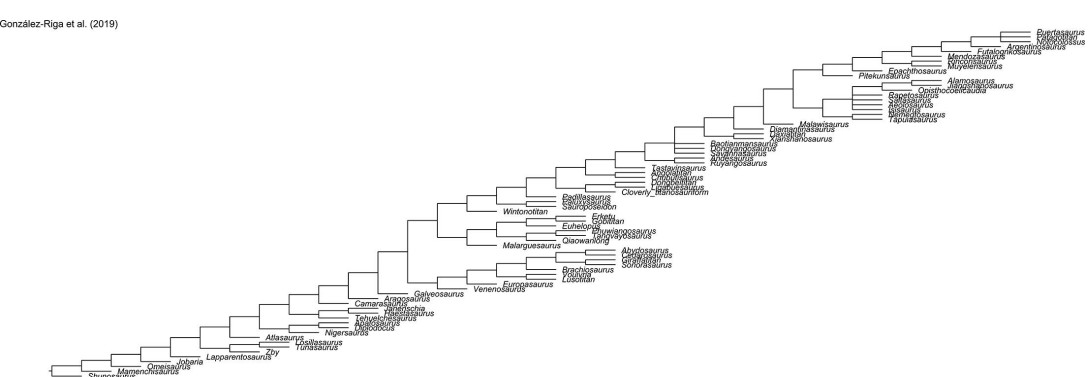

**Appendix 1—figure 8.** Phylogenetic trees used for MRP supertree estimation (continued).

Mannion et al. (2019) - SC with implied weights

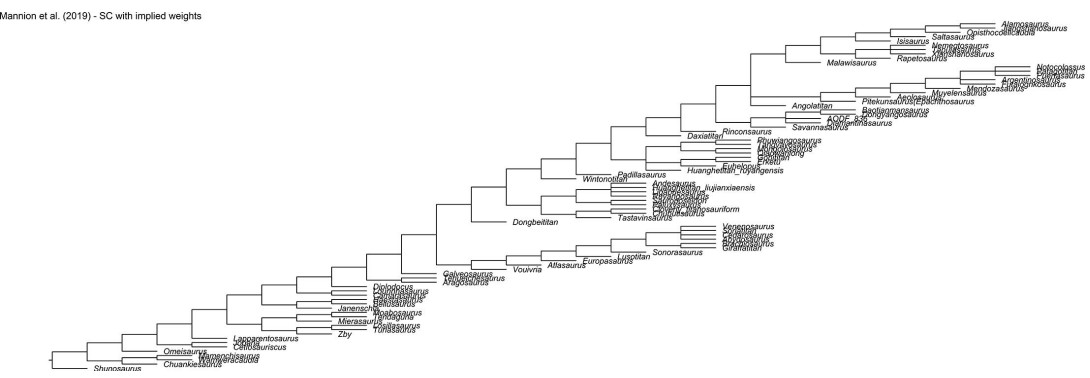

**Appendix 1—figure 9.** Phylogenetic trees used for MRP supertree estimation (continued).

Mocho et al. (2019) - SC excluding taxa

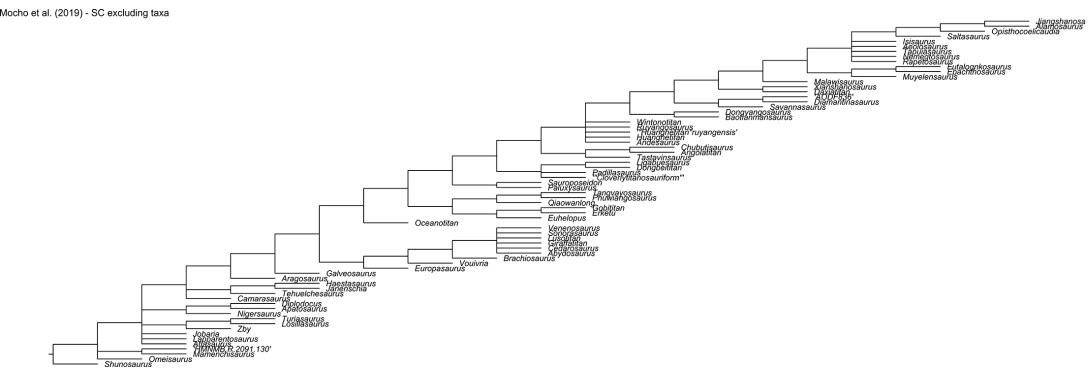

**Appendix 1—figure 10.** Phylogenetic trees used for MRP supertree estimation.

MRP Supertree

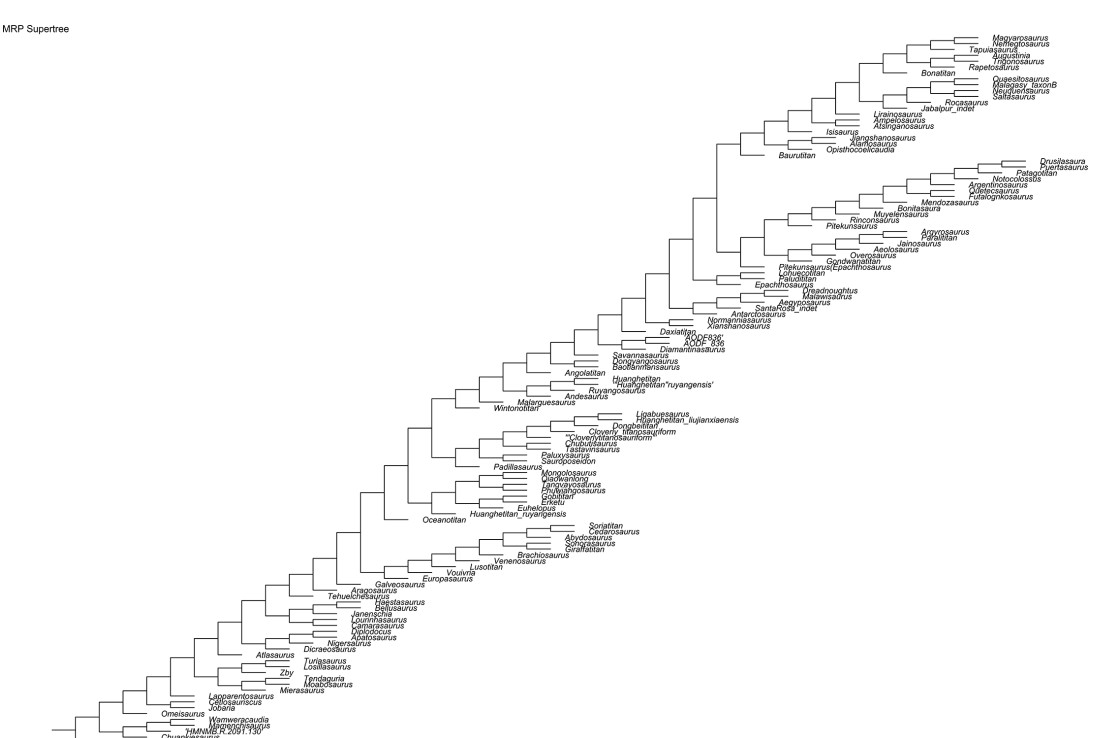

**Appendix 1—figure 11.** Resulting supertree.

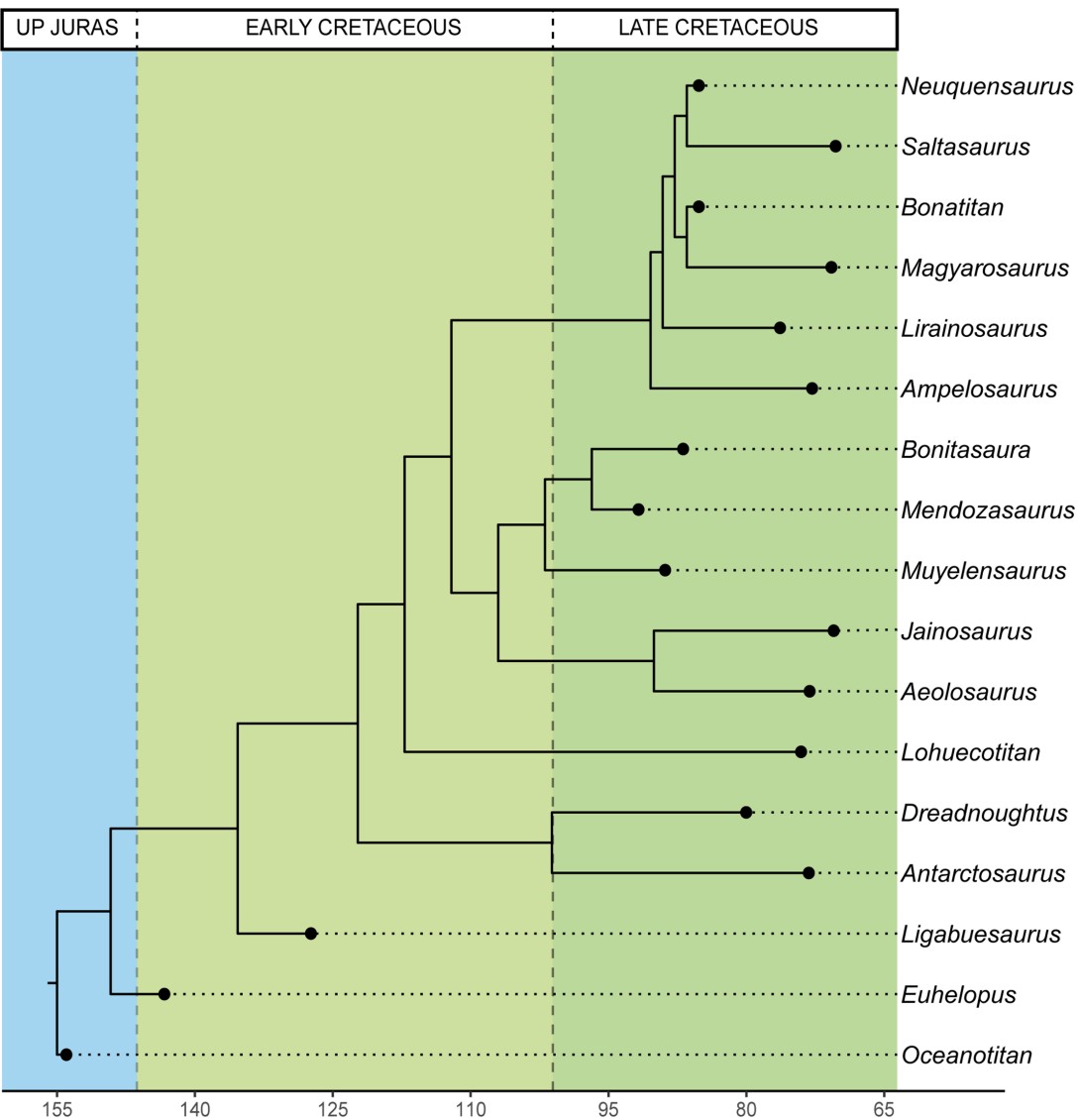

**Appendix 1—figure 12.** Time-calibrated supertree, tips limited to our current sample.

## Morphological convergence

In order to test if there is morphological convergence in a phylogenetic context, we conducted several analyses. We used the standard dissimilarity tests without use of phylogeny topology included in the statistic in order to discuss the possible morphological similarities. The testes included the shape variables and for a group label, we included the most-inclusive ingroup following the published material for each taxon (e.g. *Oceanotitan dantasi* is considered Macronaria – as non-titanosauriform macronarian; *Saltasaurus loricatus* is considered Saltasauridae, and so on). The differences between the subclades were tested via Mann Whitney U's test and Kruskal-Wallis non-parametric tests accounting for the uneven distribution of the group (sub-clade) samples.

We also conducted phylogenetic ANOVA using the time-calibrated supertree topology with the *phytools* R package (**Revell, 2012**).

## Body size proxies

The hind limb size as a proxy of sauropod body size or mass is based on the assumption that, despite absence of data on the fore limb size or the sauropod humerus alone, there is a strong correlation between the hind limb elements and the body mass (**Benson et al., 2014**; **Campione, 2017**; **Mazzetta et al., 2004**). The body mass/size estimation could improve if we take into consideration data from the humerus as they're quadrupedal animals and their hind limb alone may not be enough;

however, estimations are still pretty accurate nonetheless (*Campione and Evans, 2020*). We proxied by a measurement of the landmark configuration specimen centroid size, which is the sum of squared distances between landmarks (*Bookstein, 1991*; *Zelditch et al., 2012*) and thus summarizes the size of the stylopodial and zeugopodial elements. Centroid size may be problematic as it is independent of 'shape' at individual level but differences in the number of landmarks translate into increasing or decreasing centroid sizes of that particular configuration (centroid size equation in *Zelditch et al., 2012*). As we use landmark configurations and semilandmarks mostly bounded by type I and II landmarks (see *Zelditch et al., 2012*), the number of landmarks does not change across the sample. Therefore, the centroid size is comparable as size proxy between the specimens.

The sauropod body mass was proxied by the hind limb centroid size calculated as part of the GPA. Body mass can be estimated preferably by different sets of allometric equations using both humeral and femoral measurements (*Mazzetta et al., 2004*) or whole-body reconstruction volumetric estimations (*Bates et al., 2016*). However, it is not easy to apply both methods to our current sample as the entire skeletons were not digitized, nor publicly available and sometimes incomplete. We opted to assume that the hind limb, as it is the main sauropod body mass support, can be better used as a 'conservative-minimal' proxy to its body mass, with larger hind limb corresponding to giant macronarian taxa. We tested for allometric relationships between the sauropod hind limb shape variables (PCs) and the centroid size as proxy to body mass via Reduced Major Axis (RMA) regression using *lmodel2* R package (*Legendre, 2018*).

To assess that centroid size as a proxy is not problematic, we contrasted our findings with alternative body mass estimations using linear measurements of specimen size and extant scaling approach (following *Campione and Evans, 2020*). We used morphometrics variables such as the humeral circumference, femoral length, and circumference for estimating the evolutionary allometry according to femoral length as proxy of body mass/size (e.g. *Bonnan, 2007*), humeral-femoral circumference for the quadratic equation model from *Campione and Evans, 2012*, and lastly the femoral circumference alone following *Mazzetta et al., 2004*. These equations are derived on the original study of *Anderson et al., 1985* and include slight modifications to tune the model but offer similar results (e.g. *Campione, 2017*; *Campione and Evans, 2020*; *Mazzetta et al., 2004*). The body mass was estimated via *MASSTIMATE* R package (*Campione, 2015*). The results can be accessed in Appendix 2 and *Source code 1*.

## Ancestral character estimation and evolutionary changes

We used the time-calibrated supertree topology and our shape variables (PCs) and hind limb centroid size to generate the Pagel's lambda ($\lambda$) with *phytools* package and test for phylogenetic signal of these traits. All the different hypotheses of statistical correlations, dissimilarity tests, and phylogenetic signal tests were accepted as significant using an alpha level of 0.05 (a comprehensive report of the test results can be accessed in *Supplementary file 2*, and a copy of the R code and packages used can be accessed in *Source code 1*). Once identified, those shape PCs that exhibit significant phylogenetic signal, as well as the log-transformed centroid size, we estimated their ancestral characters (ACEs) using maximum-likelihood and a simple Brownian evolutionary model similar to the one assumed for estimation of Pagel's lambda. We used the ACEs to observe trends in the evolution of titanosaur hind limb size and morphology (based on our shape-PCs). We evaluated the differences in body size (proxied by log-transformed hind limb centroid size ACEs) between terminal taxa and internal nodes and between internal nodes of Titanosauriformes, Somphospondyli, Titanosauria, and Lithostrotia, as well as their subclades. The sum of changes, mean change, median change, positive, negative, and the total amount of changes were evaluated for each of the above clades following *Butler and Goswami, 2008*. We used a $\chi 2$ goodness-of-fit test to evaluate whether body size is increasing or decreasing, and shape-PCs occur at the same frequency (50–50% null hypothesis) or have either a positive or a negative tendency over titanosaurian evolution.

## Appendix 2

### Shape analyses

In this appendix, there is a comprehensive figuration of all the shape analyses including those shape PCs explaining up to 75% of total variance. There is also a section of discussion of the effects of taphonomic deformation and virtual restoration method on the analyses.

### Shape PCA and phylomorphospaces

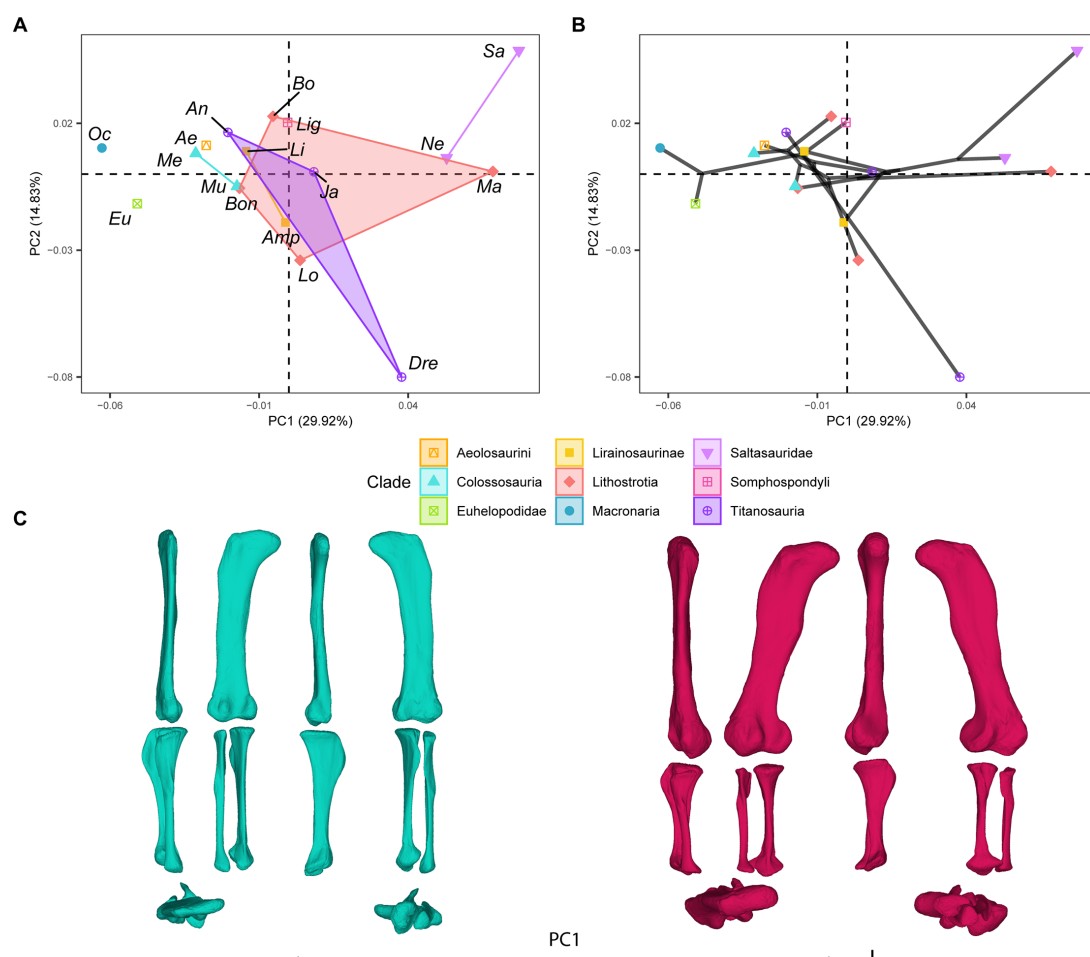

**Appendix 2—figure 1.** PCA results on the GPA aligned landmark and semilandmark curves of the hind limb. (**a**) PC1-PC2 biplot. (**b**) PC1-PC2 phylomorphospace with projected phylogenetic tree. (**c**) Representation of the shape change along PC1, blue are negative scores, red are positive scores. Percentage of variance of each PC in brackets under corresponding axis. *Ae – Aeolosaurus, Amp – Ampelosaurus, An – Antarctosaurus, Bo – Bonatitan, Bon – Bonitasaura, Dre – Dreadnoughtus, Eu – Euhelopus, Ja – Jainosaurus, Li – Lirainosaurus, Lig – Ligabuesaurus, Lo – Lohuecotitan, Ma – Magyarosaurus, Me – Mendozasaurus, Mu – Muyelensaurus, Ne – Neuquensaurus, Sa – Saltasaurus.*

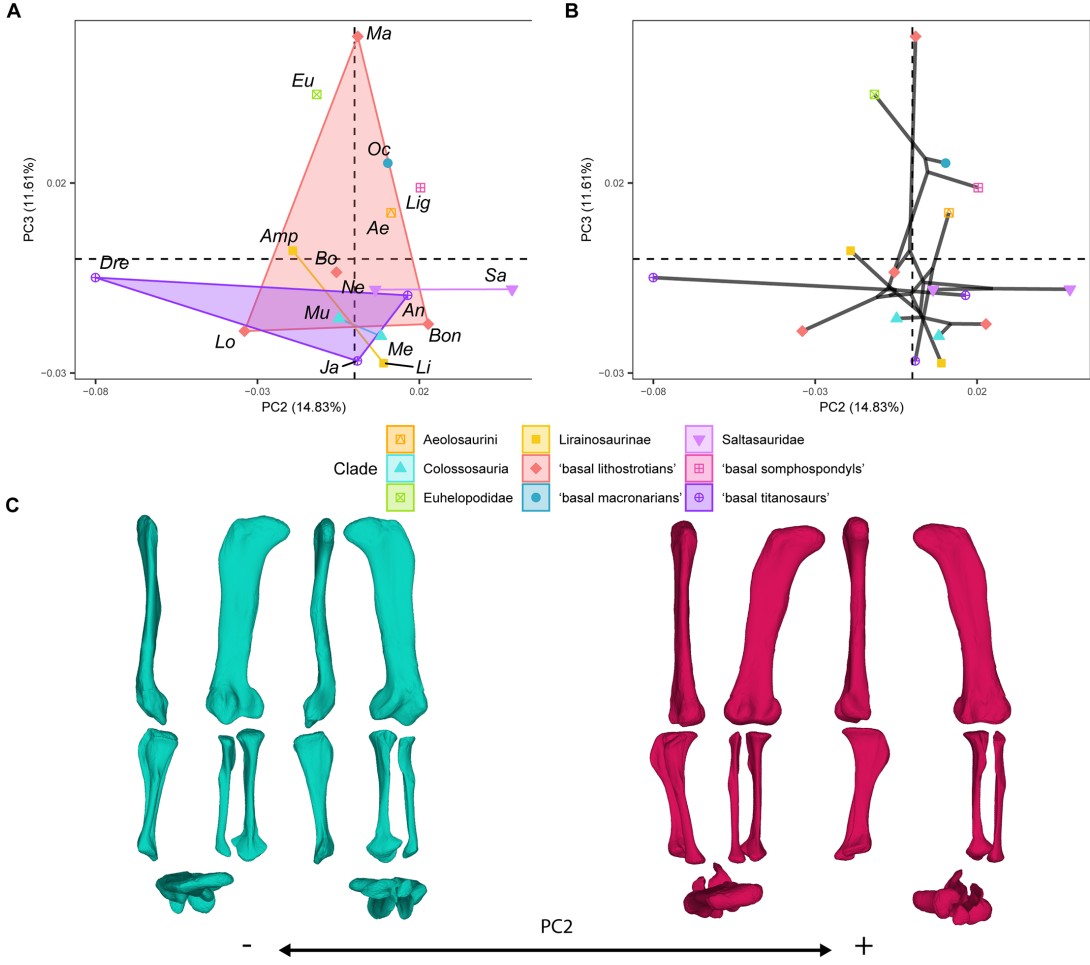

**Appendix 2—figure 2.** PCA results on the GPA aligned landmark and semilandmark curves of the hind limb. (**a**) PC2-PC3 biplot. (**b**) PC2-PC3 phylomorphospace with projected phylogenetic tree. (**c**) Representation of the shape change along PC2, blue are negative scores, red are positive scores. Percentage of variance of each PC in brackets under corresponding axis. *Ae – Aeolosaurus, Amp – Ampelosaurus, An – Antarctosaurus, Bo – Bonatitan, Bon – Bonitasaura, Dre – Dreadnoughuts, Eu – Euhelopus, Ja – Jainosaurus, Li – Lirainosaurus, Lig – Ligabuesaurus, Lo – Lohuecotitan, Ma – Magyarosaurus, Me – Mendozasaurus, Mu – Muyelensaurus, Ne – Neuquensaurus, Sa – Saltasaurus.*

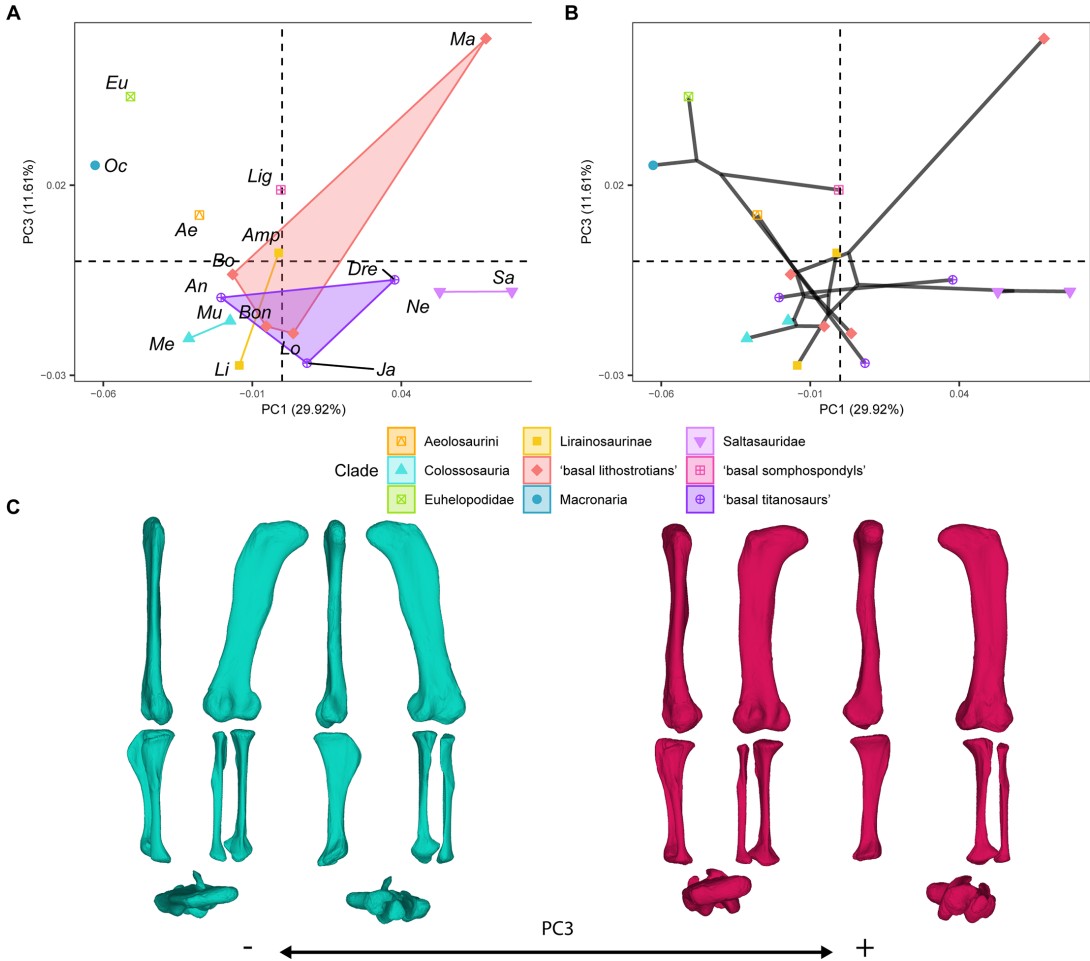

**Appendix 2—figure 3.** PCA results on the GPA aligned landmark and semilandmark curves of the hind limb. (**a**) PC1-PC3 biplot. (**b**) PC1-PC3 phylomorphospace with projected phylogenetic tree. (**c**) Representation of the shape change along PC3, blue are negative scores, red are positive scores. Percentage of variance of each PC in brackets under corresponding axis. *Ae – Aeolosaurus, Amp – Ampelosaurus, An – Antarctosaurus, Bo – Bonatitan, Bon – Bonitasaura, Dre – Dreadnoughtus, Eu – Euhelopus, Ja – Jainosaurus, Li – Lirainosaurus, Lig – Ligabuesaurus, Lo – Lohuecotitan, Ma – Magyarosaurus, Me – Mendozasaurus, Mu – Muyelensaurus, Ne – Neuquensaurus, Sa – Saltasaurus.*

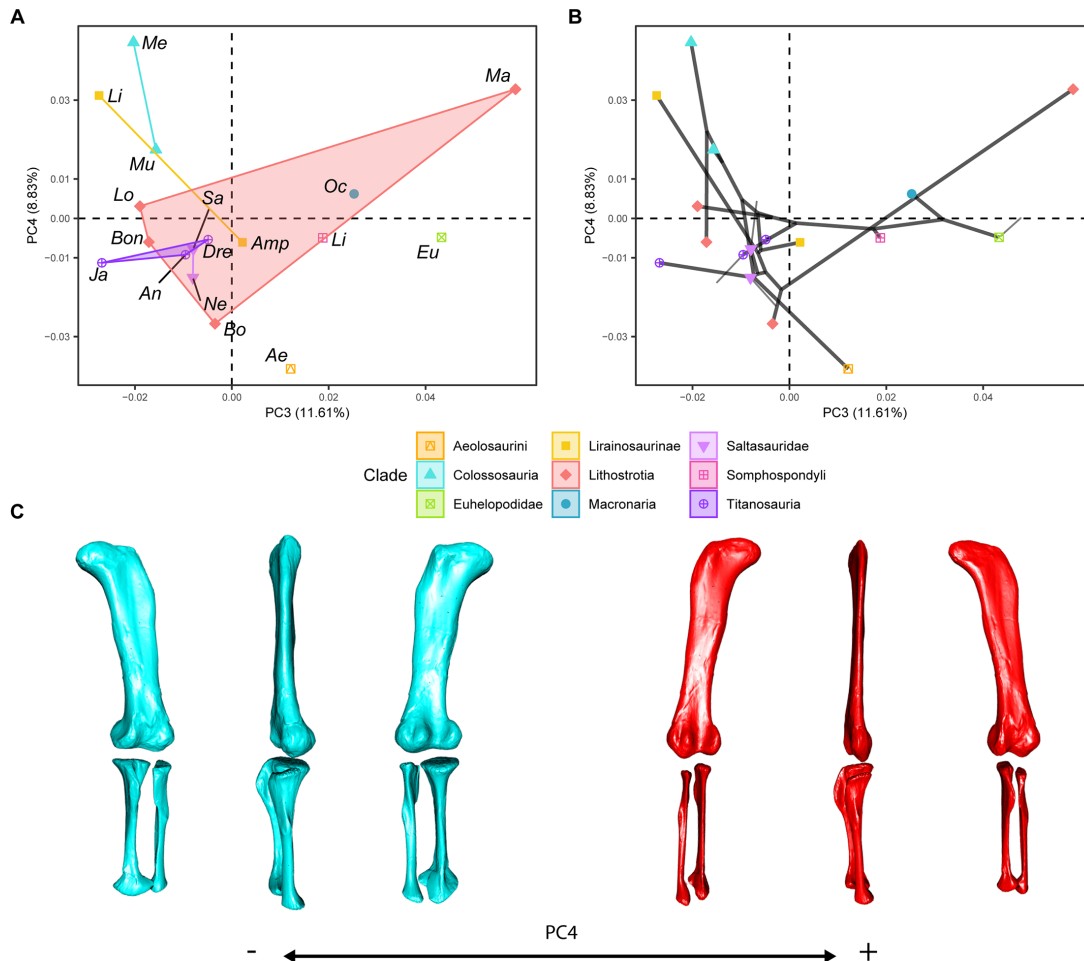

**Appendix 2—figure 4.** PCA results on the GPA aligned landmark and semilandmark curves of the hind limb. (**a**) PC3-PC4 biplot. (**b**) PC3-PC4 phylomorphospace with projected phylogenetic tree. (**c**) Representation of the shape change along PC4, blue are negative scores, red are positive scores. Percentage of variance of each PC in brackets under corresponding axis. *Ae – Aeolosaurus, Amp – Ampelosaurus, An – Antarctosaurus, Bo – Bonatitan, Bon – Bonitasaura, Dre – Dreadnoughtus, Eu – Euhelopus, Ja – Jainosaurus, Li – Lirainosaurus, Lig – Ligabuesaurus, Lo – Lohuecotitan, Ma – Magyarosaurus, Me – Mendozasaurus, Mu – Muyelensaurus, Ne – Neuquensaurus, Sa – Saltasaurus.*

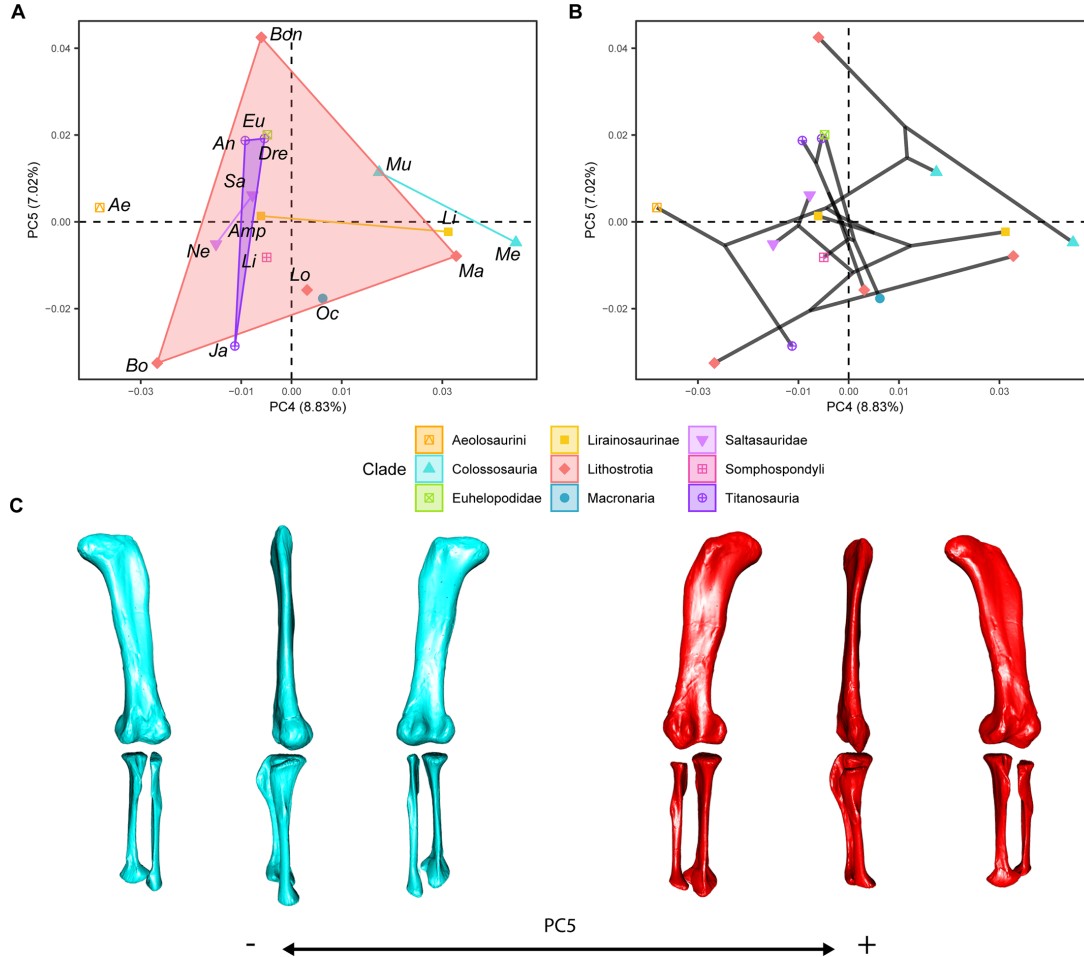

**Appendix 2—figure 5.** PCA results on the GPA aligned landmark and semilandmark curves of the hind limb. (**a**) PC4-PC5 biplot. (**b**) PC4-PC5 phylomorphospace with projected phylogenetic tree. (**c**) Representation of the shape change along PC5, blue are negative scores, red are positive scores. Percentage of variance of each PC in brackets under corresponding axis. *Ae – Aeolosaurus, Amp – Ampelosaurus, An – Antarctosaurus, Bo – Bonatitan, Bon – Bonitasaura, Dre – Dreadnoughtus, Eu – Euhelopus, Ja – Jainosaurus, Li – Lirainosaurus, Lig – Ligabuesaurus, Lo – Lohuecotitan, Ma – Magyarosaurus, Me – Mendozasaurus, Mu – Muyelensaurus, Ne – Neuquensaurus, Sa – Saltasaurus.*

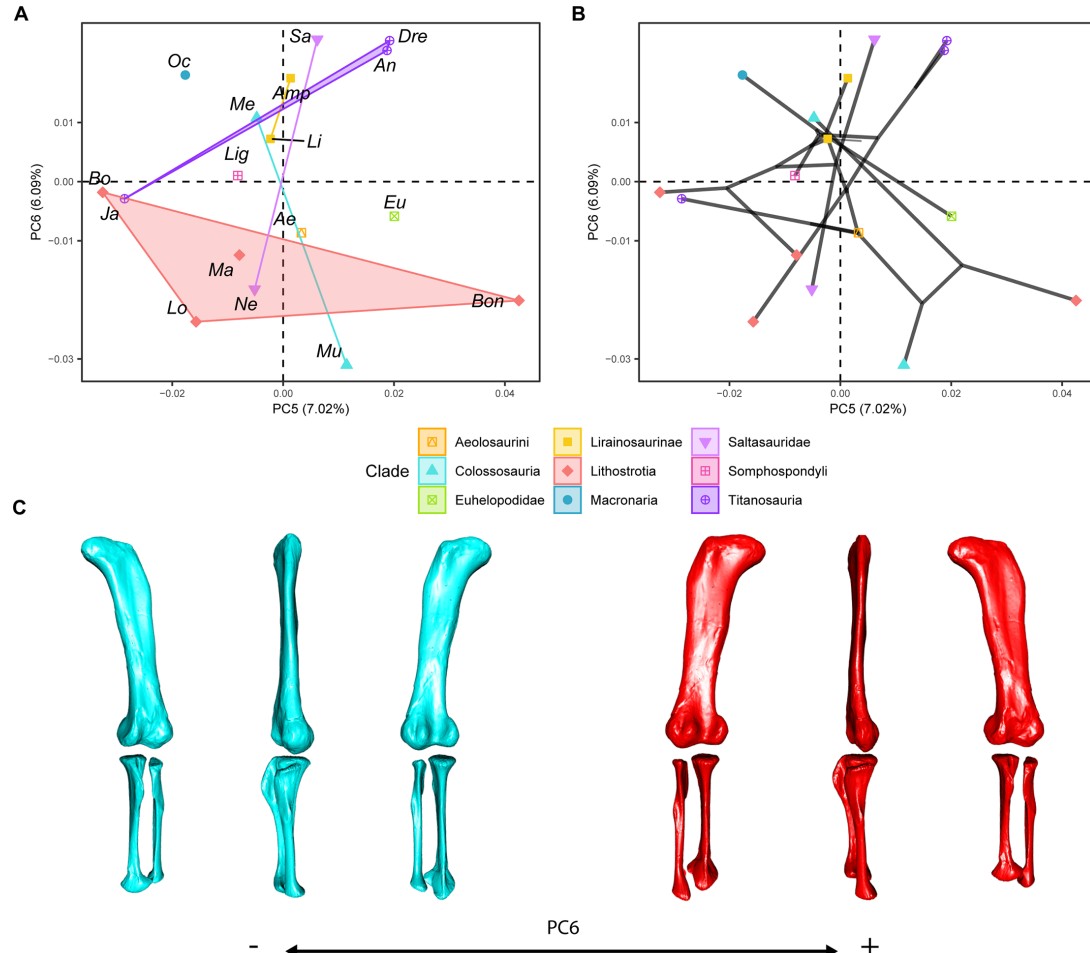

**Appendix 2—figure 6.** PCA results on the GPA aligned landmark and semilandmark curves of the hind limb. (**a**) PC5-PC6 biplot. (**b**) PC5-PC6 phylomorphospace with projected phylogenetic tree. (**c**) Representation of the shape change along PC6, blue are negative scores, red are positive scores. Percentage of variance of each PC in brackets under corresponding axis. *Ae – Aeolosaurus, Amp – Ampelosaurus, An – Antarctosaurus, Bo – Bonatitan, Bon – Bonitasaura, Dre – Dreadnoughuts, Eu – Euhelopus, Ja – Jainosaurus, Li – Lirainosaurus, Lig – Ligabuesaurus, Lo – Lohuecotitan, Ma – Magyarosaurus, Me – Mendozasaurus, Mu – Muyelensaurus, Ne – Neuquensaurus, Sa – Saltasaurus.*

## RMA models with shape PCs – centroid size

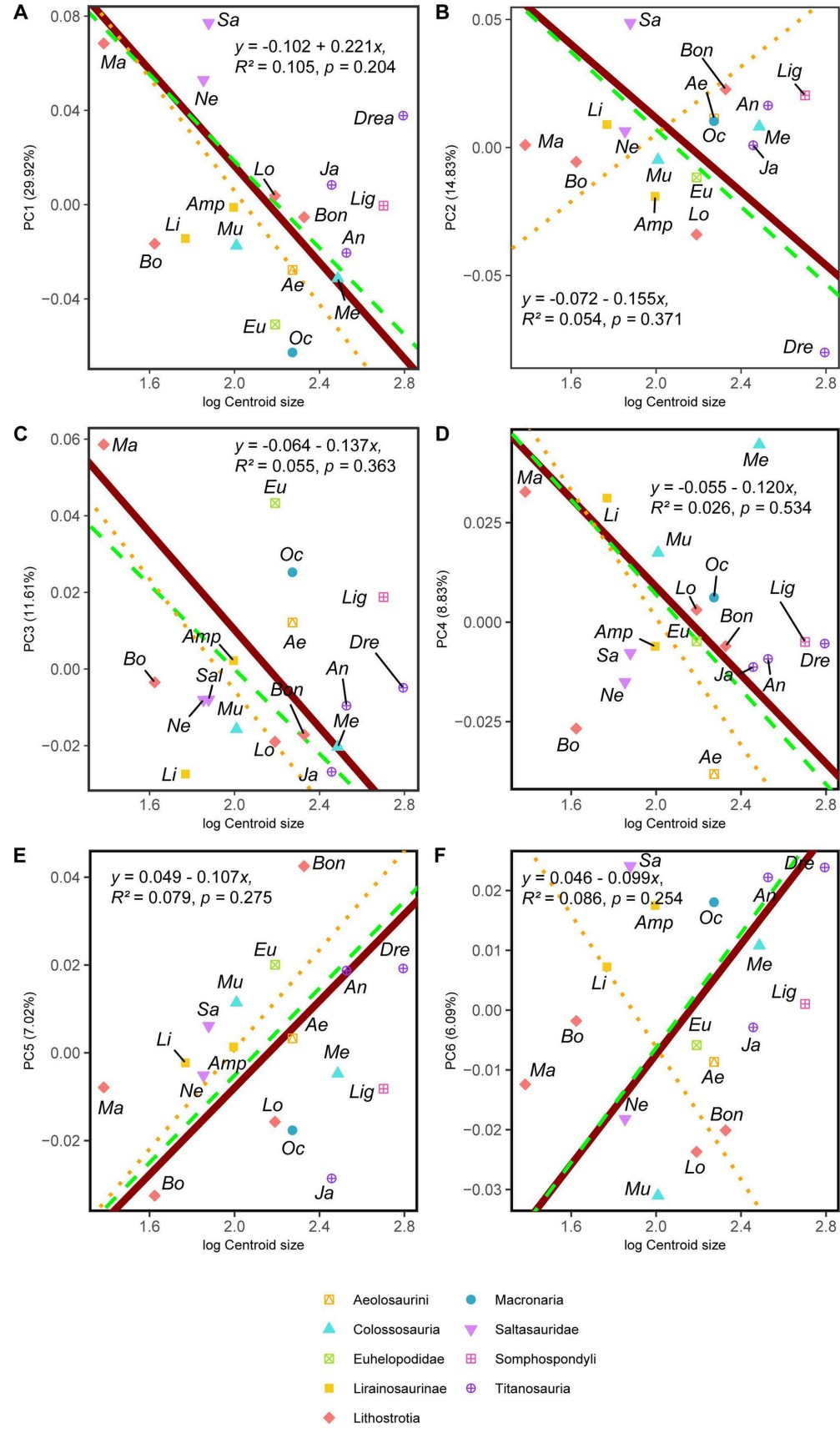

**Appendix 2—figure 7.** RMA results of the first three shape PCs against the logarithm of the hind limb centroid size. (**a**) PC1 against log-Centroid size, all taxa RMA in dark red: intercept = 0.221, slope = –0.102, $r^2$=0.105, p=0.204; Titanosauria only partial RMA in dashed green: intercept = 0.203, slope = –0.092, $r^2$=0.118, p=0.229; Lithostrotia only partial RMA in dotted orange: intercept = 0.246, slope = –0.120, $r^2$=0.319, p=0.07; (**b**) PC2 against log-Centroid size, all taxa RMA in dark red: intercept = 0.155, slope = –0.072, $r^2$=0.054, p=0.371; Titanosauria only partial RMA in dashed green: intercept = 0.158, slope = –0.075, $r^2$=0.117, p=0.232; Lithostrotia only partial RMA in dotted orange: intercept = –0.127, slope = 0.066, $r^2$=0, p=0.952; (**c**) PC3 against log-Centroid size (Csize), all taxa RMA in dark red: intercept = 0.137, slope = –0.064, $r^2$=0.055, p=0.363; Titanosauria only partial RMA in dashed green: intercept = 0.110, slope = –0.055, $r^2$=0.236, p=0.078; Lithostrotia only partial RMA in dotted orange: intercept = 0.140, slope = –0.073, $r^2$=0.313, p=0.074. (**d**) PC4 against log-Centroid size (Csize), all taxa RMA in dark red: intercept = 0.120, slope = –0.055, $r^2$=0.026, p=0.534; Titanosauria only partial RMA in dashed green: intercept = 0.126, slope = –0.060, $r^2$=0.025, p=0.592; Lithostrotia only partial RMA in dotted orange: intercept = 0.161, slope = –0.080, $r^2$=0.002, p=0.903. (**e**) PC5 against log-Centroid size (Csize), all taxa RMA in dark red: intercept = –0.107, slope = 0.049, $r^2$=0.079, p=0.275; Titanosauria only partial RMA in dashed green: intercept = –0.104, slope = –0.050, $r^2$=0.1506, p=0.171; Lithostrotia only partial RMA in dotted orange: intercept = –0.113, slope = 0.057, $r^2$=0.204, p=0.163. (**f**) PC6 against log-Centroid size (Csize), all taxa RMA in dark red: intercept = –0.100, slope = 0.046, $r^2$=0.086, p=0.254; Titanosauria only partial RMA in dashed green: intercept = –0.102, slope = –0.048, $r^2$=0.089, p=0.302; Lithostrotia only partial RMA in dotted orange: intercept = 0.105, slope = –0.055, $r^2$=0.003, p=0.871. Ae – Aeolosaurus, Amp – Ampelosaurus, An – Antarctosaurus, Bo – Bonatitan, Bon – Bonitasaura, Dre – Dreadnoughtus, Eu – Euhelopus, Ja – Jainosaurus, Li – Lirainosaurus, Lig – Ligabuesaurus, Lo – Lohuecotitan, Ma – Magyarosaurus, Me – Mendozasaurus, Mu – Muyelensaurus, Ne – Neuquensaurus, Oc – Oceanotitan, Sa – Saltasaurus.

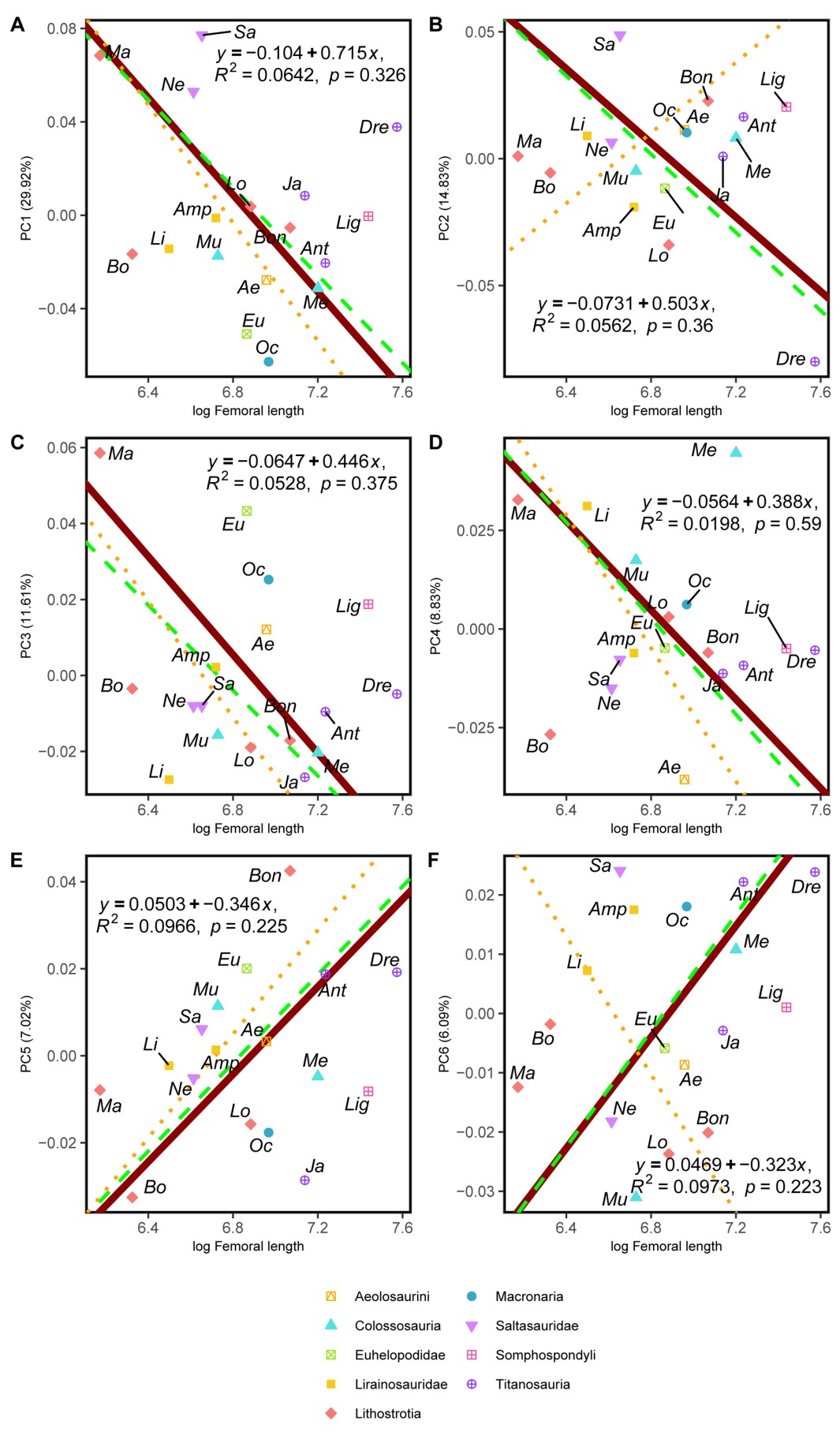

**Appendix 2—figure 8.** RMA results of the first three shape PCs against the logarithm of the hind limb femoral length in mm. (**a**) PC1 against log-femoral length, all taxa RMA in dark red: intercept = 0.715, slope = –0.104, $r^2$=0.064, p=0.326; Titanosauria only partial RMA in dashed green: intercept = 0.651, slope = –0.094, $r^2$=0.177, p=0.337; Lithostrotia only partial RMA in dotted orange: intercept = 0.856, slope = –0.126, $r^2$=0.252, p=0.115; (**b**) PC2 against log-Femoral length, all taxa RMA in dark red: intercept = 0.503, slope = –0.073, $r^2$=0.056, p=0.36; Titanosauria only partial RMA in dashed green: intercept = 0.526, slope = –0.077, $r^2$=0.125, p=0.215; Lithostrotia only partial RMA in dotted orange: intercept = –0.461, slope = 0.069, $r^2$=0.006, p=0.823; (**c**) PC3 against log-Femoral length, all taxa RMA in dark red: intercept = 0.446, slope = –0.065, $r^2$=0.053, p=0.375; Titanosauria only partial RMA in dashed green: intercept = 0.378, slope = –0.056, $r^2$=0.207, p=0.102; Lithostrotia only partial RMA in dotted orange: intercept = 0.509, slope = –0.076, $r^2$=0.296, p=0.084. (**d**) PC4 against log-Femoral length, all taxa RMA in dark red: intercept = 0.388, slope = –0.056, $r^2$=0.02, p=0.59; Titanosauria only partial RMA in dashed green: intercept = 0.416, slope = –0.061, $r^2$=0.018, p=0.647; Lithostrotia only partial RMA in dotted orange: intercept = 0.569, slope = –0.084, $r^2$=0.000, p=0.972. (**e**) PC5 against log-Femoral length, all taxa RMA in dark red: intercept = –0.346, slope = 0.05, $r^2$=0.097, p=0.225; Titanosauria only partial RMA in dashed green: intercept = –0.346, slope = 0.051, $r^2$=0.186, p=0.124; Lithostrotia only partial RMA in dotted orange: intercept = –0.401, slope = 0.06, $r^2$=0.252, p=0.116. (**f**) PC6 against log-Femoral length, all taxa RMA in dark red: intercept = –0.323, slope = 0.047, $r^2$=0.097, p=0.223; Titanosauria only partial RMA in dashed green: intercept = –0.336, slope = –0.049, $r^2$=0.104, p=0.261; Lithostrotia only partial RMA in dotted orange: intercept = 0.386, slope = –0.058, $r^2$=0.002, p=0.903. *Ae – Aeolosaurus, Amp – Ampelosaurus, An – Antarctosaurus, Bo – Bonatitan, Bon – Bonitasaura, Dre – Dreadnoughuts, Eu – Euhelopus, Ja – Jainosaurus, Li – Lirainosaurus, Lig – Ligabuesaurus, Lo – Lohuecotitan, Ma – Magyarosaurus, Me – Mendozasaurus, Mu – Muyelensaurus, Ne – Neuquensaurus, Oc – Oceanotitan, Sa – Saltasaurus.*

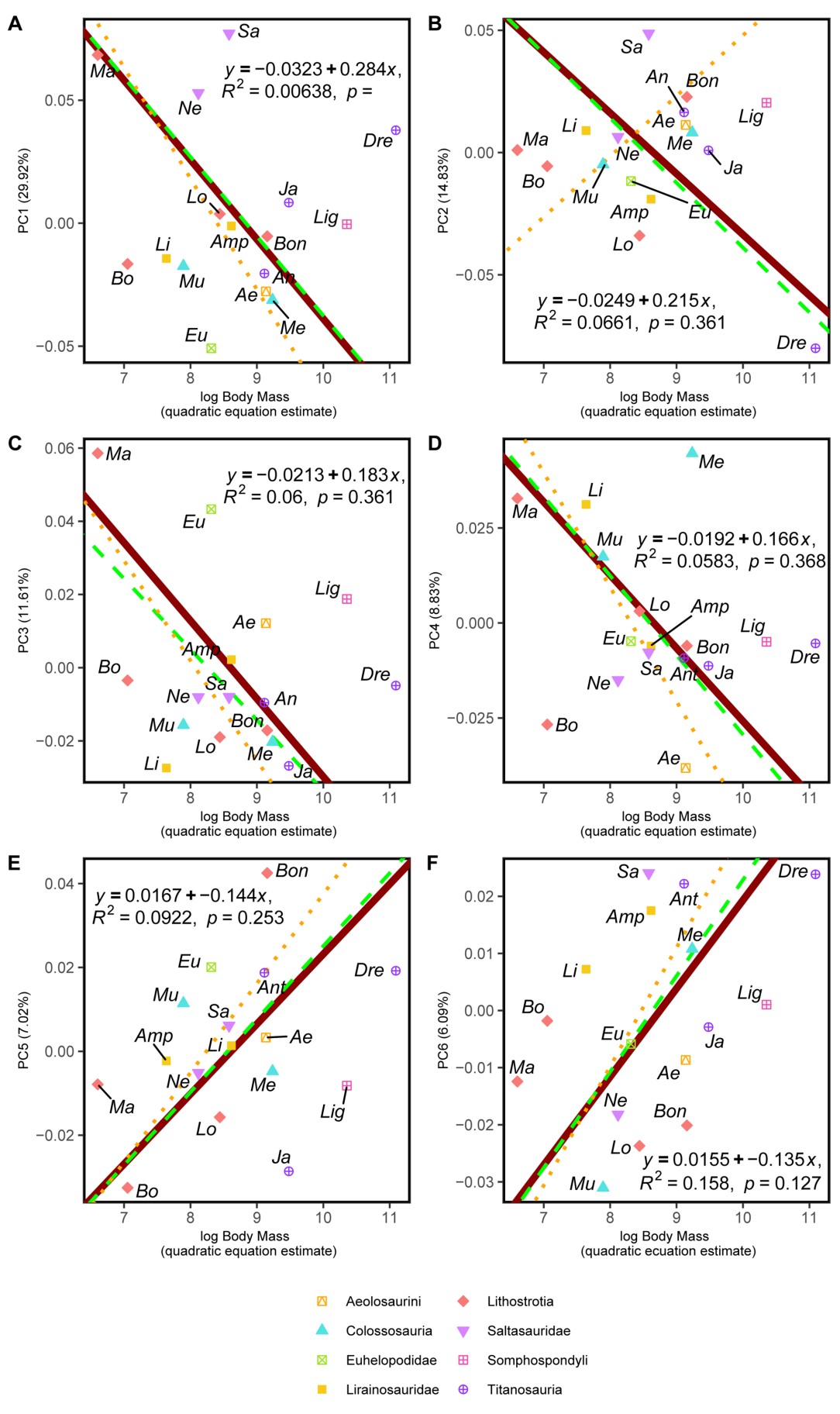

**Appendix 2—figure 9.** RMA results of the first three shape PCs against the logarithm of the body mass in kilograms, estimated via quadratic equation method (*Campione, 2017*; *Campione, 2017*). (**a**) PC1 against log-body mass, all taxa RMA in dark red: intercept = 0.284, slope = –0.032, $r^2$=0.006, p=0.769; Titanosauria only partial RMA in dashed green: intercept = 0.285, slope = –0.032, $r^2$=0.012, p=0.712; Lithostrotia only partial RMA in dotted orange: intercept = 0.379, slope = –0.045, $r^2$=0.119, p=0.299; (**b**) PC2 against log-Body mass, all taxa RMA in dark red: intercept = 0.215, slope = –0.025, $r^2$=0.066, p=0.336; Titanosauria only partial RMA in dashed green: intercept = 0.226, slope = –0.026, $r^2$=0.119, p=0.18; Lithostrotia only partial RMA in dotted orange: intercept = –0.2, slope = 0.025 $r^2$=0,053, p=0.496; (**c**) PC3 against log-Body mass, all taxa RMA in dark red: intercept = 0.183, slope = –0.021, $r^2$=0.06, p=0.361; Titanosauria only partial RMA in dashed green: intercept = 0.159, slope = –0.019, $r^2$=0.158, p=0.159; Lithostrotia only partial RMA in dotted orange: intercept = 0.22, slope = –0.027, $r^2$=0.239, p=0.127. (**d**) PC4 against log-Body mass, all taxa RMA in dark red: intercept = 0.166, slope = –0.019, $r^2$=0.058, p=0.368; Titanosauria only partial RMA in dashed green: intercept = 0.18, slope = –0.021, $r^2$=0.059, p=0.401; Lithostrotia only partial RMA in dotted orange: intercept = 0.25, slope = –0.03, $r^2$=0.038, p=0.566. (**e**) PC5 against log-Body mass, all taxa RMA in dark red: intercept = –0.144, slope = 0.017, $r^2$=0.092, p=0.253; Titanosauria only partial RMA in dashed green: intercept = –0.149, slope = 0.017, $r^2$=0.178, p=0.133; Lithostrotia only partial RMA in dotted orange: intercept = –0.176, slope = 0.021, $r^2$=0.289, p=0.088. (**f**) PC6 against log-Body mass, all taxa RMA in dark red: intercept = –0.100, slope = 0.046, $r^2$=0.086, p=0.254; Titanosauria only partial RMA in dashed green: intercept = –0.145, slope = 0.017, $r^2$=0.173, p=0.139; Lithostrotia only partial RMA in dotted orange: intercept = –0.176, slope = 0.021, $r^2$=0.025, p=0.642. *Ae – Aeolosaurus, Amp – Ampelosaurus, An – Antarctosaurus, Bo – Bonatitan, Bon – Bonitasaura, Dre – Dreadnoughtus, Eu – Euhelopus, Ja – Jainosaurus, Li – Lirainosaurus, Lig – Ligabuesaurus, Lo – Lohuecotitan, Ma – Magyarosaurus, Me – Mendozasaurus, Mu – Muyelensaurus, Ne – Neuquensaurus, Oc – Oceanotitan, Sa – Saltasaurus.*

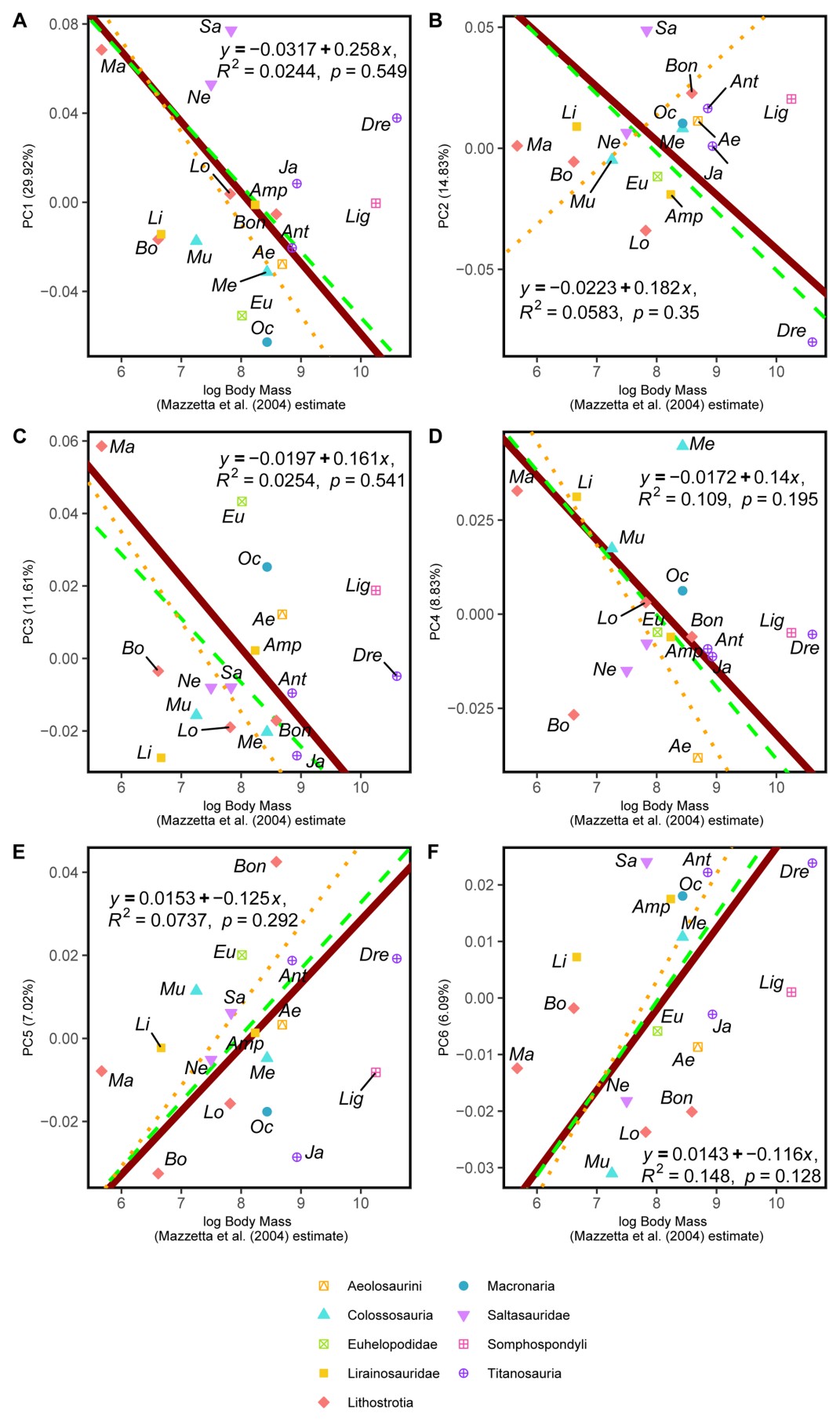

**Appendix 2—figure 10.** RMA results of the first three shape PCs against the logarithm of the body mass in kilograms, estimated via hind limb multiple regression proposed by *Mazzetta et al., 2004*. (**a**) PC1 against log-body mass, all taxa RMA in dark red: intercept = 0.258, slope = –0.032, $r^2$=0.024, p=0.549; Titanosauria only partial RMA in dashed green: intercept = 0.244, slope = –0.03, $r^2$=0.024, p=0.597; Lithostrotia only partial RMA in dotted orange: intercept = 0.319, slope = –0.041, $r^2$=0.147, p=0.245; (**b**) PC2 against log-Centroid size, all taxa RMA in dark red: intercept = 0.182, slope = –0.022, $r^2$=0.025, p=0.35; Titanosauria only partial RMA in dashed green: intercept = 0.192, slope = –0.024, $r^2$=0.149, p=0.173; Lithostrotia only partial RMA in dotted orange: intercept = –0.167, slope = 0.023, $r^2$=0.023, p=0.657; (**c**) PC3 against log-Body mass, all taxa RMA in dark red: intercept = 0.161, slope = –0.02, $r^2$=0.025, p=0.541; Titanosauria only partial RMA in dashed green: intercept = 0.135, slope = –0.018, $r^2$=0.148, p=0.174; Lithostrotia only partial RMA in dotted orange: intercept = 0.184, slope = –0.025, $r^2$=0.215, p=0.151. (**d**) PC4 against log-Body mass, all taxa RMA in dark red: intercept = 0.14, slope = –0.017, $r^2$=0.109, p=0.195; Titanosauria only partial RMA in dashed green: intercept = 0.153, slope = –0.019, $r^2$=0.122, p=0.221; Lithostrotia only partial RMA in dotted orange: intercept = 0.21, slope = –0.027, $r^2$=0.117, p=0.302. (**e**) PC5 against log-Body mass, all taxa RMA in dark red: intercept = –0.125, slope = 0.015, $r^2$=0.074, p=0.292; Titanosauria only partial RMA in dashed green: intercept = –0.127, slope = 0.016, $r^2$=0.169, p=0.145; Lithostrotia only partial RMA in dotted orange: intercept = –0.147, slope = 0.019, $r^2$=0.242, p=0.125. (**f**) PC6 against log-Body mass, all taxa RMA in dark red: intercept = –0.116, slope = 0.014, $r^2$=0.148, p=0.128; Titanosauria only partial RMA in dashed green: intercept = –0.124, slope = 0.015, $r^2$=0.17, p=0.142; Lithostrotia only partial RMA in dotted orange: intercept = –0.149, slope = –0.053, $r^2$=0.015, p=0.716. *Ae – Aeolosaurus, Amp – Ampelosaurus, An – Antarctosaurus, Bo – Bonatitan, Bon – Bonitasaura, Dre – Dreadnoughtus, Eu – Euhelopus, Ja – Jainosaurus, Li – Lirainosaurus, Lig – Ligabuesaurus, Lo – Lohuecotitan, Ma – Magyarosaurus, Me – Mendozasaurus, Mu – Muyelensaurus, Ne – Neuquensaurus, Oc – Oceanotitan, Sa – Saltasaurus.*

## Effects of the Taphonomy and the Statistical Virtual Restoration on the shape analyses

In this section, we will assess the effects of taphonomy and the virtual restoration and mean hind limb shape of the specimens, following the method of *Lefebvre et al., 2020*. Also, all the shape PCA results should be read as a composite between taphonomic deformation and shape variation between the analyzed specimens, or taphomorphospaces (*Hedrick and Dodson, 2013*). We estimated the mean shapes of the hind limb elements including only the best preserved specimens for each multi-specimen taxon previously analyzed in *Páramo, 2020*. As previous analyses suggest, effects of specimen deformation have low impact in the sample of reconstructed specimens (*Páramo, 2020*). However, some deformation in the input is unavoidable due to lack of multiple specimens of the same taxon or general degree of deformation in the entire sample for the same taxon due to preservation of a particular fossil site.

The reconstructed femur of *Oceanotitan dantasi* exhibits some degree of 'sigmoid' curvature to posterior at the height of the fourth trochanter and then to anterior again at the proximal most area and the proximal end. This might have had some impact if we considered a larger set of landmarks and curves like in *Páramo, 2020*, but not in our reduced set of landmarks compared to previous studies. The virtual reconstruction of the holotype specimens shares some similarities with the other non-Titanosaurian, columnar hind limb of *Euhelopus zdanskyi*, which also exhibits this morphology in the femur (PMU-234). In all the analyses, this displacement of the femoral head does not seem to translate into an anomalous shape variation respective to the sample, and the phylomorphospaces exhibit that it commonly falls in an area of the (tapho)morphospace dissimilar to the open hind limb of the titanosaurian sauropods due to their 'wide gauge' posture acquisition (e.g. Appendix 2.3.2). Maybe it would alter its position in PC5 which would have been in slightly higher positive values (*Appendix 2—figure 6*), probably overlapping with saltasaurid specimens. The proximal part of the hind limb, especially the proximal end of the femur, may resemble the femora of some saltasaurid sauropods with the dorsally deflected femoral head, whereas the rest of the hind limb still exhibits the plesiomorphic non-Titanosauriformes macronarian morphology. It is noticeable that the fragmentary hind limb of *O. dantasi* also exhibits some lithostatic deformation, with slightly collapsed shafts that barely translate in noticeable effects in our analyses, but some degree of warping, especially in the right tibia SHN-181–31. We cannot completely override that the proximal end of the tibia is lateromedially narrow due to lithostatic compaction (see *Mocho et al., 2019b*). However, the distal end of the tibia preserved its original lateromedial width, and it would be uncommon that one end is affected and in the other one the lithostatic deformation is barely noticeable, especially in absence of hints of complex tectonic deformations. The medial concavity of the shaft is due to deformation

and fracturing, and it does affect our results, as it conditions the articulation with the fibula and overall length of the tibia, but the effects are quite light. On the other hand, there are no hints of rotational deformation along the shaft axis that may alter the position of the distal end of the tibia, nor the placement of the fibular articulation with the tibia.

The holotype specimen of *Bonitasaura salgadoi* does not preserve the proximal end of the femur but also exhibits similar anterior bevelling of the lateral bulge (*Appendix 2—figure 11*), which may produce a morphology closer to other lithostrotian sauropods like *Lohuecotitan pandafilandi*. When inspecting the holotype specimen of *Bonitasaura salgadoi* (MPCA-460), this anterior deflection of the lateral bulge can be a by-product of taphonomic deformation as well as our virtual restoration method or either a true anterior deflection, as commented before (*Appendix 2—figure 1*). However, none of the results exhibit insights of incorrect plotting of *B. salgadoi* as the anterior deflection of the lateral bulge has low impact alone. It is often separated even in PC4 and PC5 (*Appendix 2— figure 5*) where it captures the anterior deflection of the lateral bulge and the area of the greater trochanter, despite exhibiting morphological similarities to *Ampelosaurus atacis* and *Lohuecotitan pandafilandi* due to the virtual restoration process.

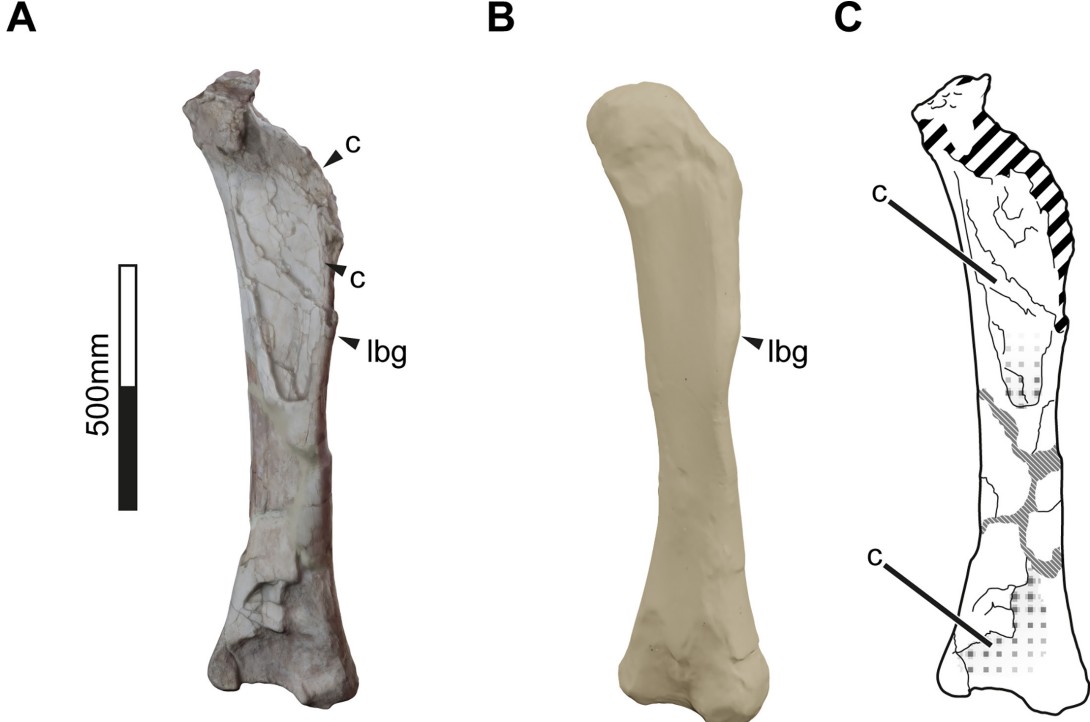

**A** **B** **C**

**Appendix 2—figure 11.** 3D reconstruction of *Bonitasaura salgadoi* holotype (MPCA-460) left femur in anteolateral view. (**A**) Original specimen 3D reconstruction. (**B**) Complete femur as reconstructed by the statistical virtual restoration method. (**C**) Diagram indicating the partial collapse of the anterior view of the shaft and the fragmentary proximal end. Abbreviations: c – collapse, lbg – lateral bulge. Discontinued lines/fill indicate fracture.

Despite possible bias in their overall position on the taphomorphospaces, our analyses seem to remain unaffected whether *B. salgadoi* lateral bulge is warped due to the collapse of the proximal end or it truly does exhibit an anterior deflection similar to *E. zadanskyi* or some deeply-branched lithostrotians like *L. pandafilandi*. Small deviations in the position of *B. salgadoi* do not coerce large differences in the phylomorphospaces, nor does it seem to increase the morphological variance in any of the main PCs.

The hind limb of *Dreadnoughtus schrani* exhibits some degree of lithostatic deformation, especially the morphology of the femoral distal end (*Ullmann and Lacovara, 2016*). The femur distal end is medially rotated with an inward deflection of the condyles. As the authors suggest, this particular rotation, apart from being characteristics, does not correlate well with any biomechanical advantage and may not be biologically produced but rather taphonomic (*Ullmann and Lacovara, 2016*). The impact on our analyses is noticeable as *D. scharani* is recovered in a particularly different

plotting compared to other deeply-branching titanosaurs due to this medial rotation of the distal end of the femur and a slight distal bevelling of the condyles (PC2; *Appendix 2—figure 2*). This extreme plotting is due to the taphonomic deformation and cannot be interpreted as a product of any evolutionary process. In fact, PC2 exhibits no significant phylogenetic signal, and whether it is due to the effects of taphonomic confounding factors or not is beyond the scopes of the current study.

Another factor observed in our sample is the lack of some information on the proximal and distal ends of some tibia. Especially the distal end, which has been identified as one of the areas of the hind limb that contributes more to the sample variance (e.g. lateromedial expansion of the distal end in PC1, *Appendix 2—figure 1*; anteroposterior orientation of the anterior ascending process on the distal condyle of the tibia in PC2, *Appendix 2—figure 2*). Among the sampled specimens, *Aeolosaurus* sp. (specimen MPCA-27100–8) has lost part of the lateral distal end of the tibia as well as part of the posterior ascending process of the distal condyle. Our reconstruction exhibits a conservative assumption of a rounded distal end with lateromedial width approximately the same as the anteroposterior width (*Appendix 2—figure 11*). It's uncertain if the lateral view of the distal end may be expanded as in *Elaltitan lilloi* (PVL-4628). But we can observe the morphology of the fossa between both ascending processes in medial view (*Appendix 2—figure 11*) and the assumption of posterior orientation of the anterior ascending process of the virtual restoration fits the information preserved in the distal end of MPCA-27100–8.

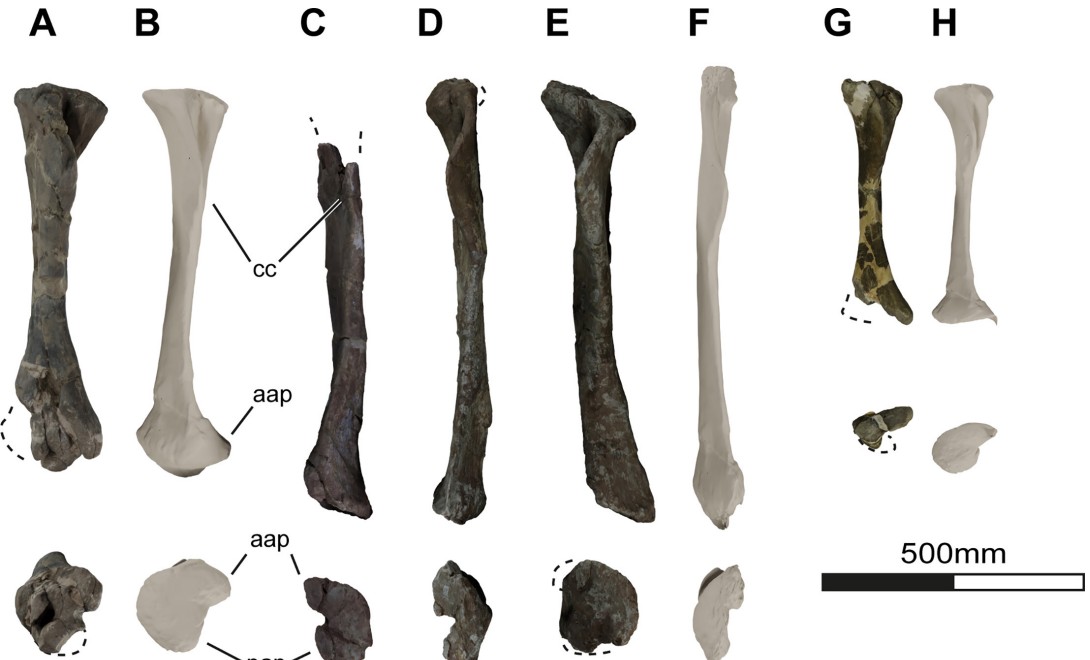

**Appendix 2—figure 12.** Comparison between tibial specimens and their corresponding 3D virtual statistical restorations. *Aeolosaurus* sp. (**A**) Left tibia MPCA-27100–8 in anterior and distal views, (**B**) Complete tibia as reconstructed by the statistical virtual restoration method in anterior and distal views. *Mendozasaurus neguyelap*: (**C**) Right tibia IANIGLA-73–2 in anterior and distal views, (**D**). Right tibia IANIGLA-73–3 in anterior and distal views, (**E**) Right tibia IANIGLA-74–1 in anterior and distal views, (**F**) Complete tibia as reconstructed by the statistical virtual restoration method in anterior and distal views. *Magyarosaurus* spp. (**G**) Left tibia NHM-R3853 in anterior and distal views, (**H**) Complete tibia as reconstructed by the statistical virtual restoration method in anterior and distal views. Abbreviations: aap – anterior ascending process, cc – cnemial crest, pap – posterior ascending process. Discontinued lines/fill indicate fracture.

The extremely gracile hind limb of *Mendozasaurus neguyelap* exhibits signs of lithostatic warping as longitudinal cracks and hints of shaft crushing, slight curvature of the midshaft, etc. The analyses recover *M. neguyelap* at some extremes of the robust-gracile hind limb spectrum (e.g. PC1-PC2; *Appendix 2—figures 1 and 2*). However, several features like the rotation and morphology of the femoral and tibial proximal and distal ends seem slightly affected and thus translate to fewer

taphonomic variance in the results of the PCs (e.g. PC1 as seen in *Appendix 2—figure 1*). Such a case is the rotation and shape of the distal end of the tibia, as seen in the specimen IANIGLA 73–2 which is completely preserved, but the incomplete specimens IANIGLA 73–3 and 74–1 also exhibit the lateromedially narrow with rounded distal end, exhibiting an anterior and posterior ascending process slightly rotated toward posterior in distal view (*Appendix 2—figure 12*). The lithostatic warping did not produce any sigmoidal component that might affect the proximodistal position or rotation of any feature. Among our results, PC2 and PC4 seem to be the most affected by this taphonomic deformation (see *Appendix 2 – figure 4-5*), but these PCs exhibit no phylogenetic signal nonetheless (*Table 6*). It seems, though, that some of our results that exhibit large variance in the morphology of the tibial proximal end may be altered by the reconstruction method (PC4-5, see *Appendix 2 – figure 4-5*), as the complete tibia IANIGLA-74–1 exhibits a slightly lateromedial-wider tibial proximal end compared with the reconstructed specimen. However, they sum up to a small portion of the total sample variance (PC4-PC5 sum up to 15.85% of the total variance) nonetheless. Similarly, *Muyelensaurus pecheni* exhibits a gracile and elongated hind limb and has also suffered from lithostatic compaction. However, all the specimens referred to *M. pecheni* from the same site exhibit some degree of crushing of the shaft, similar morphology of the proximal and distal ends, including a posterior rotation of the anterior and posterior ascending processes (e.g. MAU-PV-161 and 162). There is some room for lithostatic-induced deformation as seen in the anterior fossa of the distal end, before the anterior ascending process, as seen in specimen MAU-PV-161 compared to the same fossa exhibited by specimens MAU-PV-162 and 430. But none of them suggest that either the proximal or distal ends rotations are affected, as may be the case of the distal end of the femur of *Dreadnoughtus schrani* (*Ullmann and Lacovara, 2016*).

The hind limb of *Magyarosaurus* is based on the classical collections from *Nopcsa, 1915*. This material is sometimes fragmentary and scarce, with especial focus on the tibia as there is only one in the digitized sample (specimen NHM-R3853 referred to '*Magyarosaurus hungaricus*'). This also poses problems as *Magyarosaurus dacus* is an accepted taxon, but *M. hungaricus* and *M. transsylvanicus* have been proposed as invalid taxa and preliminarily referred to *M. dacus* (*Upchurch et al., 2004*). Among the dozen of individual recovered coming from different Maastrichtian fossil sites (*Nopcsa, 1915*), the re-study of the fossil remains deposited in the Romanian collections provides insight that some of the taxa referred to *Magyarosaurus* spp. might be chimeras and more than the original three morphotypes have been preliminary identified (*Díez Díaz et al., 2021a*). Assessment of the taxonomic distribution of titanosaurs from the Densuş-Ciula Formation (Maastrichtian) of Romania is still in debate (*Díez Díaz et al., 2023a*; *Mocho et al., 2019a*; *Mocho et al., 2024a*) and beyond the scopes of this study. For our study, we reconstructed the hind limb of *Magyarosaurus* based on the combined sample of specimens previously referred to this taxon regardless of its current taxonomic status, thus this reconstruction may be biased by an unknown interspecific variability and known intraspecific variability like size differences between the recovered individuals (e.g. see 'Effects of the ontogenetic status of the smaller specimens on the body size proxy and the shape analyses' below).

Despite being highly fragmentary, none of the studied specimens exhibit effects of lithostatic deformations imposed in the bone ends nor warping of the shafts. In general, *Magyarosaurus* spp. specimens are only affected by lack of preservation in several areas, up to the extreme of hindering the available sample. The tibial morphology has suffered the most from this lack of good preservation, as some other specimens were recovered by *Nopcsa, 1915*, but lack the proximal half (e.g. specimen NHM-R3859) or are only a fragment of the shaft (e.g. specimen NHM-R3849 referred to *M. dacus*). Lack of information in some distal or proximal ends in small percentages has not posed a problem before (*Páramo, 2020*) even if they represent missing information in the entire taxon (e.g. *Bonitasaura salgadoi* lack of the proximal end of the femur as discussed before). But large portions of missing information, especially in the same taxon, have conducted our methods to errors and incapability of the algorithm to recover a stable solution, thus not returning any estimated landmark configuration. For this matter, we opted to use the left tibia NHM-R3853 previously referred to *M. hungaricus*. Robustness differences in this specimen are not appreciated or are not enough informative to be referred to a different taxon (APB pers. Obs.). Relative robustness does not differ enough to pose a different result in our analyses, as the position of *Magyarosaurus* plotted in the taphomorphospaces may not vary greatly. It is still noticeable that there is a small mesh reconstruction spike in the landmark corresponding to the anterior ascending process (*Appendix 2—figure 12H*)

but only in the resulting restored specimen mesh. The analyzed landmarks are not affected by this mesh artifact produced during the restoration procedure in order to produce a series of complete specimen reconstructions for easily viewing the morphological variation. Lack of information in the posterior ascending process (*Appendix 2—figure 12G*) is more important; we can guess that the distal end is anterolaterally oriented. However, lack of posterior ascending process and specimen virtual restoration is the only source of information for the articulation of the fibula. The impact of our own reconstruction of the *Magyarosaurus* hind limb based on the virtual restoration of the left tibia NHM-R3853 is larger than any of the other taphonomic factors as seen in the specimen alone. A slightly more rotated fibula due to any other configuration of the posterior ascending process of the distal end of the tibia may produce less variance along PC3 (*Appendix 2—figure 3*) on our analyses and decrease the scores along this axis, thus plotting *Magyarosaurus* in less negative PC3 values and also slightly centering the rest of the sample towards zero-score. But given the position of taxa such as *Euhelopus zdanskyi*, this effect on the whole PC3 may be less appreciable in the plotting of the other taxa.

The specimens from the Bellevue area usually exhibit some degree of lithostatic compaction that usually translates to anteroposteriorly narrower proximal and distal ends of the specimens as well as some anteroposterior crushing of the shafts. It is also important to note that there are at least two different morphotypes that may be attributed to two different taxa in the sample of Bellevue area with noticeable differences in the limb morphology (*Díez Díaz et al., 2023b*; *Vila et al., 2012*). For this reason, we used only the specimens described by *Le Loeuff, 2005* with the hypothesis that this sample is referable solely to *A. atacis* and none to the second morphotype from Bellevue area (cf. *Lirainosaurus* after *Vila et al., 2012*). Taxonomic assessment is beyond the scopes of this work, and this assumption will not include problematic specimens of the second gracile form. The specimens included in the hind limb of *A. atacis* exhibit the anteroposterior crushing, and the proximal and distal end may be slightly anteroposteriorly narrower due to lithostatic compaction. The proximal end of the right femur MDE-C3-87 is extremely anteroposteriorly narrow, but despite some degree of compaction, the proximal end may be lightly warped, as the distal ends are well preserved and do not suffer noticeable deformation. Similarly, the distal elements like the tibial proximal and distal ends are well preserved and are not compressed in any direction. The fibula also exhibits the anteromedial deflection of the anterior crest (e.g. MDE-C3-48), allowing us to disregard that some of the lateromedially narrow fibular shafts may also be a product of lithostatic compaction. Despite the fragmentary nature of many of the Bellevue specimens, no taphonomic deformation may have a great impact in our analyses.

Whereas there are several femoral and tibial specimens of *Antarctosaurus wichmannianus*, there is only one left fibula, specimen MACN-6804-21, which exhibits some degree of fracturing and taphonomic deformation of the shaft. The distal end has an oblique anteroposterior fracture and some posterior concavity produced due to warping of the distal third of the shaft. This may affect the plotting of *A. wichmannianus* with a slightly straighter fibula instead and thus closer to other deeply branched non-colossosaurian non-aeolosaurini titanosaurs (e.g. in PC1; *Appendix 2—figure 1*). The overall results may not alter, but if *A. wichmannianus* exhibits less deflection of the fibula, it would remark much more the simplesiomorphic beam-like, less lateromedially wider morphology of the colossosaurian and aeolosaurini hind limbs compared to other deeply branched titanosaurians.

In general, we opted for light discussion and conservative conclusions in the evolutionary implications of the fibular rotation in lateral view, as many hind limbs of our study are affected by taphonomy, not directly in the shape of the fibula, but in lack of information on the articulation between the tibia and fibula. It is also important to note that this study lacks the analysis of the astragalus, and some of the titanosaurs do not preserve one. The astragalus will dictate the distal articulation between the fibula and tibial pair (see *Upchurch et al., 2004*). The lack of astragalus imposes a bias in how several of the hind limbs with tibia and fibula of subequal length (e.g. *Lirainosaurus astibiae*, most of Colossosauria sampled in this study, among others) articulates despite lack of apparent sigmoidal morphology despite an anterior projection of the distal end. The articulation with the femur, tibial, and fibular distal condyles coerces the height of the fibular proximal end, at the same height of the tibial proximal end or slightly below. The position and rotation of the tibial ascending processes of the distal end were taken into account. However, the lack of astragalus does not allow us to observe complex articulations beyond the medial fibular articulation with the available elements

of the tibia. In the future, we would like to expand on the biomechanical implications of the different configuration with the additional evolutionary information of the macronarian astragalus. However, it is beyond the scopes of this study, and the sample is lacking to include in our current analyses.

## Effects of the ontogenetic status of the smaller specimens on the body size proxy and the shape analyses

The assessment of the presence of potential juvenile or sub-adult specimens in the sample is important as size differences due to differences in ontogenetic status may alter greatly the results of our analyses. In order to control this altering factor, we excluded juvenile and early-juvenile specimens, considering only sub-adult specimens that exhibit some morphological features that indicate that they may have reached their maximum body size.

The relative age of a sauropod individual is usually established based on the standardized Histological Ontogenetical Stages (*Klein and Sander, 2008*). These stages were initially established through the histological analysis of sauropod long bones (*Klein and Sander, 2008*; *Mitchell et al., 2017*; *Padian et al., 2001*; *Sander et al., 2011b*) but can be determined using other parts of the skeleton such as the ribs (*Klein et al., 2012*; *Waskow and Mateus, 2017*; *Waskow and Sander, 2014*). The HOS are usually employed to determine life history and growth curves (*Klein et al., 2012*; *Klein and Sander, 2008*). Thanks to the advances in paleohistology, they have allowed us to observe that sauropods grow extremely fast (*Sander et al., 2011a*), acquiring their body size earlier than their full maturity in a few years (*Sander et al., 2011a*; *Sander et al., 2011b*). A 'young adult' or probably equivalent to Histological Ontogenetic Stage of 11 or more (*Klein and Sander, 2008*; *Stein et al., 2010*) can be safely regarded as having reached its full size. Also, recent studies indicate that titanosaurs exhibit precocial growth, and their morphology is acquired and locked at early postnatal ontogenetic stages (*Curry Rogers et al., 2016*). We opted to exclude smaller specimens, or those which exhibited hints of being subadult specimens (following the criteria established in *Griffin and Nesbitt, 2016*; *Ikejiri, 2004*; *Ikejiri et al., 2005*; applied to titanosaurs as in *Páramo et al., 2020*) from our sample prior to obtaining the theoretical reconstructions of each taxon hind limb nonetheless.

We used the holotype hind limb of *Lohuecotitan pandafilandi* to include it in our database, despite Morphotypi I of Lo Hueco site being suggested to pertain to the same taxa (*Páramo et al., 2022*) and none of the smaller, putative juvenile specimens differing from the larger specimens nor the holotype elements (*Páramo et al., 2022*). The probable juvenile specimens referred to *Lirainosaurus astibiae* are all teeth or fore limb specimens (*Díez Díaz, 2013*) and were not considered in the multivariate statistic estimation of the body size.

*Muyelensaurus pecheni* may be problematic to assess which one may pertain to a probable juvenile or subadult specimen without a proper histological sample, as it has been proposed that there are some juvenile individuals in the sample of at least five individuals (*Calvo et al., 2007*). However, there are no size differences among the appendicular elements. Neither does any of the elements exhibit features that may indicate subadult status or related to earlier ontogenetic stage according to the criteria mentioned before.

There are some smaller specimens among the sample of appendicular elements of *Saltasaurus loricatus* that may pertain to putative juvenile individuals (see *Powell, 2003*). Specimens like the smaller tibia PVL-4017-87 exhibit not only size differences but less marked scars or rugosities in the proximal and distal ends (APB pers. obs.). However, there are no morphological differences with the larger individuals, and previous analyses found these specimens plotted in the same area of the morphospace (e.g. *Páramo et al., 2020*). Their centroid size was excluded in the calculation of the mean size of each element of the operative taxonomic unit hind limb. Similarly, *Neuquensaurus* spp. includes some smaller specimens that may be referred to subadult individuals. In this case, they are more difficult to assess as some of them have lost the proximal and distal end, where usual criteria of the articular cap rugosity may apply in absence of histological sampling. We included all the sample specimens from previous analyses, despite some of them, like the tibiae MLP-CS-1123 and MLP-CS-1264, exhibiting slightly smaller sizes but no morphological differences in hand review nor in morphometric analyses. MLP-CS-1123 may or may not pertain to a juvenile individual as it does not preserve the proximal and distal ends, but on the other hand, MLP-CS-1264 exhibits the same rugosity expected as in an adult individual and the larger specimens of the *Neuquensaurus* sample. We opted to include their sizes in the estimation of the operative taxonomic unit hind limb

and consider the size differences as intraspecific variability not related to ontogenetic differences. However, we may take into account that *Neuquensaurus australis* and '*N. robustus*' morphological differences and taxonomic status of the latter are still in debate (*Otero, 2010*).

The sample of classical hind limb specimens referred to *Magyarosaurus* spp. (*Nopcsa, 1915*) exhibits some degree of size differences, but none of the sampled specimens may be referred to a juvenile individual, nor do their size differ greatly. Similarly to *Neuquensaurus* spp., the taxonomic status of '*Magyarosaurus hungaricus*' and '*M. transylvanicus*' is still debated (e.g. *Díez Díaz et al., 2023a*; *Mocho et al., 2024c*; *Upchurch et al., 2004*). Probable unknown interspecific variability that here is considered intraspecific variability could have greater impact than ontogenetic differences in our analyses. However, a preliminary study plotted all the *Magyarosaurus* specimens from Nopcsa's classic collection in the same area of the morphospace, closer between them (*Páramo, 2020*).

Lastly, there are smaller specimens identified in the sample of Ampelosaurus atacis. In this case, there is a lack of histological sample in all the sampled specimens, but we are more worried about the mixture of some specimens that may be referred to a second taxonomic unit. For now, there is a proposal of the presence of a second morphotype, referable to another different undescribed sauropod, cf. *Lirainosaurus* (*Vila et al., 2012*). In situ observations indicate there are large morphological differences as well as size differences between both morphotypes (*Díez Díaz et al., 2023b*; *Vila et al., 2012*) and our preliminary study found similar results in the morphometric analyses (*Páramo, 2020*). We feel that the interspecific variability due to an undescribed titanosaur mixed in the sample of *A. atacis* may have a larger impact in our analyses than the ontogenetic status of some of the appendicular specimens itself. For this reason and until the taxonomic status of the sample of *A. atacis* is assessed (e.g. *Díez Díaz et al., 2023b*), we used the referred specimen of *Le Loeuff, 2005* as the operative taxonomic unit hind limb of *A. atacis*. All the elements exhibit enough features to consider them at least subadult specimens.

