## [Editor Report · eLife Assessment]

The authors present **convincing** findings on trends in hind limb morphology through the evolution of titanosaurian sauropod dinosaurs, the land animals that reached the most remarkable gigantic sizes. The **important** results include the use of 3D geometric morphometrics to examine the femur, tibia, and fibula to provide new information on the evolution of this clade and on evolutionary trends between morphology and allometry.

---

## [Referee Report · Joint Public Review]

Páramo et al. used 3D geometric morphometric analyses of the articulated femur, tibia, and fibula of 17 macronarian taxa (known to preserve these three skeletal elements) to investigate morphological changes that occurred in the hind limb through the evolutionary history of this sauropod clade. A principal components analysis was completed to understand the distribution of the morphological variation. A supertree was constructed to place evolutionary trends in morphological variation into phylogenetic context, and hind limb centroid size was used to investigate potential relationships between skeletal anatomy and gigantism. The majority of the results did not yield statistically significant differences, but they did identify interesting shape-change trends, especially within subclades of Titanosauria. Many previous studies have attempted to elucidate a link between wide-gauge posture and gigantism, which in this study Páramo et al. investigate among several titanosaurian subclades. They propose that morphologies associated with wide-gauge posture arose in parallel with increasing body size among basal members of Macronaria and that this connection became less significant once wide-gauge posture was acquired within Titanosauria. The authors also suggest that other biomechanical factors influenced the independent evolution of subclades within Titanosauria and that these influences resulted in instances of convergent evolution. Therefore, they infer that, overall, wide-gauge posture was not significantly correlated with gigantism, though some morphological aspects of hind limb skeletal anatomy appear to have been associated with gigantism. Their work also supports previous findings of a decrease in body size within Titanosauriformes (which they found to be not significant with shape variables but significant with Pagel's lambda). Collectively, their results support and build on previous work to elucidate more specifics on the evolution of this enigmatic clade. Further study will show if their hypotheses stand or if the inclusion of additional specimens and taxa yields alternative results.

[Editors' note: One of the original reviewers, Reviewer 2, reviewed this revised version of the manuscript; they reported satisfaction with the changes made by the authors in response to the original reviewer comments.]

---

## [Author Response]

The following is the authors’ response to the original reviews.

**eLife Assessment**
The authors present valuable findings on trends in hind limb morphology throughout the evolution of titanosaurian sauropod dinosaurs, the land animals that reached the most remarkable gigantic sizes. The solid results include the use of 3D geometric morphometrics to examine the femur, tibia, and fibula to provide new information on the evolution of this clade and understand the evolutionary trends between morphology and allometry. Further justification of the ontogenetic stages of the sampled individuals would help strengthen the manuscript's conclusions, and the inclusion of additional large-body mass taxa could provide expanded insights into the proposed trends.

Most of the analyzed specimens, especially from the smaller taxa, come from adult or subadult specimens. None exhibit features that may indicate juvenile status. However, we lack information of the paleohistology that may be a stronger indicator on the ontogenetic status of the individual, and some of operative taxonomic units used in the study come from mean shape of all the sampled specimens.

Current information on morphological differences between adult and subadult or juvenile specimens indicates that even early juvenile specimens may share same morphological features and overall morphology as the adult (e.g., see Curry-Rogers et al., 2016; Appendix S3). We included a comprehensive analysis of the impact of juvenile specimens as one of the aspects of the intraspecific variability that may alter our results in Appendix S3.

**Public Reviews:**

**Reviewer #1:**
Weaknesses:Several sentences throughout the manuscript could benefit from citations. For example, the discussion of using hind limb centroid size as a proxy for body mass has no citations attributed. This should be cited or described as a new method for estimating body mass with data from extant taxa presented in support of this relationship. This particular instance is a very important point to include supporting documentation because the authors' conclusions about evolutionary trends in body size are predicated on this relationship.

We address this issue in the text (Line 32 & 64). Centroid size seems a good indication as it’s the overall size of the entire hind limb, and the length of the femur and tibia is well correlated independently with the body size/mass. Also, as we use few landmarks and only those that are purely type I or II landmarks, with curves of semilandmarks bounded or limited by them, centroid size is not sensible to landmark number differences across the sample in our study (as the centroid size is dependent of the number of landmarks of the current study as well as the physical dimensions of the specimens).

We have sampled and repeated all the analyses using other proxies like the femoral length and the body mass estimated from the Campione & Evans (2020) and Mazzeta et al. (2004) methods. The comprehensive description of the method is in Appendix S2, the alternative analyses can be accessed in the Appendix S3 and S4; and the code for the alternative analyses can be accessed in the modified Appendix S5. All offer similar results than the ones obtained in our analyses with the body size proxied with the hind limb landmark configuration centroid size.

An additional area of concern is the lack of any discussion of taphonomic deformation in Section 3.3 Caveats of This Study, the results, or the methods. The authors provide a long and detailed discussion of taphonomic loss and how this study does a good job of addressing it; however, taphonomic deformation to specimens and its potential effects on the ensuing results were not addressed at all. Hedrick and Dodson (2013) highlight that, with fossils, a PCA typically includes the effects of taphonomic deformation in addition to differences in morphology, which results in morphometric graphs representing taphomorphospaces. For example, in this study, the extreme negative positioning of Dreadnoughtus on PC 2 (which the authors highlight as "remarkable") is almost certainly the result of taphonomic deformation to the distal end of the holotype femur, as noted by Ullmann and Lacovara (2016).

We included a brief commentary in the Caveats of This Study (Line 467) and greatly expanded this issue in the Appendix S3. We followed the methodology proposed by Lefebvre et al. (2020) to discuss the effects of taphonomic deformation in the shape analyses.

Our shape variables (PCs obtained from the shape PCA) should be viewed as taphomorphospaces as Hedrick and Dodson, as well as the reviewer, points in such cases.

The analysis of the effects of taphonomy or errors induced by the landmark estimation method indicate that *Dreadnoughtus schrani* is one of the few sampled taxa that may have a noticeable impact on our analyses due lithostatic deformation. Other taxa like *Mendozasaurus neguyelap* or *Ampelosaurus atacis* may also induce some alterations to the PCs. In general, the trends of those PCs slightly altered by taphonomy, where *D. scharni* is the only sauropod that may alter an entire PC like PC2, did not exhibit phylogenetic signal and are a small proportion of the sample variance.

The authors investigated 17 taxa and divided them into 9 clades, with only Titanosauria and Lithostrotia including more than two taxa (and four clades are only represented by one taxon). While some of these clades represent the average of multiple individuals, the small number of plotted taxa can only weakly support trends within Titanosauria. If similar general trends could be found when the taxa are parsed into fewer, more inclusive clades, it would support and strengthen their claims. Of course, the authors can only study what is preserved in the fossil record, and titanosaurian remains are often highly fragmentary; these deficiencies should therefore not be held against the authors. They clearly put effort and thought into their choices of taxa to include in this study, but there are limitations arising from this low sample size that inherently limit the confidence that can be placed on their conclusions, and this caveat should be more clearly discussed. Specifically, the authors note that their dataset contains many lithostrotians, but they do not discuss unevenness in body size sampling. As neither their size-category boundaries nor the taxa which fall into each of them are clearly stated, the reader must parse the discussion to glean which taxa are in each size category. It should be noted that the authors include both *Jainosaurus* and *Dreadnoughtus* as 'large' taxa even though the latter is estimated to have been roughly five times the body mass of the former, making *Dreadnoughtus* the only taxon included in this extreme size category. The effects that this may have on body size trends are not discussed. Additionally, few taxa between the body masses of Jainosaurus and *Dreadnoughtus* have been included even though the hind limbs of several such macronarians have been digitized in prior studies (such as *Diamantinasaurus* and *Giraffititan*; Klinkhamer et al. 2018). Also, several members of Colossosauria are more similar in general body size to *Dreadnoughtus* than *Jainosaurus*, but unfortunately, they do not preserve a known femur, tibia, and fibula, so the authors could not include them in this study. Exclusion of these taxa may bias inferences about body size evolution, and this is a sampling caveat that could have been discussed more clearly. Future studies including these and other taxa will be important for further evaluating the hypotheses about macronarian evolution advanced by Páramo et al. in this study.

Sadly, we could not include some larger sized titanosaurians sauropods. As the reviewers points out, the lack of larger sauropods among the sampled taxa may hinder our results, as the “large-bodied” category is filled with some mid-sized taxa and the former *Dreadnoughtus schrani* which is five times larger than some of them. We tried to include *Elaltitan lilloi*, digitized for this study and included in preliminary analyses, but the fragmentary status increased greatly the error by the estimation method as there is only a proximal third or mid femur preserved from this taxon. Therefore we opted to exclude it from our database.

Other taxa considered, as the reviewer suggest, was not readily available for the authors as the time of this study was conducted and including now may have increased the possible bias of our study. *Giraffatitan brancai* is an Late Jurassic brachiosaurid, which may again increase the number of early-branching titanosauriforms with large body masses while most of the smaller taxa sampled are recovered in deeply-branching macronarians (including *Diamantinasaurus matildae* if we would have also included it). Future analyses may include a wider sample of the mid to large-bodied titanosaurians, especially lithostrotians, as well as some colossosaurs like *Patagotitan mayorum*.

**Reviewer #1 (Recommendations For The Authors):**
These are all minor comments that would improve the manuscript.- There are a few typos throughout the manuscript such as: line 70 should be 2016 and line 242 should be forelimb.

Corrected.

- To me, the most interesting aspect of your study is the diversity and trends recovered in titanosaurian subclades and I would highlight this, not gigantism, in the title if you choose to revise the title.

It has been addressed. The specificality of some of the tests and the implication to the acquisition of the spread limb posture and gigantism in early-branching taxa is important nonetheless, so we think that it may remain in the title.

- The abstract should provide more details on the results such as none of the listed trends were statistically significant.

Many of the trends exhibit phylogenetic signal, but not the allometric components. We have briefly addressed them.

- Several sentences in the manuscript need citations such as: line 48 the reference to other megaherbivores, line 66 the discussion of poor understanding of the relationship of wide gauge posture and gigantism, and the use of centroid size as an estimate of body mass (see Public Review).

We changed the line 66 to improve the focus on the current state of the art in the hypothesis of a relationship between arched limbs and in the increase of body size. We included a section relating centroid size as a proxy (due the good correlation between the femur and tibia length and the body mass) and the caveats of using it. We also expanded in the Appendix S2 the use of centroid size and the alternative models.

- With titanosaur evolution, you mention that they are adapting to new niches and topography (line 64). What support is there for this versus they are adapting to be more successful in their current environment?

Noted, we have changed the phrase to improved efficiency exploiting of inland environments, as thy can be either opening new inland niches or adapting better to current inland niches that were already exploited for less deeply branching sauropods. However, its testing is beyond the scope of the current work.

- Line 384-385: the discussion of *Rapetosaurus* should mention that it is a juvenile and some studies have suggested that titanosaur limbs grow allometrically.

We have included a small line. Whether *Rapetosaurus krausei* exhibit allometric growth or not may not change greatly the discussion, maybe only excluding it as morphologically convergent to *Lirainosaurus* and *Muyelensaurus*. But if that so, it will be further proof that small-sized titanosaurs exhibit the robust skeleton expected in the giant titanosaurs.

- I would consider addressing the question of if we are certain enough in our understanding of titanosaurian phylogeny to rule out homology, especially when you discuss the uncertainty of the placement of specific taxa. Also, *Diamantinasaurus* is not the only titanosaur that has been proposed as a member of both basal and more derived subclades (e.g., *Dreadnoughtus*).

We tried to assume a more conservative approach. We could not fully rule out that some of the features observed in the sampled deeply branching lithostrotians, especially saltasauroids, cannot be present in the entire somphospondylan lineage. However, none of the less deeply-branching or early-branching titanosaurs exhibit this kind of morphology. Recent studies propose the possibility that entire groups, included in this study like the Colossosauria, change its position in the phylogeny. However, despite the debated phylogenetic position of *Diamantinasaurus* or *Dreadnoughtus*, or even the inclusion of Colossosauria within the saltasauroids and the inclusion of the Ibero-Armorican lithostrotians as putative saltasaurids (Mocho et al. 2024). However, even considering these changes we did not notice any relevant differences in our conclusions about hind limb arched morphology nor about size. Distal hind limb overall robustness should indeed be addressed in the light of shifts in phylogenetic position and include some interesting sauropods like *Diamantinasaurus* or expand the large-sized Colossosauria or early-branching somphospondyls as it may have profound implications on the morphofunctional adaptations to specific feeding niches, e.g., see current hypotheses about rearing as mentioned in Bates et al. (2016), Ullmann et al. (2017) or Vidal et al. (2020). We had not enough information to conclude the presence of any plesiomorphic condition or analogous feature with our current sample and the debated titanosaurian phylogeny.

- I understand this is not standard in the field, but your study provides the opportunity to conduct sensitivity testing of the effects of cartilage thickness and user articulation of the bones on PCA results. This would be an inciteful addition to the field of GMM.

We are currently developing such a comprehensive analysis and several other implications on our past results. However, we feel that it is beyond the scope of the current study. We appreciate the suggestion nonetheless, as it would be a sensitivity test of the impact of several of our assumptions in the final results that is often not considered.

- In Figure 1, if all the limbs were arranged the same way it would be easier to interpret. Consider flipping panels B and D to match A and C.

Accepted.

- In Figures 2-4, the views in C should be labeled in the figure or caption. Oceanotitan is also in the PCA plot but not included in the figure caption. Also, consider changing the names to represent the paraphyletic groupings you are using instead of formal clade names. For example, change 'Titanosauria' to 'Basal Titanosaurs' to reflect that it is not including all titanosaurs in the sample.

Changes accepted for the shape PCA results. The informal (i.e., paraphyletic) terms such as “Basal Titanosaurs” were only used in the shape analyses as in the RMA, the Titanosauria (and other more inclusive groups) were used as natural groups. Each partial RMA model is based on a sample of all the taxa that are included within that particular clade (e.g., Titanosauria includes both *Dreadnoughtus* and *Saltasaurus*; *Lithostrotia* excludes the former).

- I am concerned that centroid size does not scale evenly across the wide-ranging body mass of titanosaurs. I do not know if this affects your size trends or their significance, but as I mentioned above *Dreadnoughtus* is much bigger than most of the taxa included and that isn't as drastically apparent in centroid size (in Figure 5) as it is when taxa are plotted by body mass.

Main problematic with centroid size of the hind limb is the shift in the body plan of deeply-branching titanosaurs as the Center of Masses is displaced toward the anterior portion of the body and it has been proposed due a large development of the forelimb region (e.g., Bates et al. 2016). However, it would only increase the effects of the phyletic body size reduction, as smaller taxa tend to have a 1:1 fore limb and hind limb ratio, e.g., from our past analyses as in Páramo et al. (2019), and the sacrum is not as beveled as in earlier somphospondyls, e.g., Vidal et al. (2020). The role of the low-browsing feeding habits of deeply-branching lithostrotians shall be explored elsewhere, as it may be the main driving force of this effect. Our point is, the proxy used may have some slight offset due some high-browsing giant early-branching titanosaurs which has a greater cranial region development which increase its body size and mass beyond our bare-minimum estimation based on the hind limb region. But, overall, this offset is assumed to be low. We repeated the analyses with the femoral length as proxy of body size and a mass estimation, including the quadratic equation based on both humeral and femoral lengths, and the results remain similar. Another problem that arises with the use of centroid size is the way it shall be calculated, but as we used an even number of landmarks and curve semilandmarks, and all of them bounded to anatomical features, it remains equal at least for our sample (but cannot be extrapolated to other geometric morphometric studies that do not use the same configurations)

We appreciate the reviewer concerns nonetheless, as it was on of our own when designing this study, and we in the future will try to expand the analyses, or advise anyone expanding on this study, using total body size/volume estimations following Bates et al. (2016). Which also includes test of the effects of the different whole-body estimation models.

Cites:

Bates KT, Mannion PD, Falkingham PL, Brusatte SL, Hutchinson JR, Otero A, Sellers WI, Sullivan C, Stevens KA, Allen V. 2016. Temporal and phylogenetic evolution of the sauropod dinosaur body plan. Royal Society Open Science 3:150636. doi:10.1098/rsos.150636

Mocho P, Escaso F, Marcos-Fernández F, Páramo A, Sanz JL, Vidal D, Ortega F. 2024. A Spanish saltasauroid titanosaur reveals Europe as a melting pot of endemic and immigrant sauropods in the Late Cretaceous. Commun Biol 7:1016. doi:10.1038/s42003-024-06653-0

Páramo A, Ortega F, Sanz JL. 2019. A Niche Partitioning Scenario for the Titanosaurs of Lo Hueco (Upper Cretaceous, Spain). International Congress of Vertebrate Morphology (ICVM) - Abstract Volume, Journal of Morphology. Prague. p. S197.

Ullmann PV, Bonnan MF, Lacovara KJ. 2017. Characterizing the Evolution of Wide-Gauge Features in Stylopodial Limb Elements of Titanosauriform Sauropods via Geometric Morphometrics. The Anatomical Record 300:1618–1635. doi:10.1002/ar.23607

Vidal D, Mocho P, Aberasturi A, Sanz JL, Ortega F. 2020. High browsing skeletal adaptations in Spinophorosaurus reveal an evolutionary innovation in sauropod dinosaurs. Sci Rep 10:6638. doi:10.1038/s41598-020-63439-0

**Reviewer #2:**
The authors report a quantitative comparative study regarding hind limb evolution among titanosaurs. I find the conclusions and findings of the manuscript interesting and relevant. The strength of the paper would be increased if the authors were to improve their reporting of taxon sampling and their discussion of age estimation and the potential implications that uncertainty in these estimates would have for their conclusions regarding gigantism (vs. ontogenetic patterns).

Considering the observations made by reviewer #1, we included a data about the impact of ontogenetic patterns and other intraspecific variability in the Appendix S3. We considered to increase the sample but it has not been possible at the time of this study was carried out.

**Reviewer #2 (Recommendations For The Authors):**
I have a few concerns/requests for the authors, that I hope can be easily resolved.Comments:- What drove taxon sampling?

Random sampling of somphospondylan sauropods focused on the Lithostrotia clade for the thesis project of one of the authors, APB. Logistics were also one of the bias on our sample, and based on the available titanosaurian material we left out several macronarians that has been already sampled but would further induce a early-branching large sauropod, deeply-branching small sauropod that may alter our results.

- Which phylogenies were used to create the supertree applied to the analyses? What references were used to time-calibrate the tips and deeper nodes? I couldn't find any reference to this. Additionally, more information regarding the R packages and analytical pipeline would be appreciated: e.g. were measurements used in the analyses log-transformed?

A comprehensive description of the methodology is provided in Appendix S2.

- Age estimate: can the author confirm the skeletal maturity of the sampled individuals? If this is not the case, how can the author be sure that the patterns towards gigantism are not reflecting different ontogenetic stages? I believe this should be part of both methods and discussion.

As commented before, we excluded small, probable juvenile specimens from our sample. We have no paleohistological sample backing the claims of the ontogenetic status of some of the specimens that were included or excluded were calculating the mean shape for the operative taxonomic units. However, we followed a criteria to identify the relative ontogenetic status and it has been included in Appendix S3.

- The authors used the centroid size for regressions in Figure 6. Although I believe that this is a good variable, would the author be willing to use body mass and log-transformed femur length in addition to what was done? These would be very useful considering that these variables are (relatively) independent from shape/morphology.

Accepted, we tested our hypotheses with three alternative models based on femoral length, combined femoral and humeral lengths for body mass estimations. Methodology can be found in Appendix S2, results on Appendix S4, code for the alternative methods in Appendix S5.

- Data access: will stl. Files of the limb elements be shared and freely available? In this case, where the files will be deposited?

At the time of the current study, some of the sampled specimens cannot be available (material under study) but the mean shapes can be generated after the landmarks and semilandmark curves and the “atlas” mesh.

- Additionally, outstanding references regarding limb evolution, GMM, role of ontogeny, and evolution of columnar gait are missing. The authors should reinforce the literature review with the following (alphabetical order):Bonnan, M. F. (2003). The evolution of manus shape in sauropod dinosaurs: implications for functional morphology, forelimb orientation, and phylogeny. Journal of Vertebrate Paleontology, 23(3), 595-613.Botha, J., Choiniere, J. N., & Benson, R. B. (2022). Rapid growth preceded gigantism in sauropodomorph evolution. Current Biology, 32(20), 4501-4507.Curry Rogers, K., Whitney, M., D'Emic, M., & Bagley, B. (2016). Precocity in a tiny titanosaur from the Cretaceous of Madagascar. Science, 352(6284), 450-453.Day, J. J., Upchurch, P., Norman, D. B., Gale, A. S., & Powell, H. P. (2002). Sauropod trackways, evolution, and behavior. Science, 296(5573), 1659-1659.Fabbri, M., Navalón, G., Benson, R. B., Pol, D., O'Connor, J., Bhullar, B. A. S., ... & Ibrahim, N. (2022). Subaqueous foraging among carnivorous dinosaurs. Nature, 603(7903), 852-857.Fabbri, M., Navalón, G., Mongiardino Koch, N., Hanson, M., Petermann, H., & Bhullar, B. A. (2021). A shift in ontogenetic timing produced the unique sauropod skull. Evolution, 75(4), 819-831.González Riga, B. J., Lamanna, M. C., Ortiz David, L. D., Calvo, J. O., & Coria, J. P. (2016). A gigantic new dinosaur from Argentina and the evolution of the sauropod hind foot. Scientific Reports, 6(1), 19165.Lefebvre, R., Allain, R., & Houssaye, A. (2023). What's inside a sauropod limb? First three‐dimensional investigation of the limb long bone microanatomy of a sauropod dinosaur, Nigersaurus taqueti (Neosauropoda, Rebbachisauridae), and implications for the weight‐bearing function. Palaeontology, 66(4), e12670.McPhee, B. W., Benson, R. B., Botha-Brink, J., Bordy, E. M., & Choiniere, J. N. (2018). A giant dinosaur from the earliest Jurassic of South Africa and the transition to quadrupedality in early sauropodomorphs. Current Biology, 28(19), 3143-3151.Martin Sander, P., Mateus, O., Laven, T., & Knötschke, N. (2006). Bone histology indicates insular dwarfism in a new Late Jurassic sauropod dinosaur. Nature, 441(7094), 739-741.Remes, K. (2008). Evolution of the pectoral girdle and forelimb in Sauropodomorpha (Dinosauria, Saurischia): osteology, myology and function (Doctoral dissertation, München, Univ., Diss., 2008).Sander, P. M., & Clauss, M. (2008). Sauropod gigantism. Science, 322(5899), 200-201.Yates, A. M., & Kitching, J. W. (2003). The earliest known sauropod dinosaur and the first steps towards sauropod locomotion. Proceedings of the Royal Society of London. Series B: Biological Sciences, 270(1525), 1753-1758.

We appreciate this suggestion and we already used some of the articles in our study but the selection of cites were based also in the available manuscript space enforced by the edition guidelines. We would have like to include several of these works but we had opted to include some of the works that summarize some of them, whereas excluding others.